# Contrasting signatures of genomic divergence during sympatric speciation

Andreas F. Kautt[1,4,9], Claudius F. Kratochwil[1,9], Alexander Nater[1,9], Gonzalo Machado-Schiaffino[1,5], Melisa Olave[1,6], Frederico Henning[1,7], Julián Torres-Dowdall[1], Andreas Härer[1,8], C. Darrin Hulsey[1], Paolo Franchini[1], Martin Pippel[2,3], Eugene W. Myers[2,3] & Axel Meyer[1✉]

The transition from 'well-marked varieties' of a single species into 'well-defined species'—especially in the absence of geographic barriers to gene flow (sympatric speciation)—has puzzled evolutionary biologists ever since Darwin[1,2]. Gene flow counteracts the buildup of genome-wide differentiation, which is a hallmark of speciation and increases the likelihood of the evolution of irreversible reproductive barriers (incompatibilities) that complete the speciation process[3]. Theory predicts that the genetic architecture of divergently selected traits can influence whether sympatric speciation occurs[4], but empirical tests of this theory are scant because comprehensive data are difficult to collect and synthesize across species, owing to their unique biologies and evolutionary histories[5]. Here, within a young species complex of neotropical cichlid fishes (*Amphilophus* spp.), we analysed genomic divergence among populations and species. By generating a new genome assembly and re-sequencing 453 genomes, we uncovered the genetic architecture of traits that have been suggested to be important for divergence. Species that differ in monogenic or oligogenic traits that affect ecological performance and/or mate choice show remarkably localized genomic differentiation. By contrast, differentiation among species that have diverged in polygenic traits is genomically widespread and much higher overall, consistent with the evolution of effective and stable genome-wide barriers to gene flow. Thus, we conclude that simple trait architectures are not always as conducive to speciation with gene flow as previously suggested, whereas polygenic architectures can promote rapid and stable speciation in sympatry.

Speciation has long been assumed to require geographic barriers that limit the homogenizing effects of gene flow (allopatric speciation; but see ref. [6]). Recently, the recognition that speciation can also occur in the absence of geographic barriers (under sympatric conditions) has increased, although only a few empirical examples are widely accepted[1]. Sympatric speciation is the most extreme form of speciation with gene flow. Theoretical models predict that speciation with gene flow is strongly facilitated if traits under divergent selection also contribute to assortative mating[4,6–8]. Moreover, if divergently selected traits are based on few instead of many loci, speciation with gene flow is generally thought to occur more readily, assuming that per-locus effects are larger and that selection acting on these loci will be stronger. In addition, with fewer loci, recombination is less likely to break up co-adapted alleles[4,9].

Genomic studies on the early stages of speciation with gene flow[10] have found that differentiation between incipient species is commonly restricted to a few genomic regions[2,11–14]. The discovery of such 'barrier loci', which resist the homogenizing effects of gene flow, fits theoretical expectations[4,6–8]. Apart from the proposed pivotal role of major effect loci for speciation, theoretical work has also suggested that many small-effect loci can jointly constitute effective genome-wide barriers to gene flow[15]. Classic multi-locus cline theory[16] and genomic simulations[17,18] support the hypothesis that the synergistic effects of many weakly selected alleles can promote a rapid buildup of pronounced genomic differentiation[5,13]. However, there is no empirical evidence that the concerted action of small-effect loci is sufficient to initiate speciation in sympatry.

Midas cichlid fishes form an extremely young species complex (13 described species; only about 16,700 years old) and occur in seven recently formed, small, and isolated crater lakes (CLs) that were colonized independently from the great lakes (GLs) Nicaragua and Managua (Fig. 1a, c). In the CLs Apoyo and Xiloá, Midas cichlids formed adaptive radiations, which are widely accepted textbook examples of sympatric speciation[19–21]. Key ecological traits that have been proposed to drive

[1]Department of Biology, University of Konstanz, Konstanz, Germany. [2]Max Planck Institute of Molecular Cell Biology and Genetics, Dresden, Germany. [3]Center for Systems Biology Dresden, Dresden, Germany. [4]Present address: Department of Organismic and Evolutionary Biology, Harvard University, Cambridge, MA, USA. [5]Present address: Department of Functional Biology, Area of Genetics, University of Oviedo, Oviedo, Spain. [6]Present address: Argentine Dryland Research Institute of the National Council for Scientific Research (IADIZA-CONICET), Mendoza, Argentina. [7]Present address: Department of Genetics, Institute of Biology, Federal University of Rio de Janeiro (UFRJ), Rio de Janeiro, Brazil. [8]Present address: Division of Biological Sciences, Section of Ecology, Behavior & Evolution, University of California San Diego, La Jolla, CA, USA. [9]These authors contributed equally: Andreas F. Kautt, Claudius F. Kratochwil, Alexander Nater. ✉e-mail: axel.meyer@uni-konstanz.de

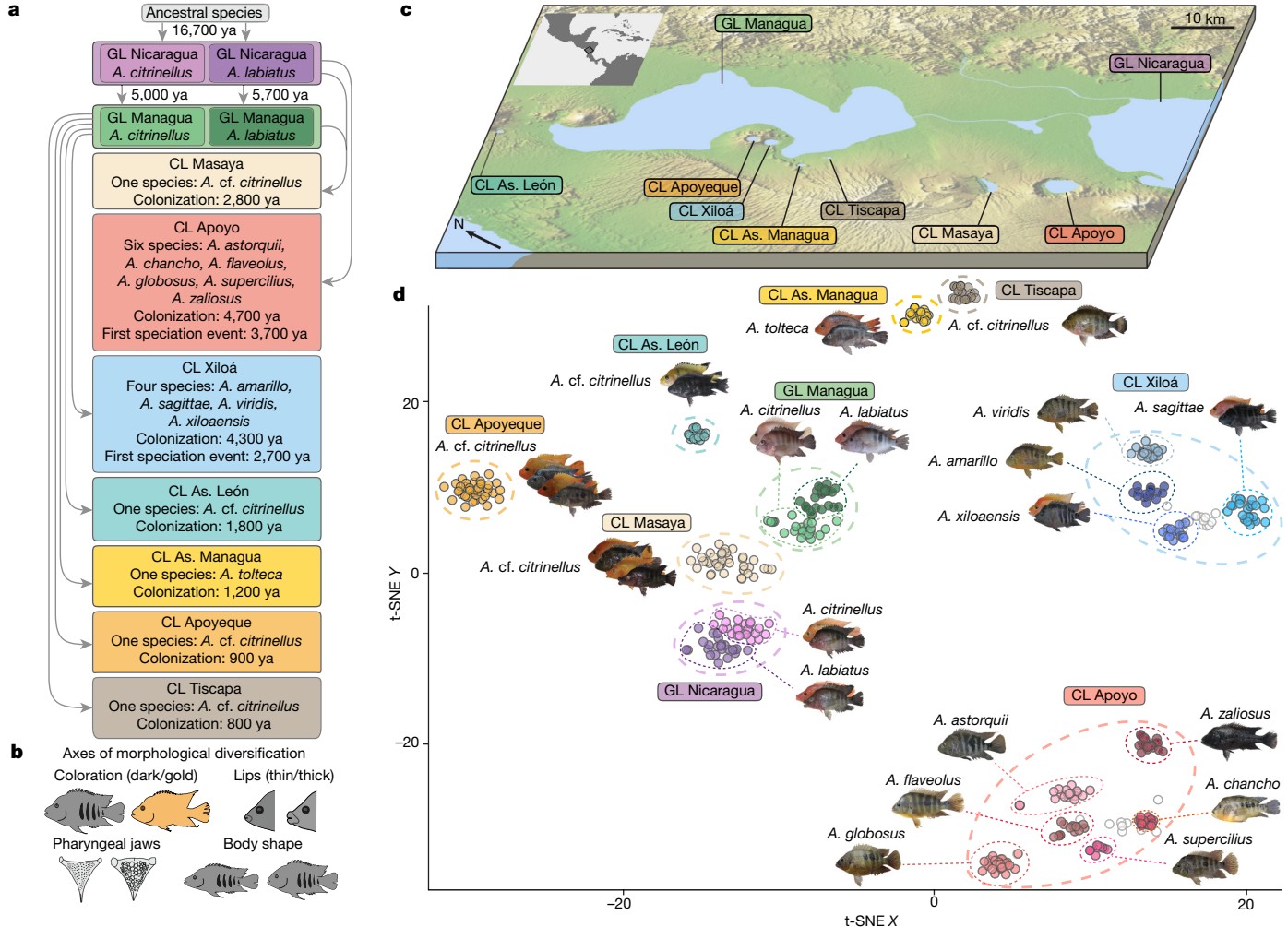

**Fig. 1 | Evolutionary relationships among all members of the Midas cichlid species complex. a**, Simplified demographic history with colonization and first sympatric speciation times (in years ago, ya) inferred from model-based coalescent simulations (Extended Data Fig. 3, Extended Data Table 1). **b**, Major phenotypic axes that have been suggested to contribute to population divergence and speciation in Midas cichlid fishes. **c**, Map of the Nicaraguan great lakes (GL) and crater lakes (CL) (image credit: NASA/JPL/NIMA).

**d**, Dimensionality reduction (t-SNE) of whole-genome genotype data reveals clustering by lake and described species (two species in GLs Nicaragua and Managua, six species in CL Apoyo, and four species in CL Xiloá). Representative specimens are shown for each species and lake population, with dark/gold and thin- or thick-lipped morphs. White circles represent individuals of mixed ancestry.

species formation include coloration, lip size, pharyngeal jaw morphology, and body shape[22] (Fig. 1b). These traits have not only been linked to divergent selection[19,20,23,24] and/or positive assortative mating[23,25,26], but have also been suggested to vary in their genetic architectures[23,27,28]. Here, we go beyond addressing the role of ecological opportunity (as examined previously[29]) and investigate whether the genetic architecture of traits under divergent selection—as suggested on the basis of theoretical work[4]—affects the propensity to form genomically diverged species.

## Phylogeny and demographic history

To reconstruct the evolutionary history of this species complex, we generated the first long-read-based Midas cichlid chromosome-level genome assembly (genome size approximately 900 Mb, contig N50 = 3.8 Mb) and high-coverage whole-genome resequencing data for 453 individuals, encompassing the entire species complex. Using this data set, we reconstructed the phylogenetic relationships among all 13 described species across all lakes and found strong support that the species flocks found in the CLs Apoyo and Xiloá

evolved from a single founder population (Extended Data Fig. 1a), as previously suggested[19–21]. Ancestry inference (Extended Data Fig. 1l–z) showed that most CL populations trace their ancestry to GL Managua, except for those in CL Apoyo (to GL Nicaragua), and CL Masaya (to both GLs). Moreover, all CL populations show ancestry contributions from both GL species *A. citrinellus* and *A. labiatus*, with similar proportions across sympatric CL species and chromosomes. Estimates of effective population size over time indicated strong bottlenecks during CL colonization (Extended Data Fig. 2). A more detailed, model-based analysis (Extended Data Fig. 3, Extended Data Table 1) suggests that the Midas cichlid complex originated only around 16,700 years ago with the split of *A. citrinellus* and *A. labiatus* in GL Nicaragua. From there, both species colonized GL Managua about 5,000 and 5,700 years ago, respectively. Notably, this timing coincides with a large underwater eruption within GL Managua[30] that probably exterminated earlier fish fauna. The CLs were colonized even more recently, only between 4,700 and 800 years ago, with founder population sizes varying from 30 to 850 individuals (Extended Data Table 1, Supplementary Notes). In agreement with previous work[21], we find that sympatric speciation in CLs Apoyo

and Xiloá was preceded by admixture from a secondary wave of colonization (Extended Data Fig. 3g, j).

## Population structure and phenotypic diversity

Investigating population structure with *t*-distributed stochastic neighbour embedding (t-SNE) shows a clear clustering of species or populations by lake of origin (Fig. 1d). Moreover, in CLs Apoyo and Xiloá, clustering agrees with the phenotypically assigned six and four species, respectively. Clustering according to the species *A. citrinellus* and *A. labiatus* was clear in both GLs, but although they diverged much earlier, differences were clearly less pronounced than among the CL species. These patterns were also recovered by principal component analyses (Extended Data Fig. 1b–k). Consistent with previous findings[29], we found no support for multiple distinct genetic clusters within the other CLs (Fig. 1d, Extended Data Fig. 1g–k). However, Midas cichlids in most CLs exhibit phenotypic variation in putatively ecologically relevant traits such as dark/gold coloration[24,26], lip size[23], and pharyngeal jaw and body shape[22] (Extended Data Fig. 4j–n).

To assess the potential roles of these four major morphological axes (Fig. 1b) for speciation, we performed partial least squares (PLS) regressions between them and the primary axes of genomic divergence within each lake (Extended Data Fig. 4a–i). Of all analysed traits, lip size was the most important explanatory variable for genomic divergence in GL Nicaragua, GL Managua, and CL Masaya, whereas in the two multi-species CLs, body shape (CL Apoyo) and body shape together with pharyngeal jaws (CL Xiloá) were the most important predictors of genomic divergence among sympatric species. None of the other lake populations were structured and we therefore found no association between genomic divergence and phenotypic trait variation in these lakes. This lack of association also applies to the population in CL Apoyeque, which is characterized by pronounced bimodality in lip size (Extended Data Fig. 5p). Having identified the most important focal traits related to sympatric genomic divergence, we next tested whether and why divergence in some ecologically relevant traits might lead to speciation or not in some CLs. Therefore, we characterized the genetic architecture of these traits, and quantified genomic differentiation and signatures of divergent selection.

## Coloration is a trans-specific polymorphism

Midas cichlids owe their name to the presence of 'golden' morphs (named after the Greek myth of King Midas), which occur at low frequency (1.9–23.9%) in most lakes[22,26], but are exceedingly rare or even absent in three CLs (Extended Data Fig. 4o). Coloration has long been implicated in speciation in Midas cichlids[24,26], and assortative mating by colour morph[25,26] would tend to support this hypothesis. Golden Midas cichlids lose their melanic pigmentation during adolescence, and it has been proposed that this trait is monogenic[25,28]. Using genome-wide association (GWA) mapping in wild-caught samples (*n* = 273) in combination with pedigree-based mapping (Extended Data Fig. 5a, b), we identified a 230-kb region on chromosome (chr.) 11 that harbours the causal locus for this trait (Fig. 2a, Extended Data Fig. 5a–d). Several highly associated non-coding variants surrounding a serine/threonine-protein kinase (*stk*) gene are likely to constitute the molecular basis for the dark/gold phenotype (Extended Data Fig. 6a). Haplotype sharing across gold individuals from different lakes and species suggests that the same genetic basis underlies this trait in all populations and that causal alleles were introduced into the CLs from the source lakes (Extended Data Fig. 5c, d).

As a next step, we investigated whether this Mendelian trait plays a role in sympatric divergence. We did not detect substantial genomic differentiation associated with dark/gold coloration in populations where golden morphs are common (mean Hudson's fixation index ($F_{ST}$), 0–0.027; Fig. 3a). Thus, we conclude that the dark/gold coloration

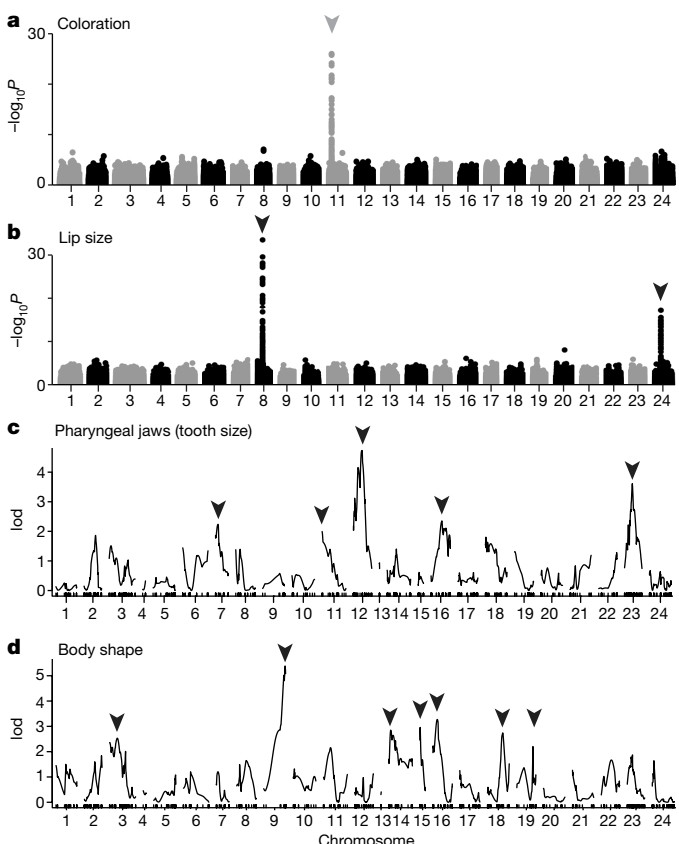

**Fig. 2 | Genotype–phenotype association mapping. a, b**, GWA mapping for dark/gold coloration (**a**) and lip size (**b**). For coloration there is one peak of high association (arrowhead) on chr. 11; for lip size we found two peaks, one on chr. 8 and one on chr. 24. **c, d**, QTL mapping for pharyngeal jaw tooth size (**c**) and body shape (geometric morphometrics PC1 scores; **d**). For pharyngeal jaws and body shape we detected five and seven QTLs (arrowheads) of small effect (2–7% of variation) that together explain 22.7 and 29.8% of the phenotypic variance, respectively. GWA analyses for pharyngeal jaws and body shape are shown in Extended Data Fig. 6d, e. lod, log odds ratio.

does not substantially contribute to genome-wide differentiation and, therefore, speciation. Despite the simple, Mendelian genetic basis and previous evidence for both ecological divergence[24] and assortative mating[25,26], the conspicuous dark/gold phenotype appears to constitute a polymorphism only.

## Genomic divergence due to lip size is subtle

Apart from dark/gold coloration, hypertrophied lips have also been suggested to be involved in speciation in Midas cichlids. Fish with extensive, bimodally distributed variation in lip size—here referred to as thin- and thick-lipped—are found in four lakes[23] (Extended Data Fig. 5p). Lip size causes a trade-off in feeding efficiency: thin-lipped fish are better at catching evasive prey, whereas thick-lipped fish are better at feeding from rocky crevices[23]. Moreover, thin- and thick-lipped fish show positive assortative mating in the laboratory and the field[23].

Consistent with previous evidence that lip size is an oligogenic trait[23], we found high genomic associations for lip size (*n* = 178) at only two loci (Fig. 2b). One peak of association on chr. 8 is present in all four lake populations with thick-lipped fish, whereas the second peak on chr. 24 is found only in GL Nicaragua (Extended Data Fig. 5e–h). The most highly associated variant on chr. 8 explains 77% of lip size variation in CL Apoyeque (where no evident confounding effects of population structure exist). The core haplotype is shared across

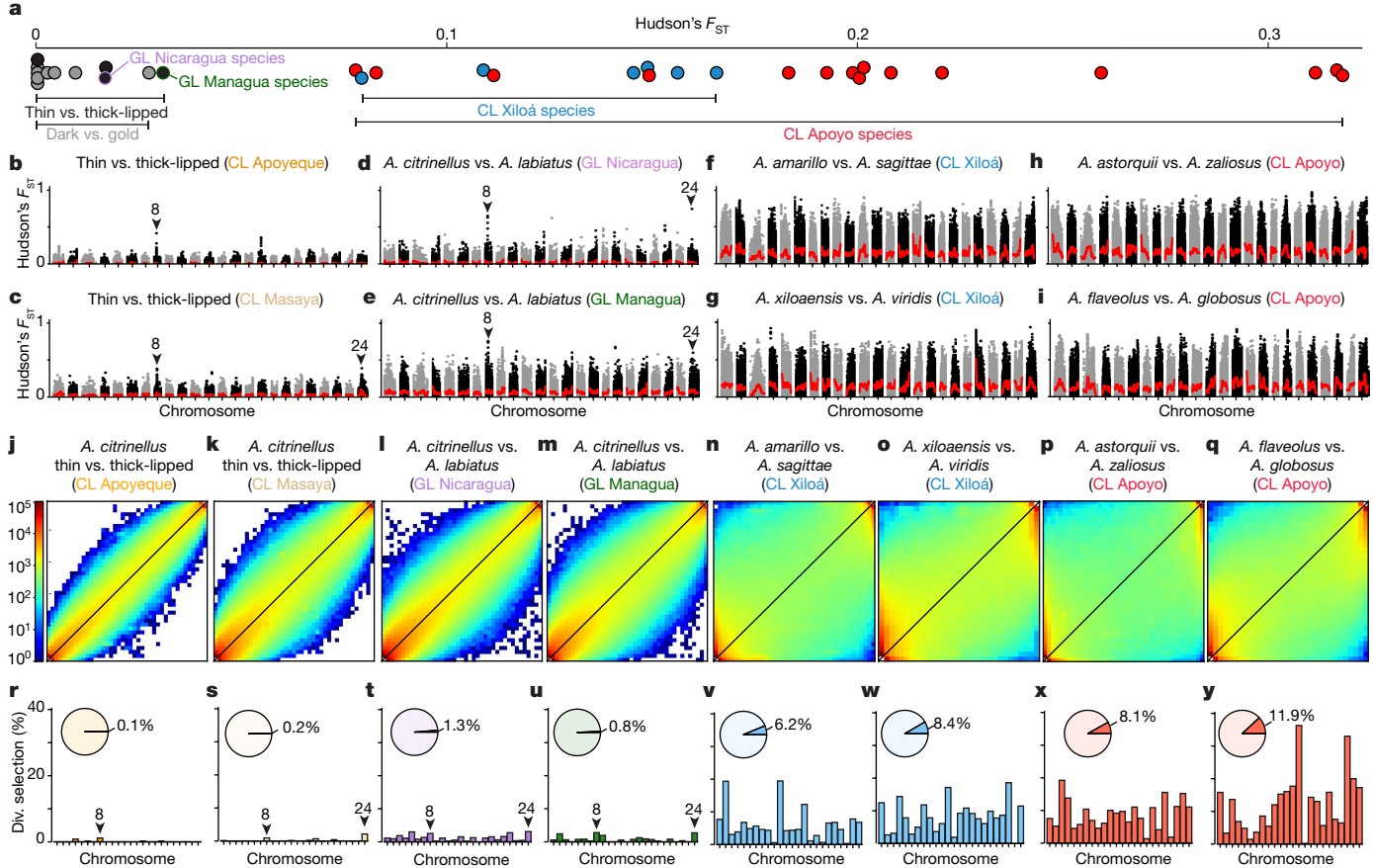

**Fig. 3 | Genomic differentiation across the species complex. a**, Genome-wide levels of genetic differentiation among sympatric populations differ greatly, ranging from low differentiation between dark and gold-coloured individuals ($F_{ST} = 0-0.027$), thin- and thick-lipped ecotypes in CLs Apoyeque and Masaya ($F_{ST} = 0-0.016$) or thin- and thick-lipped species in the great lakes ($F_{ST} = 0.016-0.031$) to substantial differentiation between young, sympatric species of CLs Xiloá ($F_{ST} = 0.08-0.17$) and Apoyo ($F_{ST} = 0.08-0.32$). **b**–**i**, Differentiation across the genome (dots, Hudson's $F_{ST}$; non-overlapping 10-kb windows; red line, loess smoothing) is overall low between thin- and thick-lipped populations and species (**b**–**e**), but has pronounced, shared peaks of high differentiation on chromosomes 8 and 24. By contrast, 3–10 times higher levels of genomic differentiation are found among species pairs from CLs Xiloá (**f**, **g**) and Apoyo (**h**, **i**). (see Extended Data Fig. 7 for additional comparisons). **j**–**q**,

Two-dimensional unfolded site frequency spectra, in which variants deviating from the diagonal indicate genetic differentiation, reveal a similar pattern. Differentiation is weak between thin- and thick-lipped morphs (**j**, **k**) and species (**l**, **m**), whereas it is pronounced and genome-wide among sympatric species in CLs Xiloá (**n**, **o**) and Apoyo (**p**, **q**). **r**–**y**, Genome-wide (pie charts) and chromosome-specific (bar plots) proportions of non-overlapping genomic windows under divergent selection within lakes (≥0.99 support). Only a few windows are detected in CL Apoyeque (**r**), CL Masaya (**s**) and GLs Nicaragua (**t**) and Managua (**u**), including the regions on chromosomes 8 and 24 that are associated with lip size (arrowheads). In pairwise comparisons of sympatric species from CLs Xiloá and Apoyo (**v**–**y**), there are 5–122 times more windows under divergent selection.

thick-lipped fish in all four lakes (Extended Data Fig. 5m–o), suggesting a shared genetic basis. Given the demographic histories of the populations (Fig. 1a), this haplotype was probably introduced from GL Nicaragua into GL Managua and from the GLs into CLs Masaya and Apoyeque. Moreover, this haplotype shows signatures of a selective sweep (Extended Data Fig. 5i–m), consistent with divergent selection acting on lip size[23]. In close vicinity to the highest-associated variants on chr. 8 are two inward rectifier potassium channel genes (*kcnj2* and *kcnj16*; Extended Data Fig. 6b), while the interval on chr. 24 harbours the G-protein-coupled receptor gene *ptger4* (Extended Data Fig. 6c).

As for the dark/gold polymorphism, genome-wide differentiation associated with lip ecotypes is low to absent in CLs (mean $F_{ST}$, 0.016 and approximately 0 for CL Masaya and CL Apoyeque, respectively; Fig. 3a). Even in the GLs, genome-wide differentiation is very low (mean $F_{ST}$, 0.016 and 0.031 for GL Nicaragua and GL Managua, respectively; Fig. 3a). Only a few islands of differentiation overlapping the regions identified by GWA mapping distinguish thin- and thick-lipped ecotypes or species, with a prominent shared peak of differentiation on chr. 8

(Fig. 3b–e). Given that the two morphs form distinct genetic clusters in the GLs and because both CLs that harbour thick-lipped fish were colonized less than 3,000 years ago (Fig. 1a, Extended Data Fig. 3), this suggests that prior population differentiation in the source population was homogenized after CL colonization. The subtle and seemingly unstable genome-wide differentiation between lip ecotypes implies that divergence along this phenotypic axis is inefficient or insufficient for sympatric speciation.

## Polygenic selection facilitates differentiation

Genome-wide differentiation in the species flocks of CLs Apoyo and Xiloá is 3–10 times higher (mean $F_{ST}$, 0.08–0.32, Fig. 3a) than the extremely low differentiation between dark and gold or thin- and thick-lipped fish, including the two GL species (mean $F_{ST}$, 0–0.03, Fig. 3a). These contrasting patterns of genomic divergence are also reflected in two-dimensional site frequency spectra (Fig. 3j–q). Among the sympatric CL species, genomic differentiation is heterogeneous and widely distributed (Fig. 3f–i, Extended Data Fig. 7). The sympatric

phenotypic diversification of species in CLs Apoyo and Xiloá (Fig. 1a) has occurred in parallel along similar axes, including traits that characterize limnetic–benthic niche divergence[19–22]. In this regard, body shape and pharyngeal jaws are among the most ecologically relevant traits separating the species[19,20].

The lack of any prominent GWA mapping signals for body shape and pharyngeal jaws ($n = 453$ and $n = 269$, respectively; Extended Data Fig. 6d, e) is consistent with a polygenic basis, as small-effect loci are unlikely to be detected without much larger sample sizes. Because of this limitation and caveats to association mapping across species, we performed quantitative trait locus (QTL) mapping to independently confirm the polygenic basis of both traits. We mapped both traits in an $F_2$ mapping panel derived from an intercross between the CL Apoyo species *A. astorquii* and *A. zaliosus*, which differ in both focal traits (Extended Data Fig. 4k–n), also in the laboratory[27]. The final QTL models identified five small-effect loci (2.2–5.6% of variance explained) for maximum pharyngeal tooth size and seven loci for body shape (2.1–6.7% of variance explained), accounting together for 22.7 and 29.8% of the phenotypic variance, respectively. In combination with GWA mapping and previous reports[27], our results indicate that pharyngeal jaw type and body shape variation have clearly different genetic architectures from dark/gold coloration or lip size.

New and previously published mate choice experiments in the laboratory with sympatric species from CLs Apoyo and Xiloá demonstrate that mating between benthic and limnetic ecomorphs within lakes is almost completely assortative (Extended Data Table 2, Supplementary Notes). Given the absence of distinct habitats in the laboratory, this shows that spatial separation (that is, habitat isolation) is not necessary for maintaining pre-zygotic reproductive isolation. Next, we tested whether these species would mate assortatively by ecomorph in phylogenetically controlled, between-lake mate choice assays[31], and found that mating between lakes was random (Extended Data Table 2). This suggests that independently evolved traits, which are characteristic of adaptation to particular trophic niches (for example, body shapes related to benthic or limnetic lifestyles[19]), are not likely to be important for mate choice.

The pronounced levels of genome-wide differentiation in the adaptive radiations of CLs Xiloá and Apoyo are unlikely to be due to assortative mating alone, as we found several genomically admixed hybrid individuals between sympatric species (Extended Data Fig. 1c, d). However, despite such interbreeding, genome-wide differentiation remains high (Extended Data Fig. 7a, n, p), suggesting that selection against hybrids is strong enough to prevent homogenization of the genomes of the parental species through backcrossing.

Given that genomic regions of extraordinary differentiation do not necessarily correspond to those under divergent selection or reduced gene flow[32], we expanded the genome-wide analyses by conducting machine learning-based screens to identify genomic windows that are affected by divergent selection between pairs of sympatric species. In CLs Apoyo and Xiloá, we found numerous, genome-wide distributed signatures of divergent selection (6.2–11.9% of the genome; Fig. 3v–y). Notably, we found 5–15 and 30–120 times fewer windows under divergent selection between the GL species (0.8–1.3%; Fig. 3t, u) and between thin- and thick-lipped morphs in CLs Masaya and Apoyeque (0.1–0.2%; Fig. 3r, s), respectively. Three windows centred around the lip locus on chr. 8 were classified to be under divergent selection in all four comparisons between thin- and thick-lipped populations and species (Extended Data Fig. 6f). Although many more genomic windows were under divergent selection in the comparisons for CLs Apoyo and Xiloá, none was shared across all of them (Extended Data Fig. 6g). In line with the substantial variation in morphological traits, including pharyngeal jaw and body morphology, genomic windows under divergent selection in species from CLs Apoyo and Xiloá were enriched in genes involved in developmental processes and anatomical structure development (Extended Data Fig. 6h, i).

To test whether differences in genetic drift could explain the higher levels of genomic differentiation among species in CLs Apoyo and Xiloá compared to the GL species, we performed simulations. Our results show that differences in drift alone cannot explain the observed differences in genomic differentiation (Extended Data Fig. 8). Moreover, estimates of effective gene flow show that reproductive isolation is stronger among species from CLs Apoyo and Xiloá than between the GL species (Extended Data Table 1). Combined with the population genomic results, these analyses strongly indicate that widespread and strong extrinsic post-zygotic barriers are not only likely to contribute to, but also might be necessary to achieve, genome-wide differentiation in the small adaptive radiations of CLs Apoyo and Xiloá.

## Discussion

We have compared population divergence along four major phenotypic axes in the Midas cichlid species complex using 453 re-sequenced genomes. These genomic analyses suggest that the genetic architectures of traits under divergent selection—in addition to ecological factors[29]—make important contributions to determining whether phenotypically variable populations will form new species in sympatry. We found that only one to two major effect loci underlie two of the most conspicuous phenotypic traits in Midas cichlids: the dark/gold coloration and hypertrophied lips. The simple genetic architecture of these traits, together with their well-characterized ecological function and strong effects on assortative mating, suggest coupling of pre- and post-zygotic isolation (mate choice and ecological performance). Such 'magic traits' or 'multiple effect traits'[4,7,8] are commonly thought to suffice in efficiently reducing gene flow, allowing the buildup of genome-wide differentiation that ultimately characterizes distinct species[6–8]. However, population divergence and genome-wide differentiation between dark/gold morphs and lip-associated ecotypes was either completely absent or very shallow and probably unstable (Fig. 3a). This is exemplified by the two older (16,700 years) and morphologically distinct GL species. These two species appear stalled at a stage of extremely shallow genomic divergence, and this divergence also seems to break down easily, as evidenced by the diminished population structure in CLs Masaya and Apoyeque—despite support for strong assortative mating in their current environment (Extended Data Table 2).

By contrast, the sympatric species of the radiations in CLs Apoyo and Xiloá exhibit pronounced genome-wide differentiation (Fig. 3), form distinct genetic clusters (Fig. 1d), and show strong assortative mating (Extended Data Table 2), supporting the notion that they are 'good' biological species. Unlike the case of the dark/gold polymorphism and lip-associated ecotypes, pre-zygotic reproductive barriers among species in CLs Apoyo and Xiloá are likely to be backed by strong extrinsic post-zygotic isolation on the basis of the many loci under divergent selection (Fig. 3v–y). Moreover, our mate choice experiments suggest that divergence in their ecologically relevant traits (at least the ones previously linked to divergence in Midas cichlids) is not coupled with assortative mating (Extended Data Table 2). Thus, even if demographic and/or environmental fluctuations lead to a temporary breakdown of assortative mating, divergent selection against hybrids might suffice to maintain species boundaries in the CL Apoyo and Xiloá flocks. Divergent selection affecting a large number of loci across the genome (Fig. 3v–y)—by acting, for example, on one or several traits with a polygenic basis or, alternatively, a combination of multiple traits each with a simpler genetic basis (multifarious selection)—seems therefore most effective at building up and maintaining genomic differentiation in sympatry. This is consistent with the notion that highly polygenic barriers are likely to underlie the maintenance of *Heliconius* butterfly species[33] and with simulations that have suggested that speciation with gene flow often requires selection on many unlinked genes[34].

In conclusion, we show that phenotypes with simple genetic architectures may not necessarily lead to population divergence and speciation in sympatry, even if these traits are important for both ecological performance and mate choice. By contrast, as evidenced by the extremely rapid multispecies outcomes of sympatric speciation in CLs Apoyo and Xiloá, polygenic selection might be more effective in driving the buildup of persistent allele combinations[18], probably until genomic 'tipping points'[5] are reached and speciation unfolds. We propose that this could be a more general and underappreciated feature of speciation with gene flow. By comprehensively investigating a single species complex, we provide empirical evidence that the genetic architectures of traits under divergent selection strongly affect whether genome-wide differentiation will progress along the speciation continuum, eventually leading to 'well-defined species,' as Darwin might have called them.

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

## Methods

### Reference genome assembly

To generate the first Midas cichlid chromosome-level genome assembly (PacBio long-reads in combination with scaffolding by BioNano optical maps and Hi-C chromosome conformation capture; genome size ~900 Mb), we obtained high-molecular-weight genomic DNA from liver tissue that was extracted from a single female *A. citrinellus* wild-caught individual from GL Nicaragua that was raised in the animal research facility at the University of Konstanz (permit number T-16/13, Regierungspräsidium Freiburg, Abteilung 3, Referat 35, Veterinärwesen & Lebensmittelüberwachung, Germany). Tissue was immediately shock-frozen in liquid nitrogen and stored at −80 °C. Tissue was lysed in high-salt lysis buffer (100 mM Tris HCl pH 7.0, 200 mM NaCl, 30 mM EDTA, pH 8.0, 0.4% SDS, 100 µg/µl Proteinase K) and extracted using a phenol-chloroform procedure. Ice-cold ethanol was added, and DNA was spooled on shepherd hooks and dissolved in Tris-EDTA, pH 8.0. Wide-bore pipetting tips were used to avoid damaging the genomic DNA. RNA was removed by Rnase A treatment. Glycogen traces in the spooled genomic DNA were precipitated by adding 0.3 volumes of 3 M Na-acetate pH 6.8 followed by centrifugation for 20 min at 13,000$g$ at room temperature. Pulse-field gel electrophoresis (PFGE, SAGE Pippinpulse) revealed DNA molecule lengths between 40 and 300 kb.

For the preparation of four Pacific Biosciences (PacBio) libraries, the DNA was purified using AMPure XP beads. Shearing of genomic DNA was performed using a Megaruptor device (Diagenode) (fragment sizes: 35kb). We prepared PacBio SMRTbell libraries according to the instructions of the SMRTbell Template Prep Kit 1.0, 'Procedure and Checklist – 20 kb Template Preparation Using BluePippin™ Size-Selection System.' Size selection of SMRT bell libraries was performed using the BluePippin™ system (Sage Science) (minimum fragment length cutoff between 12 kb (3 libraries) and 15 kb (1 library)). SMRT bell libraries were loaded to PacBio SEQUEL SMRT cells after primer annealing and polymerase binding using MagBeads. We successfully sequenced 19 SEQUEL SMRT cells using SEQUEL binding and polymerase v.2.0 and sequencing chemistry v.2.1 (movie length 10 h). The average N50 of subreads was 14.4 kb, resulting in a total of 80 Gb (~89 × coverage) of long-read sequencing data.

Long reads were assembled using the MARVEL assembler[35,36] (https://github.com/MartinPippel/DAmar) with default parameters unless mentioned otherwise. MARVEL consists of three major steps: the setup phase, the patch phase, and the assembly phase. In the setup phase, reads were filtered by choosing only the longest read of each ZMW and requiring subsequently a minimum read length of 11 kb. The resulting 2.5 million reads (~50× coverage) were stored in an internal database. The patch phase detects and corrects read artefacts, including previously missed adapters, polymerase strand jumps, chimaeric reads, and long low-quality read segments that are the primary impediments to long, contiguous assemblies. The local alignment computation is by far the most time- and storage-consuming part of the pipeline. Therefore, a repeat masking strategy was applied that differs from the default MARVEL pipeline, but that can be more easily applied to computing clusters. Low-complexity intervals, such as microsatellites or homopolymers, were annotated with Dbdust (https://github.com/thegenemyers/DAZZ_DB; commit: 0bd5e07) and tandem repeat elements were analysed using datander and TANmask (included in MARVEL developmental branch). Furthermore, local alignments of 1× coverage against 1× coverage of the genome were computed using daligner (https://github.com/thegenemyers/DALIGNER) and then alignment piles of size 10 and greater were used to generate repeat interval tracks. The resulting repeat tracks (dust, tan, repeat) were subsequently used to compute all local alignments between all blocks of the database. The patched reads (~47× coverage) were then used for the final assembly phase, beginning with determining all overlaps of patched reads. The previously created repeat annotation was reused and the trace spacing was set to 126 to force daligner to store the traces into a 16-bit buffer. This modification increases the storage demands on average by 20% but ensures the use of a modified version of Lastitch. This modified version stitches short alignment artefacts resulting from bad sequencing segments within overlapping read pairs to align through low complexity or tandem repeat elements without creating an overflow when using the default 8-bit compression of traces. This step was followed by a more precise repeat annotation and the generation of the overlap graph. The final contigs were generated by touring the overlap graph. Finally, to correct base errors, we first used the correction module of MARVEL, which makes use of the final overlap graph and corrects only reads that were used to build contigs. By using an alignment-based approach, the final contigs were further separated into a primary and an alternative contig set.

Contigs were first scaffolded using optical maps, generated at Rockefeller University. Purified DNA sequence-specific labelling was performed by the nick, labelling, repair, and staining steps according to the Saphyr preparation protocol. Sequence specificity was provided by the nickase Nt.BspQI and Nt.BssSI. Labelling was carried out by a nick translation process in the presence of a fluorophore-labelled nucleotide. The labelled nicks were repaired to restore strand integrity and DNA molecules were stained for visualization of the backbone visualization. These molecules were then imaged using the Saphyr system, which loads stained molecules automatically into Bionano Genomics nanochannel chips using electrophoresis. Label positions and lengths of DNA molecules were recorded by the on-board CCD camera using green and blue lasers in the Bionano Genomics Saphyr system. Data were generated from a total of one flow cell per nickase. A genome map was assembled de novo and used to order and orient the contigs from the MARVEL PacBio assembly, and to correct remaining contig misassemblies. Consensus physical maps (CMAP) were assembled using Bionano Solve v3.1 (https://bionanogenomics.com/support-page/bionano-solve/). Molecules were filtered for a minimum length of 150 kb, a minimum of nine labels on each molecule, and a backbone maximum intensity of 0.6 (Nt.BspQI: $n$ = 1,741,322; approximately 468× raw 1,010× coverage; Nt.BssSI: $n$ = 1,786,236; approximately 466× raw 929× coverage). A $P$ value threshold for the optical mapping assembly was set to at least $1 \times 10^{-10}$. For Nt.BspQI a total of 1,587 CMAPs (N50 of 1.152 Mb; total CMAP length of 1,435.559 Mb) and for Nt.BssSI a total of 1,069 CMAPs (N50 of 2.078 Mb; total CMAP length of 1,503.491 Mb) were generated. We used the two-enzyme workflow of Bionano Solve 3.1 hybrid-scaffolding pipeline, with default parameters. The process of hybrid scaffolding includes alignment of the PacBio assembly to the Bionano physical maps, identifying and resolving conflicting alignments, merging of nonconflicting assembly and CMAPs into hybrid scaffolds, and then a final translation back to FASTA format. For realigning the PacBio reads to the assembled scaffold we used PacBio's pbalign tool (https://github.com/PacificBiosciences/pbalign; commit: 0669a4e;), which internally uses blasr (5.3.2-a579bd5) to map PacBio raw reads back to the scaffolds.

To further order and orient scaffolds to chromosome scale, genome-wide chromatin interaction data (Hi-C reads) and the 3D de novo assembly pipeline 3d-dna (https://github.com/theaidenlab/3d-dna; commit: 5baf854)[37] were used. Hi-C reads were sequenced at PhaseGenomics (in total 513M Illumina paired-end 80 bp reads; ~46× coverage). As input for the 3d-dna pipeline, the Bionano hybrid scaffolds were used together with a duplicate-removed alignment file of the Hi-C read pairs, generated with Juicer v.1.7.6[38]. Juicer maps Hi-C read pairs to contigs/scaffolds, removes duplicates and near-duplicates and creates a list of valid Hi-C contacts. Subsequently the 3d-dna workflow was run with default parameters. Hi-C scaffolding resulted in 24 superscaffolds, corresponding to the expected chromosome number for Midas cichlids[39].

To reach an error rate of Q40, the final scaffolds were further polished. To do so, the whole data set of PacBio raw reads (89× coverage) were mapped to the scaffolds and the consensus sequence was called with PacBio's Arrow tool (https://github.com/PacificBiosciences/GenomicConsensus; commit: c92ef5d). To improve the performance,

especially in repetitive regions, Arrow polishing was applied twice consecutively. A gap-filling step was indirectly performed in the second polishing round, in the sense that Arrow creates a consensus sequence based on the alignment piles of the scaffolds and all PacBio raw reads. In cases where alignment piles spanned gap regions, Arrow closed them. To further correct remaining base errors, we used the variant detector freebayes v.1.1.0[40] to detect polymorphic positions and fix erroneous non-polymorphic sites in the reference sequence using samtools v.1.8/bcftools v.1.7 consensus[41]. The final assembly consisted of 8,683 contigs and the cumulative number of gaps was 30,847,507 bp. The contig N50 value was 3.84 Mb.

Finally, we renamed the polished superscaffolds to chromosomes according to maximum homology with the Nile tilapia assembly[42], that is, we aligned our assembly to Nile tilapia using LASTZ[43] and renamed our superscaffolds according to whichever chromosome contained the majority of their sequence (ignoring smaller translocations and the split chromosome 8 (a and b) in tilapia). Finally, we manually added the complete *A. citrinellus* mitogenome sequence[44] to our assembly.

### Genome annotation

We generated gene models with EvidenceModeller v.1.1.1[45] based on three lines of evidence: ab initio gene predictions, RNA sequencing (RNA-seq)-derived transcripts, and protein homology. Gene prediction was performed with Braker v.2.0.4[46], using a compilation of RNA-seq data (4,650,216,577 reads in total) sets comprising various developmental stages (1 day post hatch, 1 month, 3 months, adult) and tissues (whole body, eyes, lips, pharyngeal jaws, skin) of Midas cichlids for training[47–51]. Prior to that, we mapped reads to the newly assembled reference genome (see 'Reference genome assembly') with HISAT v.2.1.0[52]. The same set of RNA-seq reads were also assembled into transcripts with Stringtie v.1.3.3b[53] to serve as transcriptomic evidence. For this, individual binary alignment map (BAM) files were merged before assembly by tissue and RNA-seq data set and the resulting transcripts were finally merged using Stringtie to generate a single transcriptome-based evidence file. Homology-based gene evidence was determined by aligning the full set of proteins of the following seven species to the reference genome using exonerate[54]: Nile tilapia (*Oreochromis niloticus*), three-spined stickleback (*Gasterosteus aculeatus*), zebrafish (*Danio rerio*), spotted gar (*Lepisosteus oculatus*), chicken (*Gallus gallus*), mouse (*Mus musculus*), and human (*Homo sapiens*). Protein sequences were downloaded from ENSEMBL (release 91). Relative weights for ab initio, transcriptomic, and protein homology-based evidences in EvidenceModeller were set to 1:2:5. Finally, we used the PASA v.2.0.2[55] annotation pipeline to update the EvidenceModeller consensus predictions by adding untranslated region (UTR) annotations and models for alternatively spliced isoforms. First, a set of transcripts was generated by combining de novo and genome-guided assemblies generated by the program Trinity v.2.6.0[56] in two independent runs using as input the previously described RNA-seq alignments and default parameters. The combined set of Trinity transcripts were processed using the seqClean tool (https://sourceforge.net/projects/seqclean) in order to remove poly-A tails and other contaminant sequences. Second, gene structures were identified according to the HISAT2 mapping results using Cufflinks v.2.2.1[57]. Last, the original EvidenceModeller annotation, the Trinity generated transcripts and the Cufflinks gene structures were imported into the PASA pipeline. The annotation output from the first PASA run (that is, transcripts and gene structures) were then used as input for an additional PASA run to further refine the gene models and produce the final genome annotation.

In total, 78,420 genes were supported (including purely predicted genes). We aligned the translated protein sequences of these genes to the NCBI non-redundant protein database (downloaded 8 July 2018) using BLASTp (-evalue 1e-3 -outfmt 5 -show_gis -word_size 3 -num_alignments 20 -max_hsps 20). In order to produce a more stringent annotation, only genes with evidence from exonerate and/or Stringtie were retained (22,495 genes; set 1). Further, a less stringent annotation was produced by adding genes that also aligned to known proteins in the NCBI non-redundant database to the first set of genes (66,370 genes; set 2). The completeness of the two gene sets was assessed using gVolante v.1.2.1[58], using the orthologue search pipeline BUSCO v.2/v3[59] and 233 core vertebrate genes (CVGs)[60] as the reference gene set. The two sets of annotations captured 230 (98.71%) and 231 (99.14%) complete core genes, respectively. To add functional annotation to the predicted genes of our set 1, coding sequences were extracted from the *A. citrinellus* genome and translated into amino acid sequences using gffreads v.0.11.4[61]. Proteins were then aligned to the ENSEMBL *O. niloticus* (GCA_001858045.3) protein set using BLASTp with an $e$ value cutoff of $1 \times 10^{-6}$. Orthology was assigned to each *A. citrinellus* protein that aligned to an *O. niloticus* protein with the lowest $e$ value. The ENSEMBL BioMart tool was finally used to retrieve *O. niloticus* gene names, gene descriptions, and gene ontology (GO) terms that were used to functionally annotate the Midas cichlid genes.

### Whole-genome resequencing

Adult fish (standard length 14.54 ± 3.43 cm (mean ± s.d.)) were collected with gill nets or by harpooning in field expeditions of the Meyer lab to Nicaragua between 2003 and 2015 (permit numbers DGRNB-ACHL-0078, DGRNB-IC-006-2007, No. 026-11007/DGAP y DGPN/DB-27-2010, DGPN/DB/DAP-IC-0003-2012, DGPN/DB-02-2012, DGPN/DB-IC-004-2013, DGPN/DB-011-2014, DGPN/DB-IC-015-2015, Ministerio del Ambiente y los Recursos Naturales (MARENA), Nicaragua). The GL fish collections were augmented with fish purchased from local fishermen, mostly from the big fish market in Granada (Lake Nicaragua fish) and Mateares (Lake Managua fish). All specimens were photographed in a standardized manner from the lateral view on site. We further obtained tissue samples and photographs for eleven *A. globosus* and ten *A. supercilius* specimens from CL Apoyo (holotypes and paratypes) from the Zoologische Staatssammlung München, Germany. We aimed to sample at least 20 individuals per species, lake and/or ecomorph whenever possible. The total number of Midas cichlid samples in this study is 453: GL Nicaragua *A. citrinellus* ($n = 24$), GL Nicaragua *A. labiatus* ($n = 24$), GL Managua *A. citrinellus* ($n = 25$), GL Managua *A. labiatus* ($n = 25$), CL Apoyeque *A.* cf. *citrinellus* (thin- and thick-lipped; $n = 20$ and 20), CL Apoyo *A. astorquii* ($n = 23$), CL Apoyo *A. chancho* ($n = 16$), CL Apoyo *A. flaveolus* ($n = 16$), CL Apoyo *A. globosus* ($n = 25$), CL Apoyo *A. supercilius* ($n = 10$), CL Apoyo *A. zaliosus* ($n = 21$), CL Apoyo admixed individuals ($n = 9$), CL As. León *A.* cf. *citrinellus* ($n = 20$), CL As. Managua *A. tolteca* ($n = 20$), CL Xiloá *A. amarillo* ($n = 21$), CL Xiloá *A. sagittae* ($n = 27$), CL Xiloá *A. viridis* ($n = 24$), CL Xiloá *A. xiloaensis* ($n = 16$), CL Xiloá admixed individuals ($n = 14$), CL Masaya *A.* cf. *citrinellus* (thin- and thick-lipped; $n = 20$ and 20), CL Tiscapa *A.* cf. *citrinellus* ($n = 20$).

High-molecular-weight DNA was extracted from fin or muscle tissue from all 453 Midas cichlids and five *Archocentrus centrarchus* (an evolutionary outgroup) using commercial kits (QiaGen Dneasy Blood & Tissue kit), including an RNase A treatment step. DNA integrity was manually inspected on agarose gels and concentrations were determined on a QuBit fluorometer. Genomic libraries were prepared using Illumina TruSeq DNA Nano kits (Illumina) aiming for 350-bp insert sizes. Genomic libraries were paired-end sequenced ($2 \times 150$ bp) on a HiSeq 4000 or HiSeq X-Ten Illumina platform at the Beijing Genomics Institute (BGI, Hong Kong). Pooling four to five individuals per lane resulted in an average effective genome coverage (counting only reads with mapping quality ≥30, nucleotides with base quality ≥20, and no read duplicates) of 25.6× ± 6.3× per individual.

### Pre-processing, mapping, and variant and genotype calling

After demultiplexing, we converted raw reads to unmapped BAM files for long-term storage using Picard tools v.2.9.4 (https://broadinstitute. github.io/picard), adding read group information and marking adapter sequences in the process (using the FastqToSamMark and

IlluminaAdapters modules). Reads were then converted back into fastq format (SamToFastq) for mapping with BWA mem v.0.7.15[62] to the newly assembled *A. citrinellus* reference genome (see 'Reference genome assembly'). We used the default settings and marked shorter split hits as secondary alignments (-M option). Read group and adaptor content information was incorporated into the final BAM files using MergeBamAlignment. Finally, we marked PCR and optical duplicates with MarkDuplicates for exclusion from downstream analyses. Considering all samples together, we jointly called variants and individual genotypes with freebayes v.1.1.0[40] using default parameters and applying standard quality filters (mapping quality ≥30, base quality ≥20). Information on population assignment was provided in the form of a popfile to obtain sensible priors for freebayes' genotype-calling algorithm. Subsequently, hard filters were applied using the vcffilter script from the vcflib package (https://github.com/vcflib/vcflib) (-s -f "QUAL > 1 & QUAL / AO > 10 & SAF > 0 & SAR > 0 & RPR > 1 & RPL > 1") to remove low-quality variant sites. Variant representation was normalized using vt norm[63] and a custom python script was applied to decompose multi-nucleotide variants into single nucleotide variants. Individual genotype calls based on a read depth smaller than five were set to missing for all downstream analyses. In total, we called 7,560,356 single-nucleotide polymorphisms (SNPs) and 597,215 insertions/deletions (indels) across the 453 samples. Genomic data handling and filtering steps were performed using vcftools v.0.1.15[64] and plink v.1.90/v.2.00[65]. Unless otherwise noted, all analyses were performed with data from the 24 chromosomes only; smaller scaffolds showed signs of low quality including aberrantly high SNP density and heterozygosity and were therefore excluded from analyses.

## VCF polarization, masking, and functional annotation

We polarized the sites that were polymorphic in the Midas cichlid samples by assessing allele frequencies in an outgroup of five *A. centrarchus* individuals from GL Managua. We assigned an ancestral allele to each variant site if at least four out of the five outgroup individuals had a valid genotype and the outgroup was monomorphic for an allele shared with the Midas cichlid samples.

To minimize the impact of potentially misassembled regions on downstream analyses, we applied a conservative masking strategy to the *A. citrinellus* reference genome. We hard-masked the following sites in the assembly: i) sites with a sequencing coverage across all Midas cichlid samples more than four s.d. above the mean; ii) sites with a mappability score of less than 0.5 (mappability was calculated using the gem-mappability program v.1.315 of the GEM library[66], using a *k*-mer size of 150 bp and allowing for up to two mismatches); iii) sites within 5 bp of an InDel variant; iv) sites within annotations of repetitive regions (repCov2), gaps, low complexity regions, or tandem repeats produced by MARVEL; v) sites in non-overlapping 10-kb windows with an average root mean square of mapping quality less than 30. In total, we masked 37.99% of all sites in the reference genome.

## Read-aware statistical haplotype phasing

Haplotypes were inferred by statistical phasing with SHAPEIT2 v.2.r900[67], making use of phase-informative reads[68]. In a first step, we extracted phase-informative reads from individual BAM files with the extractPIRs program using default filters for mapping and base quality. We then performed read-aware phasing for each of the 24 chromosomes separately with 200 conditional states and a window size of 0.5 Mb, running the algorithm for 10 burn-in, 10 pruning, and 50 main iterations. The resulting output files were converted to variant call format (VCF) and concatenated to obtain a single VCF file of phased variants.

## Recombination rate estimation

We estimated population-scaled recombination rates ($\rho$, $4N_e r$) for each population/species in non-overlapping 50-kb windows using the machine-learning approach implemented in FastEPRR v.2.0[69]. To create the input files, we excluded all masked sites in the reference FASTA file (see 'VCF polarization, masking, and functional annotation') from the phased VCF files (see 'Read-aware statistical haplotype phasing'). Windows with less than 20% unmasked sites were omitted completely from the analysis. To estimate recombination rates for each remaining window, we ran 100 replicates in parallel, using default $\rho$ values for simulating the training sets, but extending the second set with $\rho$ values of 400.0, 500.0, 600.0, and 800.0 to account for high-recombination rate windows. To properly model the demographic history of the population of interest, we converted the corresponding MSMC estimates (see 'Demographic inference') into a demographic model string in Hudson's ms format. We then estimated $\rho$ for each window using the trained model. To convert $\rho$ into raw recombination rates, we calculated $\pi$ as an estimator of $4N_e \mu$ for the same windows (see 'Genome scans'). By dividing the estimated $\rho$ by $\pi$ for each 50-kb window, we obtained a local estimate of $r/\mu$. Finally, we estimated the recombination rate by multiplying this ratio with a local estimate of the mutation rate obtained from the mean sequence divergence between the Midas cichlid population of interest and the *A. centrarchus* outgroup samples (see 'VCF polarization, masking, and functional annotation'), assuming a genome-wide mean mutation rate of $3.5 \times 10^{-9}$ per site per generation (see 'Demographic inference').

## Species tree inference and gene tree–species tree discordance

To infer phylogenetic relationships, we extracted 5,574 loci with a length of 2,000 bp each and included five phased alleles for each ingroup species and two alleles for the outgroup *A. centrarchus* (98 alleles in total for 20 lineages) from our whole-genome data set. These loci were selected randomly from the reference genome, requiring a minimum distance of 20 kb between loci, a minimum distance of 5 kb to any annotated exon, and 2,000 unmasked sites within a physical distance of less than 3 kb. Gene trees were obtained with RAxML v8[70] using the rapid bootstrap analysis and search of best-scoring maximum likelihood tree (option a) under a GTR+G substitution model and including 100 bootstrap replicates. Subsequent species tree estimation was performed using ASTRAL III v5.6.1[71] based on all individual unrooted gene trees under the multi-species coalescent model. A total of 200 bootstrap trees were obtained and used to plot the density tree (Extended Data Fig. 1a) with the program DensiTree included in BEAST v.2.4.7[72].

We implemented the program PhyParts v.0.0.1[73] to calculate the level of gene tree–species tree discordance. The main advantage of this method is that we can obtain observed gene tree discordance among all nodes of the species tree, instead of traditional estimations of a single value of discordance for the topology (for example, Robinson-Foulds distance[74] or branch length score[75]). PhyParts calculates the number of bipartitions across gene trees in conflict with each node of a given species tree, using the calculations introduced by Salichos et al.[76]. Under this approach, each edge of a tree is deconstructed to obtain the set of all bipartitions. Then each gene tree is examined to detect whether a given bipartition is in concordance (or in conflict) with the species tree. Given a set of rooted trees, a bipartition *h* is in conflict with a species tree *s* if (i) the ingroup of *h* contains any of the ingroup of *s*; (ii) the ingroup of *h* contains any of the outgroup of *s*; and (iii) the ingroup of *s* contains any of the outgroup of *h*. We summarized this information as follows (per node): number of bipartitions in concordance with the main topology, number of bipartitions in concordance with a specific main alternative topology, and remaining number of bipartitions supporting other topologies. In addition, we applied a bootstrap filter where edges with low bootstrap values were ignored from the analysis. Specifically, bootstrap values lower than 50% were considered as polytomies. This prevents errors of inflating the level of concordance or conflict given a high uncertainty in gene tree estimation. The analysis was ran using the ASTRAL species tree and all 5,574 gene trees estimated by RAxML as described above.

## Ancestry proportion estimation

We used ChromoPainter v.2[77] in combination with GLOBETROTTER[78] admixture modelling to obtain estimates of ancestry proportions for each crater lake species. Using the phased VCF files (see 'Read-aware statistical haplotype phasing'), we generated input files for each chromosome including the 10 individuals with the highest genome-wide sequencing coverage from each of the 19 populations or species. We omitted all variable sites that were masked in the reference genome (see 'VCF polarization, masking, and functional annotation') or had more than 20% genotypes represented as missing (including those with less than 5× sequencing coverage). To account for recombination rate variation, we included a genetic map based on the window-wise estimates of recombination rates (see 'Recombination rate estimation'). We ran ChromoPainter in a two-step procedure, specifying the four great lake populations as donors with equal prior probabilities. First, we divided the 190 individuals into subsets of 10 individuals and estimated the average switch rate parameter and global mutation probability using 10 E-M (expectation-maximization) iterations in parallel for each subset and chromosome. In a second step, we averaged the resulting E-M estimates and ran ChromoPainter over all samples with fixed parameter values. For each chromosome, we ran GLOBETROTTER on the distribution of ancestry chunk lengths estimated by ChromoPainter to obtain chromosome-wise ancestry proportions for each crater lake population.

## Demographic inference

We used the multiple sequentially Markovian coalescent model implemented in MSMC v.2.1.2 (https://github.com/stschiff/msmc2)[79] to reconstruct changes in effective population size (within populations) and gene flow (between populations) through time. Analyses within populations were based on the 12 individuals with the highest mean sequencing coverage per population. For analyses between populations, we used the three individuals with the highest sequencing coverage per population to calculate the relative cross-coalescence rate (RCCR) as a proxy for gene flow. We only included sites that had at least 5× coverage in 80% of individuals for each analysed population. We ran MSMC2 with default settings for the number of iterations and the time segment pattern, but restricted haplotype pairs for the within-population analyses to within-individual pairs to eliminate potential effects of phasing errors. To convert the resulting scaled values into years, we assumed a mutation rate of $3.5 \times 10^{-9}$ per site per generation[80] and a generation time of 1.5 years.

In a complementary approach to the exploratory MSMC analysis, we explicitly compared models of crater lake colonizations and estimated model parameters by fitting simulated multidimensional site-frequency spectra to the empirical data using Fastsimcoal v.2.6[81]. To estimate the empirical multidimensional site-frequency spectrum as accurately as possible, we used the genotype-free likelihood method implemented in ANGSD v.0.929[82]. In a first step, we generated site allele frequency likelihood (SAF) files for each population using the BAM files of 10 individuals each with the highest mean sequencing coverage (see 'Pre-processing, mapping, and variant and genotype calling'), using the GATK[83] model to calculate genotype likelihoods. We required reads to be mapped in proper pairs and to have a minimum mapping quality of 30 and a minimum base quality of 20. We considered only sites with at least 80% of individuals having at least 5× sequencing coverage after filtering. Additionally, we omitted non-biallelic SNPs and SNPs with a significant strand bias or deviation from Hardy–Weinberg equilibrium ($P < 0.01$). The resulting SAF files across all unmasked sites in the reference genome were used to calculate whole-genome estimates of $\pi$ and $D_{XY}$. For demographic inference, to keep the influence of selection as small as possible, we excluded all sites within 1 kb of any annotated exon in the Midas cichlid reference genome (see 'Genome annotation'). We polarized the SAF files by providing an ancestral reference genome (see 'VCF polarization, masking, and functional annotation') and estimated the unfolded multidimensional site-frequency spectra for the populations of interest by optimizing the corresponding SAF files.

We fitted the demographic models to the observed unfolded multidimensional site-frequency spectra by running 100 independent Fastsimcoal runs from different parameter starting values. We optimized parameters for 100 ECM cycles, estimating the expected site-frequency using 200,000 coalescent simulations and assuming an infinite site mutation model. To estimate confidence intervals around the maximum likelihood parameter estimates, we applied a parametric bootstrapping approach. Using the optimization output with the highest likelihood, we simulated 100 times an unfolded multidimensional site-frequency spectrum with Fastsimcoal. To match the actual observed data as closely as possible, we simulated 24 independent chromosomes following the length distribution of the Midas cichlid reference genome and keeping the total number of simulated sites identical to the observed spectra. For the simulations, we assumed a mutation rate of $3.5 \times 10^{-9}$ per site per generation and a recombination rate of $1.05 \times 10^{-8}$ per site per generation. We optimized parameters for each simulated site-frequency spectrum 10 times, using 40 ECM cycles and 200,000 coalescent simulations. The run with the highest likelihood from each of the 100 bootstrap replicates was then used to generate 95% confidence intervals around the maximum likelihood parameter estimate.

## Estimation of population structure and overall genetic differentiation

To assess and effectively visualize population structure in our genomic data set, we used t-SNE[84]. Instead of applying t-SNE to previously calculated principal component scores, we applied it directly to our genotype data[85], and used the default values for hyperparameters (including perplexity). However, independently, we also used principal component analyses (PCAs) implemented in EIGENSOFT v.7.2.1[86] to assess population structure, using its least squares regression option (lsqproject) to account for missing data. In addition, we used the model-based approach of Admixture v.1.3.0[87] to exclude individuals with more than 25% admixture (Fig. 1d, Extended Data Fig. 1).

Using the ratio of averages approach[88], overall pairwise genetic differentiation in terms of Hudson's $F_{ST}$ was estimated with EIGENSOFT, setting the 'fsthiprecision' flag. The statistical significance of $F_{ST}$ (against the null hypothesis of $F_{ST} = 0$) was assessed using the implemented block jackknife approach by activating the 'fstz' option.

## GWA mapping

We used the mixed model approach implemented in EMMAX[89] to account for population structure in genotype–phenotype association mapping. More specifically, for each pooled or lake-specific analysis we incorporated a Balding–Nichols (BN) kinship matrix, calculated using a function included in EMMAX. Moreover, markers that deviated from Hardy–Weinberg equilibrium ($P < 0.01$) within a single species or morph included in an analysis, with more than 20% missing data, or with a minor allele frequency (MAF) of less than 0.05 were excluded. The proportion of phenotypic variance explained by a marker was derived from phenotype–genotype regressions. We note that for binary traits, association mapping based on an allelic model and allele frequency differences between groups are conceptually very similar. For these analyses we did not use the masked version of the genome (see 'VCF polarization, masking, and functional annotation').

GWA mapping for lip size and dark/gold coloration was conducted in populations, species or lakes that were polymorphic for the respective trait. For lip size, the respective data set comprised a total of 178 individuals (out of which 89 were thick-lipped) from GL Nicaragua ($n = 48$; 24 thick-lipped), GL Managua ($n = 50$; 25 thick-lipped), CL Masaya ($n = 40$; 20 thick-lipped) and CL Apoyeque ($n = 40$; 20 thick-lipped). For dark/gold coloration, the data set comprised 273 samples (out of which

88 were gold) from GL Nicaragua ($n = 48$; 24 gold), GL Managua ($n = 50$; 14 gold), CL Masaya ($n = 40$; 20 gold), CL Xiloá (*A. sagittae* and *A. xiloaensis*; $n = 55$; 16 gold), CL As. León ($n = 20$; 3 gold), CL As. Managua ($n = 20$; 5 gold) and CL Apoyeque ($n = 40$; 6 gold). For body shape (that is, the first principal component of a global geometric morphometric analysis), we used the entire data set ($n = 453$) and for the pharyngeal jaw trait all individuals for which we had measurements ($n = 268$), namely samples from GL Nicaragua ($n = 37$), GL Managua ($n = 47$), CL Apoyo ($n = 66$), CL Xiloá ($n = 60$), CL Masaya ($n = 26$) and CL Apoyeque ($n = 32$).

## Genome scans

For Manhattan plots of genome-wide differentiation, we calculated Hudson's $F_{ST}$ statistic[90] as a ratio of averages in non-overlapping 10-kb windows along the 24 chromosomes of the masked Midas cichlid reference genome. We considered only sites with at least 5× coverage in 80% of individuals for each analysed population, requiring at least 2,000 valid sites for a window to be included in the analysis. For plotting along chromosomes, we performed a smoothing of $F_{ST}$-values across adjacent windows using the 'loess' function in R (span = 0.01, degree = 1, family = 'gaussian').

## Genome-scale coalescent simulations

An alternative explanation for the substantially higher levels of genomic differentiation among species in CL Apoyo and Xiloá compared to the great lake species would be a stronger role of genetic drift in the relatively small crater lake populations. To test this hypothesis, we used the most-supported models from the Fastsimcoal analyses (see 'Demographic inference') to simulate genome-scale data using Fastsimcoal v.2.6. This allowed us to manipulate certain parameters to evaluate their impact on the genome-wide distribution of $F_{ST}$ values, particularly the relative influence of intra-lake migration rates (that is, effective gene flow, a proxy for the strength of reproductive isolation) versus genetic drift mediated by founder and current population sizes. We simulated 100 complete genomes comprising the 24 chromosomes in the Midas cichlid reference genome, using sample sizes matching our empirical data. We assumed a genome-wide mutation rate of $3.5 \times 10^{-9}$ per site per generation and a recombination rate of $1.05 \times 10^{-8}$ per site per generation (see 'Demographic inference'). The resulting genotype files were then processed exactly like the empirical data to calculate pairwise Hudson's $F_{ST}$ in non-overlapping 10-kb windows (see 'Genome scans'), using the masked Midas cichlid reference genome to filter sites and considering only windows with at least 2,000 valid sites. The distributions of $F_{ST}$-values based on the 100 simulations of entire genomes were very consistent and showed a close fit to those in our empirical data (Extended Data Fig. 8). To test whether genetic drift alone could explain the difference in the extent of differentiation between crater lake and great lake species in the empirical data, we repeated the simulations, but exchanged the past and present effective population sizes of the two respective crater–source lake pairs (that is, GL Nicaragua–CL Apoyo and GL Managua–CL Xiloá).

## Haplotype networks

To investigate whether shared genetic bases underlie lip size and the dark/gold coloration across populations or species, we used a self-organizing map-based approach implemented in the program Saguaro r44[91]. We generated a multi-FASTA file with aligned haplotypes derived from the phased VCF file for each of the 24 chromosomes. Sites masked in the reference genome or with less than 80% of individuals having a sequencing coverage of at least 5× were coded as missing. We converted the multi-FASTA files to binary feature files with the Fasta2HMMFeature program included in the Saguaro package and ran Saguaro with default settings. Haplotype networks were based on the non-recombining segments inferred by Saguaro using the haploNet function in the R package pegas v.0.11[92].

## Inference of local gene trees

In an effort to infer local gene trees around the highest-associated SNPs for the lip trait, we used Relate v.1.0.16[93]. We generated input files for each of the 24 chromosomes using the phased genotype data (see 'Read-aware statistical haplotype phasing') from all populations/species where thick-lipped fish occur (GLs Managua and Nicaragua, CLs Apoyeque and Masaya). We kept only sites that were not masked in the reference genome (see VCF polarization, masking, and functional annotation) and where all populations had less than 20% missing data, considering all genotypes with less than 5× sequencing coverage as missing. Distances between variable sites in the input haplotype files were adjusted to account for missing non-variable sites. As for ChromoPainter, we provided a genetic map based on window-wise estimates of recombination rates (see Recombination rate estimation). After the initial chromosome-wise runs to infer local genealogies, we estimated the population size trajectories for the populations of interest across all 24 chromosomes, running the algorithm for 10 iterations. Using the updated branch lengths after optimizing population sizes, we obtained local genealogies for each position of interest, assuming a generation time of 1.5 years (see 'Demographic inference').

## Detection of signatures of selection

To detect signatures of divergent selection between pairs of populations, we modified the supervised machine learning approach implemented in diploS/HIC[94]. This method uses simulated data of single populations to train a convolutional neural network (CNN) for classification of genomic windows into different selection categories. This model is then used to predict selection categories in genomic windows in the empirical data. As we were specifically interested in detecting genomic windows under divergent selection between pairs of sympatric species, we applied a pairwise strategy in place of the single-population approach of diploS/HIC. To simulate the training and test data sets under realistic demographic histories, we used the detailed demographic models inferred above (see 'Demographic inference') to parameterize simulation runs with MSMS v.1.3[95]. We simulated genomic data with a locus size of 1.05 Mb and simulations were either neutral or involved a site under shared or divergent selection between the two focal populations. The onset of selection was assumed to directly follow lake colonization or the sympatric species split for shared and divergent selection, respectively, and to proceed from either de novo mutations or standing genetic variation. The following simulation parameters were drawn randomly from probability distributions: mutation rate over locus per generation [uniform(0.000668, 0.00668)], recombination rate over locus per generation [exponential(mean: 0.018375, max: 0.055125)], relative position of selected site within the locus [uniform(0.4, 0.6)], selection coefficient [uniform(0.01, 0.1)], and initial frequency of the selected allele [loguniform($10^{-5}$,$10^{-2}$)]. We generated a total of 4,000 training data sets and 2,000 test data set for each category and population pair.

We used libsequence v.1.9.8[96] in a custom C++ program to generate feature vectors from the simulated training and test data sets. In a first step, we masked sites in the simulated data to mirror patterns of missing data in the empirical data. For this, we first generated a FASTA mask using the same criteria as for filtering the empirical data (see below). For each simulated data set, we drew a random locus of 1.05 Mb from the mask to filter out sites in the simulated data. We then divided the simulated 1.05-Mb regions into 21 subwindows of 50 kb each and calculated a set of within-population ($\pi$, Tajima's $D$, Fay and Wu's $H$, $H_1$, $H_{12}$, $H_2/H_1$ (ref. [97]) and 1-HAF[98]) and between-population ($F_{ST}$, $D_{XY}$, $G_{min}$ (ref. [99]), and SS-$H_{12}$ (ref. [100])) summary statistics for each subwindow. All summary statistics were normalized by dividing each statistic by its sum across all subwindows. Using the normalized feature vectors, we trained a CNN by closely following the 'train' mode of diploS/HIC for a maximum of 100 training epochs. For prediction, we calculated

feature vectors from the phased genotype data (see 'Read-aware statistical haplotype phasing') in the same manner as for the simulated data, excluding sites masked in the reference genome (see 'VCF polarization, masking, and functional annotation') or with less than 80% of individuals having a sequencing coverage of at least 5×. Only genomic windows with at least 20% unmasked sites were kept for the prediction of selection categories using the trained CNN. For quantitative comparisons (Fig. 3r–y) we retained only genomic windows that were present in all pairwise comparisons.

To more comprehensively compare and describe windows that were classified to be under divergent selection (>99% support), we screened for overlap in windows across multiple comparisons using the intersect function and plotted the results as Venn diagrams using the VennDiagram package[101]. To perform gene ontology (GO) term analyses, we selected all genes in our annotation that overlapped windows with >99% support to be under divergent selection in any of the four comparisons of CL Apoyo and Xiloa species using the BEDTools v2.29.2[102] intersect function. To identify enriched biological processes, we used the ShinyGO v.0.61 pipeline[103] using the Nile tilapia (*O. niloticus*) genome as a reference and standard settings (0.05 FDR *P* value threshold).

In a complementary, more targeted approach, we screened for signatures of selection within the shared region underlying lip size variation on chr. 8 based on unusually high or low (depending on which population is reference) cross-population extended haplotype homozygosity (XP-EHH) using REHH v.2.0.2[104] with the default settings.

### Site frequency spectra

We generated unfolded two-dimensional site-frequency spectra for all pairs of populations using the plotting functions in dadi v.1.7.0[105]. We determined the ancestral and derived alleles for each variant site as described above (see 'VCF polarization, masking, and functional annotation'), considering only biallelic SNPs. We down-sampled each variant position by randomly selecting 40 alleles (for the comparisons *A. xiloaensis* versus *A. viridis* and *A. flaveolus* versus *A. globosus*, 30 alleles) per population without replacement, excluding sites with an insufficient number of valid genotype calls.

### Phenotypic measurements and geometric morphometrics

Lip size was measured as previously described[106]. In brief, lip area was determined using Fiji (ImageJ) v.2.0.0 from standardized photographs of the lateral views of individual fish. Lip size was standardized by dividing it by body area (area contained by landmarks 3, 7, 11 and 15; Extended Data Fig. 4r). The standardized values were then $\log_{10}$-transformed.

For pharyngeal jaw measurements (maximum tooth size on the lower pharyngeal jaw), we dissected the fifth ceratobranchial, or lower pharyngeal jaw, from the fish. These bony elements were cleaned of all muscle and fascia and allowed to dry. Then, we measured the mass of each pharyngeal jaw to the nearest 0.001 mg using a digital scale. Subsequently, we took a digital image of the dorsal surface of the jaws with a size standard and imported it into Fiji (ImageJ). The areas of six teeth were then measured digitally. We measured the three most posterior teeth along each side of the midline of the pharyngeal jaw both because these teeth were likely to be homologous and they tend to be the largest teeth on the pharyngeal jaw. Then, we determined the maximum size tooth from these six measurements. To size-standardize these two pharyngeal jaw measurements, the cube root of the pharyngeal jaw mass and square root of the maximum tooth area were taken. Then, these two measurements were individually regressed against standard length (SL)—measured with calipers as the distance from the upper jaw tip to the caudal peduncle—and the residuals of each measurement were obtained for comparisons among Midas cichlid populations.

The gold phenotype is clearly visible on photographs[28] (Fig. 1d) and it is essentially impossible to mis-identify golden individuals, as they are red to yellow or white, sometimes with some residual melanin in

the skin or fin tissue. Phenotypic scoring was performed by several of the authors (A.F.K., G.M.S., and C.F.K.) without any mismatch in phenotypic assignments.

For geometric morphometric analyses, a total of 761 photographs were included, corresponding to the 453 individuals used for genomic analyses as well as 308 individuals from an $F_2$ mapping panel of an intercross between *A. zaliosus* und *A. astorquii* (see 'QTL mapping'). The configurations of points used in morphometric analyses of body shape (Extended Data Fig. 4r) comprised twelve fixed landmarks and six semi-landmarks. Points were digitized on body photographs using tpsDig v.2.32[107]. All further analyses were performed with the geomorph v3.0.6 R package[108]. Landmarks were aligned using a full Procrustes superimposition with the function gpagen. Allometry was accounted for by regressing shape variables on body standard length (taken from LM1 to LM8; Extended Data Fig. 4r) and using regression residuals in subsequent analyses. PCAs were performed using the prcomp R function. $F_2$ mapping panel individuals were excluded from this step (see 'QTL mapping'). Procrustes distances between groups of interest were performed using the procD.lm function. Consensus shapes were obtained using the mshape function, and plots of deviation were constructed with the plotRefToTarget function. Discriminant function analyses for comparison of different groups of interest were performed using the lda function of the MASS v7.3 library[109].

### Frequency estimation of golden morph

Frequencies of golden individuals in the great lakes were estimated from photographs of Midas cichlid catches on Granada Market (GL Nicaragua) and local fishermen (GL Managua) in 2018 and 2020. For CL As. León, frequencies were estimated from fish caught with gill nets.

### PLS regression of genetic divergence with phenotypic traits

To investigate which of the focal phenotypic traits (dark/gold coloration, lips, pharyngeal jaws, and body shape) are most correlated with the primary axes of sympatric divergence in Midas cichlids, we performed PLS regression analyses in the plsdepot v.0.1.17 R package[110]. As independent variables, we used the same trait measurements that were used in GWA and QTL mapping. More specifically, we coded dark/gold coloration as a binary trait and used $\log_{10}$-transformed, normalized lip size as a measure for lips. As a measure of pharyngeal jaw diversity, we used the residual, maximum pharyngeal tooth size after correcting for size (standard length). As body shape is a multivariate trait, we used the scores of the first three axes of a geometric morphometric PCA (see 'Phenotypic measurements and geometric morphometrics'). As dependent variables, we attempted to capture the primary axes of genetic divergence. In the case of lakes with only one or two described species, we reasoned that this corresponds to the first axes of lake-wise, genetic PCAs (see 'Estimation of population structure and overall genetic differentiation'). In the case of lakes with multiple species (CLs Apoyo and Xiloá), the PC axes do not necessarily maximally separate one of the species from all other sympatric species and do therefore not exactly correspond to the primary axes of divergence. Thus, in the case of these two lakes, we performed separate between-group PCAs for each species versus all other sympatric species grouped together and then projected (rotated) the original, individual, genetic PC scores onto the first axes of the between-group PCAs. Thereby, we obtained measures corresponding to the primary axes of genetic divergence for each species.

Excluding samples with missing data and dark/gold coloration as an independent variable for CLs Apoyo and As. León (only one and three golden samples in dataset, respectively), we first used the cross-validation technique implemented in the plsreg2 algorithm of plsdepot to determine the optimal number of latent variables to use. In a second step, we then performed the actual PLS regression step, using plsreg1 for lakes with only one dependent variable (all except CLs Apoyo and Xiloá) and plsreg2 for lakes with multiple species (CLs Apoyo and Xiloá).

The variable importance in projection (VIP) scores for each phenotypic trait over all used latent variables was then calculated[111]. Ninety-five per cent confidence intervals were obtained from the 0.025 and 0.975 quantiles of 1,000 non-parametric bootstrap replicates (that is, resampling with replacement).

## Mapping panel dark/gold phenotype

The $F_2$ mapping panel resulted from a cross of a homozygote gold *A. citrinellus* from CL Masaya and a homozygote normal *A. citrinellus* from GL Nicaragua. A total of 41 normal and 124 gold $F_2$s were obtained. To analyse the phenotype–genotype relationship, we designed microsatellite primers to genotype fish at the target locus (identified by GWA mapping): CKm04_HEX_fwd: TCTCGGCGCTTCCTTCATTT; CKm04_rev: TCTCGGCGCTTCCTTCATTT; CAAGGCCACGTGAAGTTTGG; alleles: 445/461. Fine-mapping was performed on three recombinant dark individuals using additional microsatellites and one nucleotide polymorphism (CKm21_FAM_FWD: GGGAAAGTGGATGCTCAGCA; CKm21_REV: ACCTCGGAGTGGATCATCCA; alleles: 366/394, CKm20_HEX_FWD: GCTCCAGCAGTTCCCTTCTT; CKm20_REV: TGCACACCCACACATTTGTT; alleles: 195/199, CKm19_FAM_FWD: CAGCCAACCCAGTGACATCA; CKm19_REV: CTTCTGGCCTTTGCGGTCTA; alleles: 360/385, CKm22_FAM_FWD: TGTTGCTGTATCCAAGCGGT; CKm22_REV: CCGTGCAATTGTGAGCTGTC; alleles: 286/311, CKm25_HEX_FWD: TTAAGTTGAGCTGAGCACAGT; CKm25_REV: CCGGACAACCTGCTAAACCT; alleles: 363/367, CKm26_FAM_FWD: CCATAGCCCATTAAGAAAAAGCCA; CKm26_REV: GGGTGAGGAGAACGGTAAGC; alleles: 360/400, CKm28_FAM_FWD: TCCTTCTGTCTCCTCACCCA; CKm28_REV: GGAAAGAAAGAACGAAGGAAGCT; alleles: 432/437, CKm09_FAM_FWD: TGCAAACAGTAACTACTCAGCA; CKm09_REV: ACCCACAGTGTGACTTTGCA; alleles: 430/434, CKm07_FAM_FWD: AAGACACACTGGCATGCTGG; CKm07_REV: GAGGGGCAGTTCAGACAGTC; alleles: 325/345, CKm05_FAM_FWD: GCCAACAGTTGTGCCAAACA; CKm05_REV: CCCCTCCACTGAGCATGTTT; alleles: 272/300, CKm03_FAM_FWD: GCTCTTGGACGGGGAGAAAA; CKm03_REV: GCCAACACTCTGCAGTACCT; alleles: 350/352, CKm02_HEX_FWD: GCCACACAGACGTCTGAGAA; CKm02_REV: GTGACTGAGCCTTTGCAAGC; alleles: 304/308, CKs7109560_FWD: TCGCCAAGTGCTTGCTCATA; CKs71_REV: GCTGCAACAGGAGAGTGAGT; alleles: A/G, CKm01_FAM_FWD: TGTGCCTTGCAGCAAAAGAC; CKm01_REV: CTCACTTGAGGTCGGTGCTT; alleles: 259/271, CKm11_FAM_FWD: ACAGCCATCCACCATTAGCA; CKm11_REV: GCACAACTTCAAGCAGGTCG; alleles: 204/214, CKm08_HEX_FWD: TGACAGCAACAACAGGCAGT; CKm08_REV: TGTCAGCACTTTTCCCTGCA; alleles: 370/374, FHmc85_HEX_FWD: TGACAGTGATGTGTTTCTTTGCT; FHmc8_REV: TTCCTCAAGGGCTTCACAGT; alleles: 204/214, FHm08_HEX_FWD: CCCTGCCTCAGGTAACACTC; FHm08_REV: ACCAGGCTCGATGTTTCAGT; alleles: 322/332.

## QTL mapping

To map pharyngeal jaw and body morphology, we used $F_2$ individuals derived from an intercross between *A. astorquii* and *A. zaliosus* (CL Apoyo). Further details on the cross have been provided in previous studies[27,112]. For linkage map construction, we obtained markers using double digest restriction-site associated DNA (ddRADseq) together with whole-genome resequencing of the parents of the cross. Raw reads were quality-controlled using Trimmomatic v.0.36[113] and aligned to the *A. citrinellus* reference genome using bwa-mem v.0.7.15[62] with default settings. We used the MarkDuplicates module of PicardTools v.1.141 (http://broadinstitute.github.io/picard) to remove duplicates. Genotypes were called using Freebayes v.1.3.1[40]. For further analysis, only those markers were retained for which parentals were homozygous for different alleles or one of the parentals was heterozygous, and for which $F_2$ genotype frequencies followed the expected segregation ratio ($\chi^2$ test; $P = 0.001$ threshold). We included only individuals with at least 90% called genotypes and loci present in at least 80% of the individuals. The linkage map was constructed with a regression-based algorithm implementing the Kosambi mapping function in JoinMap v.4.0[114]. The final linkage map consisted of 279 individuals and 594 markers with 2.6% missing data.

Both focal traits (maximum tooth size as a proxy for pharyngeal jaw morphology and body shape using geometric morphometric analyses) were measured the same way as the wild-caught samples (see 'Phenotypic measurements and geometric morphometrics'). Owing to the loss of transponder tags and quality of jaw preparations we did not obtain measurements for all individuals (the final data after removal of individuals with incomplete marker information was $n = 279$ individuals for body shape and $n = 246$ for pharyngeal jaw morphology). To be able to map the naturally occurring body shape variation in our dataset within the lab-raised mapping panel, we projected the QTL samples onto the PCA space of the 453 wild-caught samples and extracted their first PC scores. QTL mapping was performed in r/qtl v.1.46-2[115]. Candidate QTL loci were identified using the scanone function with Haley-Knott regression[116] and added to a multiple-QTL model using the fitqtl function in r/qtl[115]. Models were evaluated using multiple regression $F$-tests ($P > 0.05$) and model selection was performed by dropping non-significant QTL.

## Mate choice experiment

We performed mate choice experiments to test (i) whether the limnetic and benthic species would mate assortatively under laboratory conditions (that is, in the absence of different habitats), and (ii) whether fish would mate assortatively by ecomorph when exposed to fish from different lakes. These two experiments were performed in a very large arena ($340 \times 160 \times 80$ cm) under controlled temperature ($28 \pm 1$ °C) and light conditions (artificial fluorescent light in a 12:12 h on–off daily cycle; see Supplementary Notes). The first experiment, testing intra-lacustrine mate choice, comprised two trials corresponding to the two crater lakes that harbour sympatric species, CL Apoyo and CL Xiloá. In the first trial, 114 adult fish (>2 years old) were available. A total of 48 of them were tested, comprising the limnetic species *A. zaliosus* ($n_{females} = 25$ available (13 tested), $n_{males} = 13$ (12)) and the benthic species *A. astorquii* ($n_{females} = 30$ (10), $n_{males} = 46$ (13)) from CL Apoyo. In the second trial, 115 (33 tested) adult fish were available, comprising the limnetic species *A. sagittae* ($n_{females} = 26$ (6), $n_{males} = 17$ (6)) and the benthic species *A. amarillo* ($n_{females} = 12$ (11), $n_{males} = 60$ (10)) from CL Xiloá. The trials were terminated when no more naive fish of a certain sex and species were available for replacement (indicated as the final number of tested fish above). For both intra-lacustrine trials, the following design was used: initially, five fish of each sex of both the limnetic species and the benthic species (within crater lake, total of 20 fish at any given time) were placed in the experimental tank and allowed to mate freely. After pair formation and spawning (that is, only after laying of eggs) fish were removed from the experimental tank and the species identity of each member of the pair was recorded. To keep the relative frequencies of fish constant over the course of the experiment, spawning pairs were replaced with naive fish of the same sex and species not previously used in the experiment. We counted matings between conspecifics as assortative and between heterospecifics as disassortative. There was no evidence that the timing at which fish were introduced into the testing arenas affected their mating probability, as some of the fish formed pairs and were consequently removed from the experimental tank within four days of their original introduction. Indeed, based on our experience with Midas cichlids, handling has little effect on subsequent mating probabilities. In the case of fish from CL Apoyo, the trial was terminated 19 days after fish were originally introduced into the arena with a total of 19 pairs being formed (no more *A. zaliosus* males were available to replace mated ones). For CL Xiloá, it was terminated after 15 days with 12 pairs being formed (no more females of *A. amarillo* were available to replace those that formed pairs).

In a second experiment, we aimed to test whether fish would mate assortatively by ecomorph even when exposed to fish from a different lake (inter-lacustrine experiment). To test this hypothesis, we exposed

five females of each of the limnetic and benthic species from one of the two crater lakes harbouring small adaptive radiations (that is, CLs Apoyo and Xiloá) to five males of each of the limnetic and benthic species from the other lake. Again, this was conducted in two different trials, one with females from CL Xiloá ($n_{A. sagittae}$ = 26 available (12 finally tested), $n_{A. amarillo}$ = 12 (10)) and males from CL Apoyo ($n_{A. zaliosus}$ = 13 (11), $n_{A. astorquii}$ = 30 (9)), and a second one with females from CL Apoyo ($n_{A. zaliosus}$ = 25 (21), $n_{A. astorquii}$ = 36 (12)) and males from CL Xiloá ($n_{A. sagittae}$ = 22 (13), $n_{A. amarillo}$ = 36 (18)). The experimental procedure was the same as described above for the first experiment and the trials were terminated when no more naive fish of a certain sex and species were available for replacement. We counted matings between fish of the same ecomorph (that is, limnetic from CL Apoyo with limnetic from CL Xiloá or benthic from CL Apoyo with benthic from CL Xiloá) as assortative and matings between fish of the alternative ecomorph as disassortative (that is, limnetic from CL Apoyo with benthic from CL Xiloá and vice versa).

In both experiments, we tested whether mating departed from random expectations using a two-sided $\chi^2$-test with 1 degree of freedom. Two caveats should be noted for these experiments: (a) most fish of each of the species came from a single brood of a wild-caught couple, and (b) we could not attempt to replicate these experiments. These two caveats arise from the same issue: Midas cichlids require a lot of tank space to grow to maturity (>24 months old). They also reach a very large size at maturity compared to most other cichlid fishes: experimental females weighted 111.5 ± 27.4 g (mean ± s.d.) and their standard length was 135.6 ± 12.5 mm. Experimental males weighted 355.4 ± 62.9 g and their standard length was on average 196.5 ± 15.5 mm. They are extremely aggressive and would potentially kill each other under space restriction. Their large size and aggressive behaviour have two consequences. First, we were constrained in the number of fish that could be simultaneously reared to maturity. For the mate choice experiments described above we needed a large number of mature, but previously unmated (that is, naive), adult fish from four different species, all of which had to be of approximately the same size at the time of the experiments to render comparable results. Several large tanks were therefore necessary to rear all these fish at the same time. The immediate consequence of this is that we were limited to rearing one family per species. Second, because of the spatial needs of these fish and their aggressive behaviour during mating, the experiments had to be conducted in a large arena (340 × 160 × 80 cm). Only one such arena was available, and the different experiments and trials had to be conducted sequentially. This space limitation, in addition to the limited number of unmated fish, restricted our ability to replicate these experiments. Hence, interpretations based on the results of these experiments should be taken with caution. However, because the findings of these experiments are in agreement with other mate choice experiments conducted in Midas cichlids (Extended Data Table 2; Supplementary Notes), we are confident about the general robustness of these findings. Fish used in the experiments were between two and three years old. The order of trials was randomized, but only one arena was available for the experiment. Data were collected without knowledge of fish identity as the fish were individually tagged with transponders (blinding). No statistical methods were used to predetermine sample size. Mate choice experiments were approved by the German authorities (permit number G-15/89, Regierungspräsidium Freiburg, Abteilung 3, Referat 35, Veterinärwesen & Lebensmittelüberwachung, Germany).

### Plotting and statistical analysis
All plots and additional statistical analysis were performed in R/ Rstudio[117,118] using custom scripts and the following Cran R packages: corrplot[119], cowplot[120], DescTools[121], dplyr[122], formattable[123], ggplot2[124], ggpubr[125], ggridges[126], Hmisc[127], lattice[128], MASS[109], pheatmap[129], Rcolor-Brewer[130], scatterplot3D[131], stringr[132], tidyr[133]. Figures were arranged and polished in Adobe Illustrator CC 2018.

### Reporting summary
Further information on research design is available in the Nature Research Reporting Summary linked to this paper.

### Data availability
The genome assembly has been deposited at DDBJ/ENA/GenBank under accession JACBYM000000000. The version described in this paper is version JACBYM010000000. Whole-genome resequencing data for all 453 samples in the form of unmapped BAM files (PRJEB38173) and unpublished transcriptomic data (PRJNA635556) have been uploaded to ENA and NCBI/SRA, respectively. Geometric morphometric data, information on samples, and downstream data to reproduce our results can be downloaded from Dryad (https://doi.org/10.5061/dryad. bcc2fqz91)[51]. Source data are provided with this paper.

### Code availability
Custom code used for the genome assembly (https://github.com/ MartinPippel/DAmar) and custom code for genomic and morphometric analyses (https://github.com/alexnater/midas-genomics) can be accessed on GitHub.

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

**Acknowledgements** We thank the scientific computing cluster (SCC) and the animal research facility (TFA) of the University of Konstanz; the Long Read Project Team of the DRESDEN-concept Genome Center for their support; the Zoologische Staatssammlung München (Germany) for tissue samples and photographs of holotypes and paratypes of *A. globosus* (*n* = 11) and *A. supercilius* (*n* = 10); the Nicaraguan Ministerio del Ambiente y los Recursos Naturales (MARENA); the Empresa Nicaragüense de Acueductos y Alcantarillados Sanitarios (ENACAL); current and previous members of the Meyer laboratory for their help; and J. Feder, S. Flaxman, S. Gavrilets, Z. Gompert, and J. Wolf for comments and discussions. This work was principally supported by an European Research Council Advanced Grant (ERC "GenAdap" 293700) to A.M., the Zukunftskolleg of the University of Konstanz to A.F.K. and M.O., a European Molecular Biology Organization fellowship to A.F.K., a Swiss National Science Foundation fellowship to A.N. (P300PA_177852), Alexander von Humboldt fellowships to G.M.-S. and M.O., the Deutsche Forschungsgemeinschaft (DFG) to C.F.K., G.M.-S, P.F., C.D.H., and A.M. (5363031, 219669982, 243870899, 253390846, 290977748, 366312182, 423396155), and a grant of the Federal Ministry of Education and Research (01IS18026C) to M.P.

**Author contributions** Sample collection: A.F.K., G.M.-S., J.T.-D., A.H., M.O., A.M.; sample selection, extraction and library preparation: A.F.K., G.M.-S.; sequencing coordination: A.F.K.; genome assembly: M.P., A.F.K., E.W.M., genome annotation: A.F.K., P.F.; mapping, variant and genotype calling, phasing: A.F.K., A.N.; phenotyping and analyses: C.F.K., M.O., C.D.H., A.F.K.; population structure, phylogenetic and demographic analyses, selection tests: A.N., A.F.K., M.O.; gold mapping panel: C.F.K., F.H.; QTL mapping: P.F., F.H., C.F.K.; mate choice experiments: A.F.K., G.M.-S., J.T.-D.; funding acquisition: A.M., E.W.M., G.M.-S., A.F.K.; conceptualization: A.F.K., C.F.K., A.N., G.M.-S., A.M.; manuscript draft and figures: A.F.K., C.F.K., A.N., A.M. All authors contributed to, improved and approved the final manuscript.

**Competing interests** The authors declare no competing interests.

**Additional information**
**Correspondence and requests for materials** should be addressed to A.M.

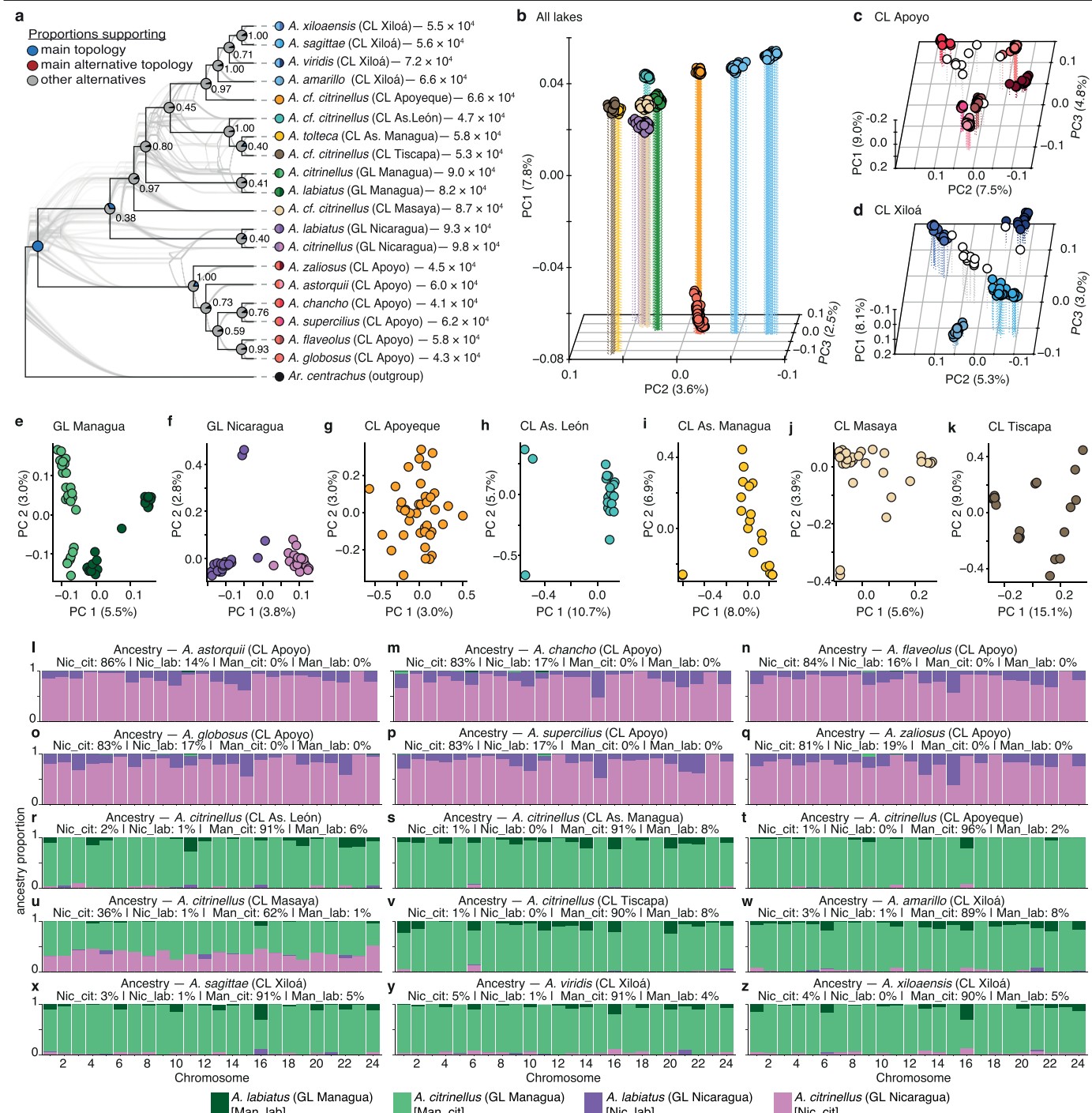

**Extended Data Fig. 1 | Ancestry relationships in the Midas cichlid species complex. a**, Inferred ASTRAL species tree, together with node support, DensiTree visualization of bootstrapped species trees, and detected levels of gene tree – species tree discordance. Numbers after species names denote genome-wide $\pi$ estimates, symbols before species names indicate colour code used in the rest of the figure (left half colour for lakes, right half colour for species). **b**–**d**, Clustering based on the first three dimensions of principal component analyses (PCAs) across all lakes (**b**), only CL Apoyo (**c**) and only CL Xiloá (**d**). Empty symbols indicate hybrids (that is, individuals with >25% admixed ancestry). **e**–**k**, Clustering based on the first two dimensions of principal component analyses of genomic variation in GLs Managua (**e**) and Nicaragua (**f**), CLs Apoyeque (**g**), As. León (**h**), As. Managua (**i**), Masaya (**j**), and Tiscapa (**k**). Please note the proportion of variation explained depends on the number of samples in an analysis and is influenced by the overall variation (which is small in single species lakes). **l**–**z**, Proportions of ancestry derived (using ChromoPainter) from the two great lake species, *A. citrinellus* (lighter green/violet) and *A. labiatus* (darker green/violet) inhabiting the two great lakes Nicaragua (violet) and Managua (green) for each crater lake population/ species and chromosome. CL Apoyo (**l**–**q**) fish derive their ancestry from GL Nicaragua. CLs As. León (**r**), As. Managua (**s**), Apoyeque (**t**), Tiscapa (**v**), and Xiloá (**w**–**z**) fish derive their ancestry from GL Managua. CL Masaya (**u**) has mixed contributions from both GL Managua and GL Nicaragua.

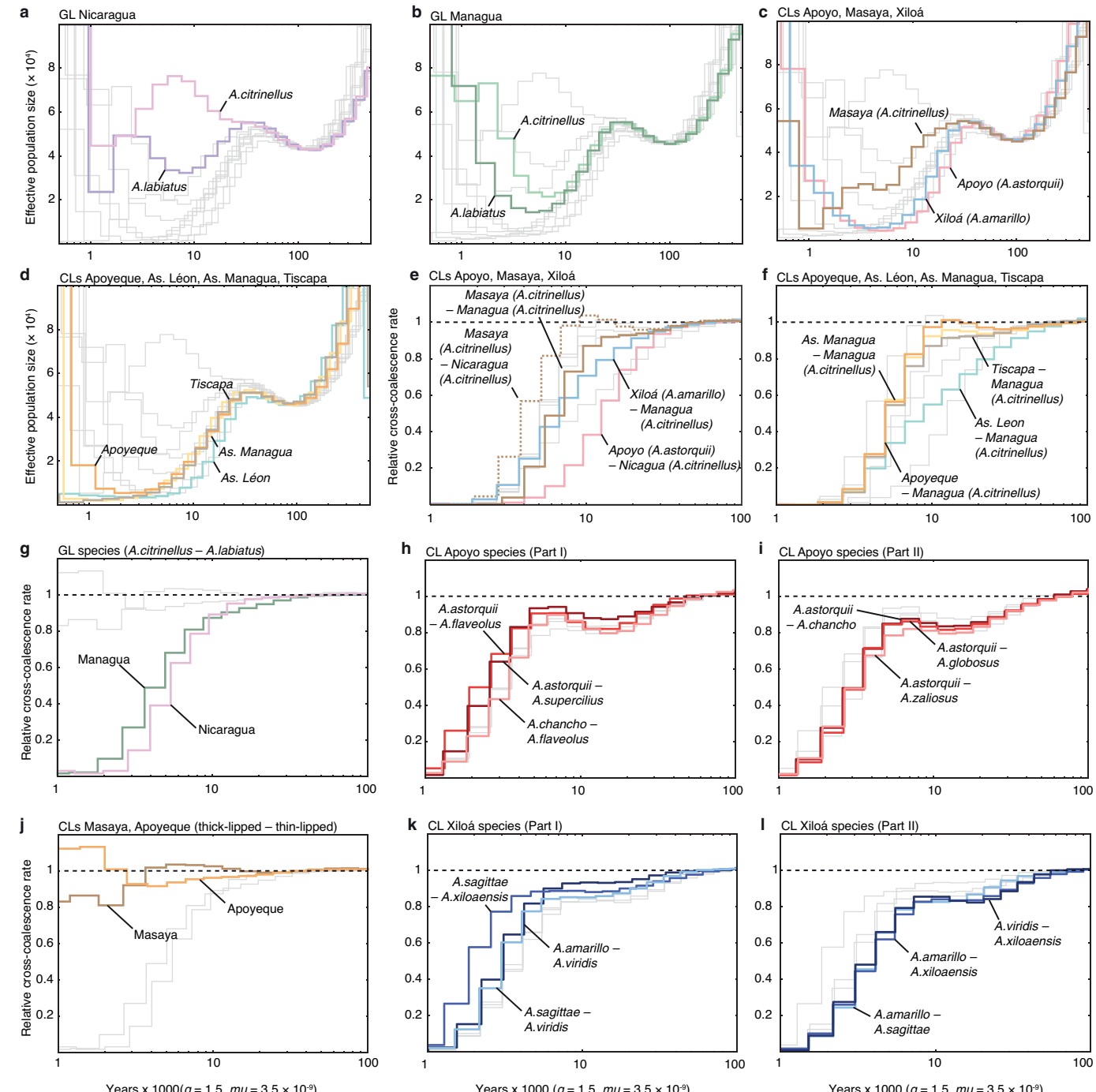

**Extended Data Fig. 2 | Multiple sequentially Markovian coalescent (MSMC) inferences. a–d**, Inferred effective population sizes through time in GL Nicaragua (**a**), GL Managua (**b**), CLs Apoyo, Masaya and Xiloá (**c**) and CLs Apoyeque, As. León, As. Managua, and Tiscapa (**d**). **e–l**, Inferred relative cross-coalescence rates for CL populations/species and GL source populations (**e** and **f**) and between sympatric species in GLs Managua and Nicaragua (**g**), CLs Apoyo (**h** and **i**), thin- and thick-lipped ecotypes in CLs Masaya and Apoyeque (**j**), Xiloá (**k** and **l**). *g* = generation time in years, mu = mutation rate per site per generation. Comparisons not highlighted in the respective plots are indicated as thin grey lines.

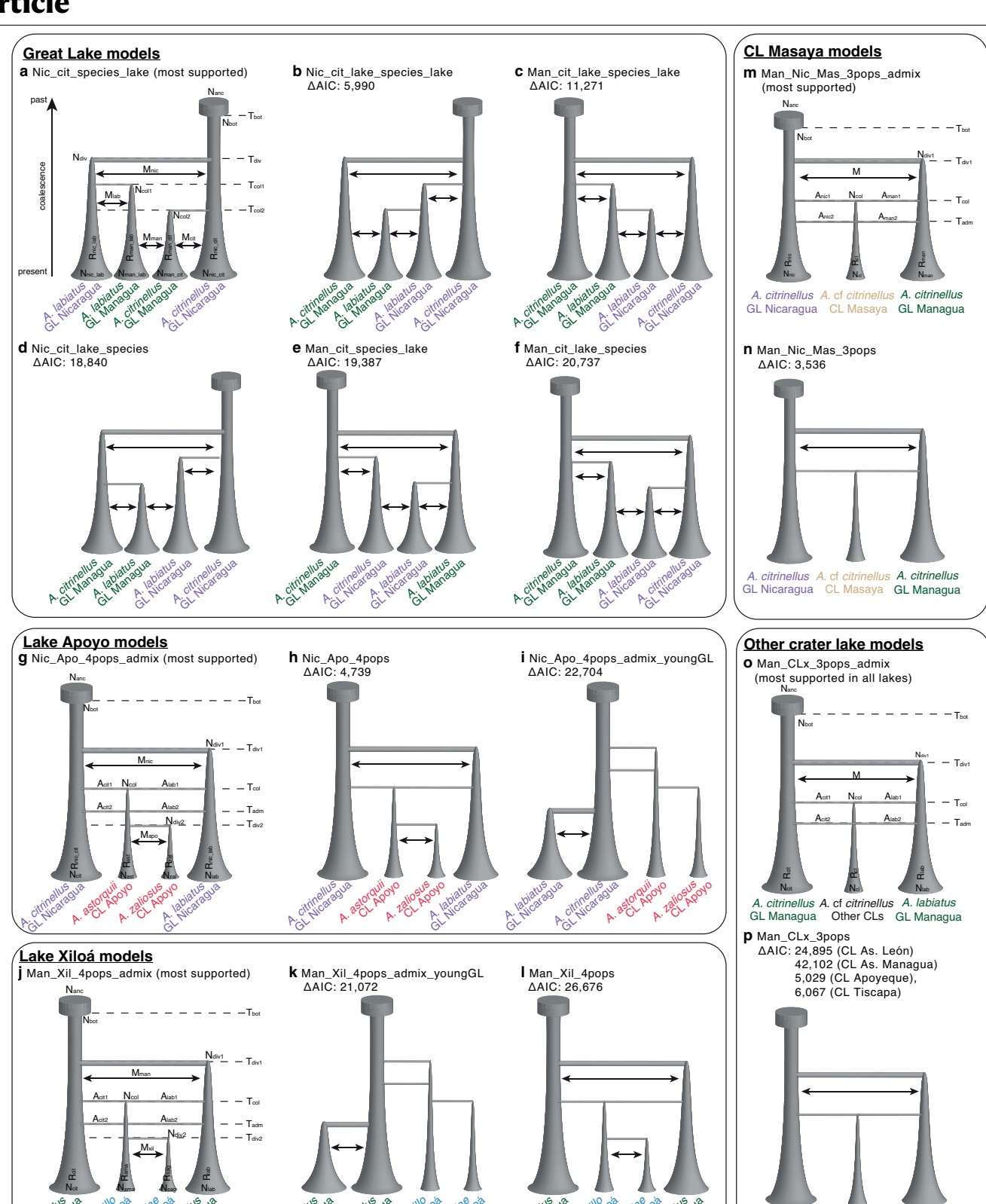

**Extended Data Fig. 3 |** See next page for caption.

**Extended Data Fig. 3 | Schematic illustrations of all tested main demographic models. a**–**f**, Models of great lake species divergence, including colonization of GL Managua from GL Nicaragua after species divergence within GL Nicaragua (**a**; most supported model), colonization of GL Managua from GL Nicaragua before species divergence within GL Nicaragua (**b**), colonization of GL Nicaragua from GL Managua before species divergence within GL Managua (**c**), colonization of GL Managua from GL Nicaragua, but species divergence within both GLs (**d**), colonization of GL Nicaragua from GL Managua after species divergence within GL Managua (**e**), and colonization of GL Nicaragua from GL Managua, but species divergence within both GLs (**f**). **g**–**i**, Models of colonization and sympatric speciation in CL Apoyo, with colonization and subsequent intralacustrine divergence from GL Nicaragua after GL species divergence with (**g**; most supported model) and without (**h**) admixture, and with colonization of CL Apoyo and subsequent intralacustrine divergence from GL Nicaragua before GL species divergence (**i**). **j**–**l**, Models of colonization and sympatric speciation in CL Xiloá, with colonization and subsequent intralacustrine divergence from GL Managua after GL species divergence with (**j**; most supported model) and without (**l**) admixture, and with colonization of CL Xiloá and subsequent intralacustrine divergence from GL Managua before

GL species divergence (**k**). **m**, **n**, Models for colonization of CL Masaya, which has support for both GLs acting as source populations, with (**m**; most supported model) and without (**n**) admixture from the source populations. **o**, **p**, Models for all other CLs with one species (Apoyeque, As. Managua, As. León, Masaya, and Tiscapa), with (**o**; most supported model) and without (**p**) admixture from the source populations of GL Managua. Parameters for time (T), population size (N), migration rate (M), admixture (A) and population growth (R) are indicated in the most supported models. Maximum-likelihood point estimates and confidence intervals for these parameters are provided in Extended Data Table 1. Abbreviations: A, admixture (proportion of gene pool that was replaced); adm, admixture event; ama, *A. amarillo*; anc, ancestral population; asl, CL As León; ast, *A. astorquii*; aye, CL Apoyeque; bot, bottleneck; cit, *A. citrinellus*; cl, crater lake; col/col1: (first) colonization event; col2, second colonization event; div/div1, (first) divergence event (speciation in GLs); div2, second divergence event; lab, *A. labiatus*; M, symmetric migration rate (probability of lineages to move between populations per generation); man, GL Managua; mas, CL Masaya; N, population size (in individuals); nic, GL Nicaragua; R, population growth; sag, *A. sagittae*; T, time (in generations); tsc, CL Tiscapa; zal, *A. zaliosus*.

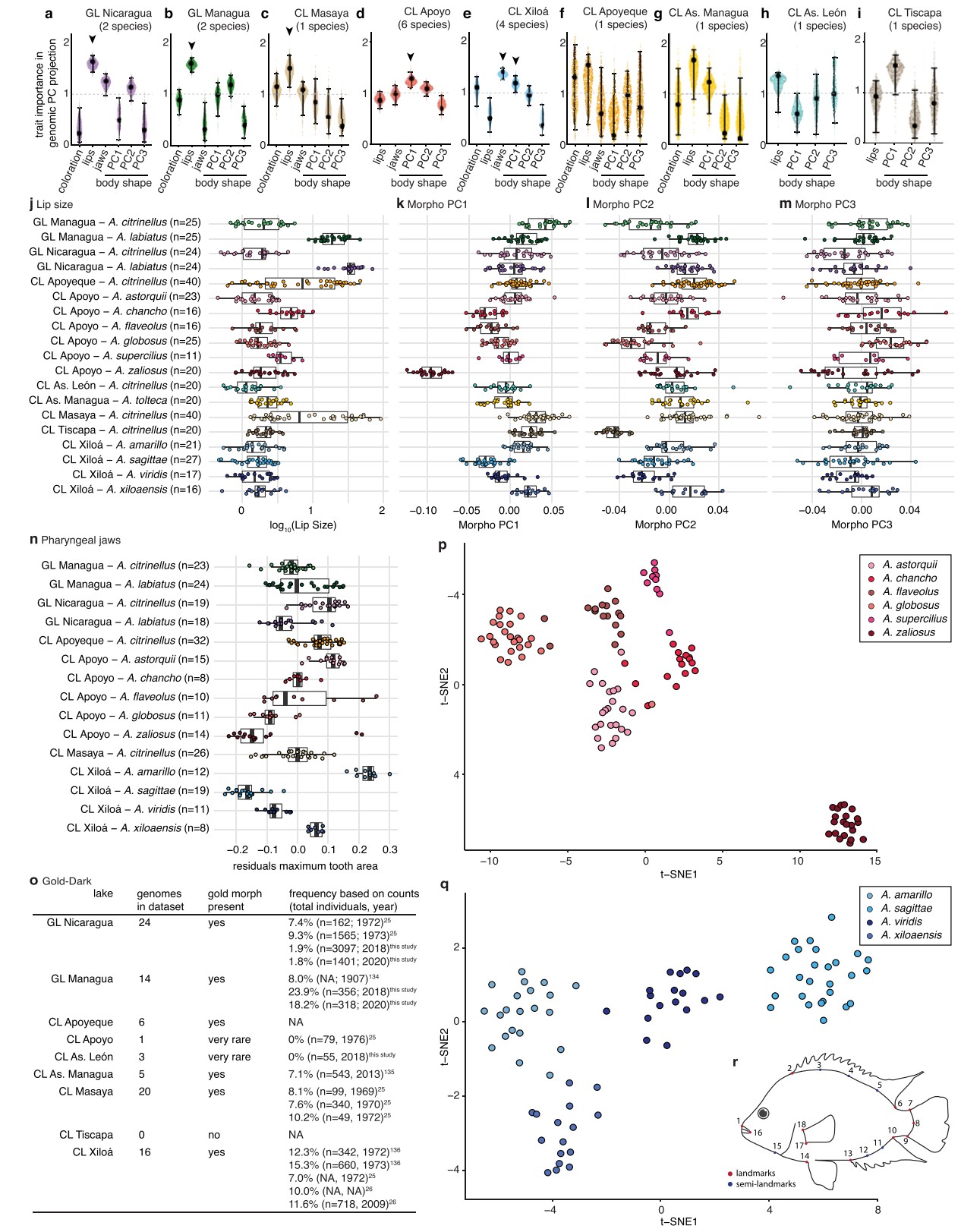

**Extended Data Fig. 4** | See next page for caption.

**Extended Data Fig. 4 | Focal phenotypic traits and their relationship to population divergence. a–i**, Variable Importance in Projection (VIP) scores (black point; based on complete data) together with 95% confidence intervals (error bars) and distributions of $n = 1,000$ non-parametric bootstrap replicates (coloured points; based on re-sampling with replacement) of partial least squares (PLS) regressions of focal traits with the primary axes of genomic divergence within GLs Nicaragua (**a**) and Managua (**b**) and CLs Masaya (**c**), Apoyo (**d**), Xiloá (**e**), Apoyeque (**f**), As. Managua (**g**), As. León (**h**) and Tiscapa (**i**). Trait VIP scores for which the lower CI bounds exceed 1, and which are thus deemed important for genomic divergence, are highlighted with an arrowhead. Note that coloration was excluded in **d**, **h** and **i**, because golden fish are virtually absent in these lakes (**o**). Pharyngeal jaw data was only available for lakes with evident population structure (**a–e**) and CL Apoyeque (**f**). **j–n**, Phenotypic distributions for lip size (**j**) (normalized by body area and $\log_{10}$-transformed), the first three axes of a principal component analysis on geometric morphometric data (**k–m**), and maximum pharyngeal tooth area (**n**) (normalized). Box plots are shown as median (solid line), interquartile range (IQR, that is, 25th–75th percentiles, box), and $\pm 1.5 \times$ IQR (whiskers) of the trait values. **o**, Number of golden specimens in our data set and estimated frequencies in the wild[25,26,134–136]. **p, q**, t-SNE based on linear discriminant scores of geometric morphometric landmark data shows that all six and four described, endemic, sympatric species in CL Apoyo (**p**) and CL Xiloá (**q**), differ in body shapes, respectively. Admixed individuals were excluded from these analyses. **r**, Landmark positions used for geometric morphometric analyses.

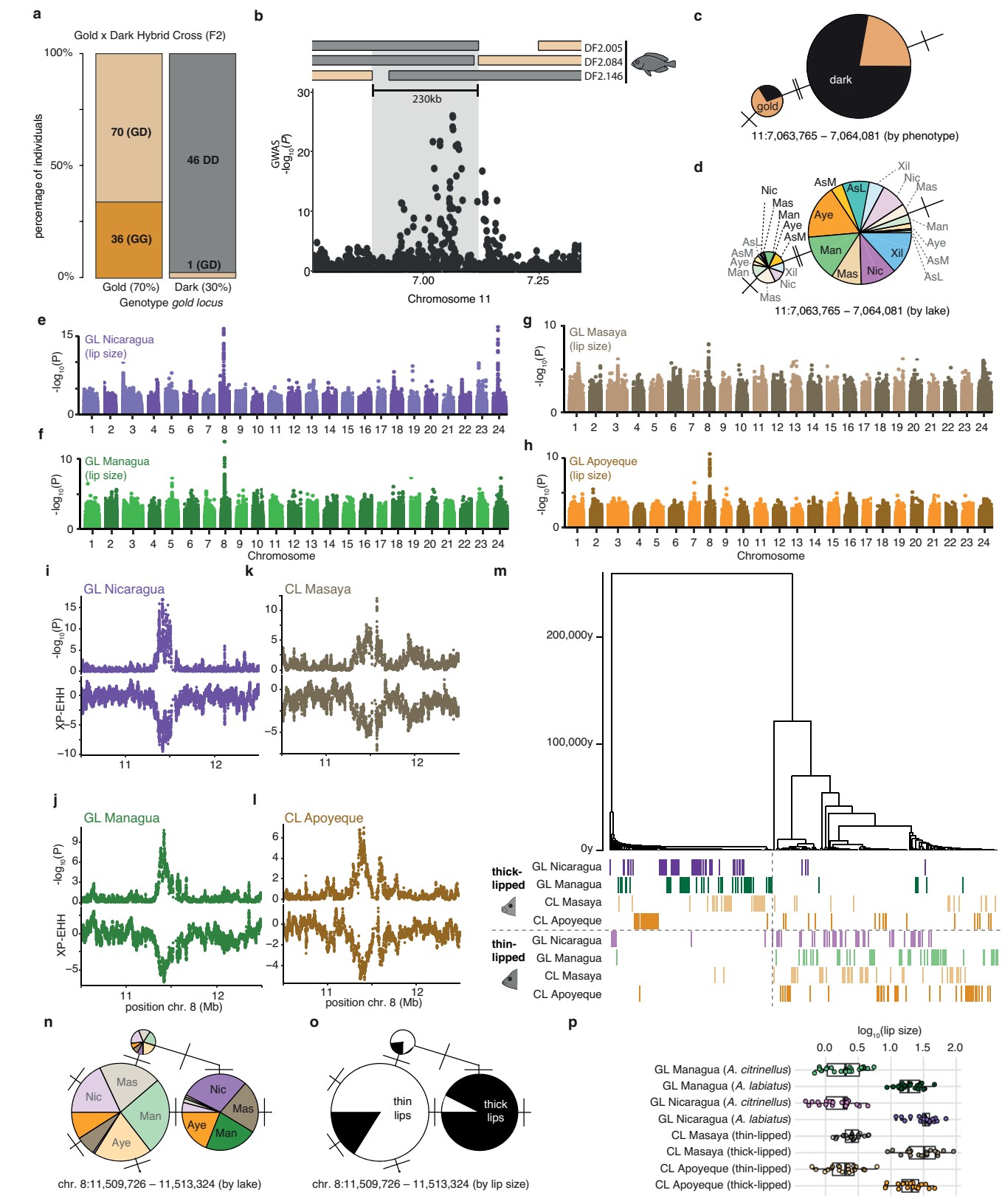

**Extended Data Fig. 5** | See next page for caption.

**Extended Data Fig. 5 | Analyses of candidate regions underlying focal traits. a–d**, Genotype-phenotype association in an $F_2$ mapping panel of a cross of a homozygote golden (GG) and dark (DD) individual. Genotypes are based on microsatellite data at chr. 11 position 7,085,452 confirming that the dark/gold phenotype constitutes a dominant Mendelian trait with high penetrance (>99%) (**a**). Three recombinants localize the causal region to a 230-kb interval (11:6,890,589–7,119,761), which overlaps with the peak of high genotype-phenotype association in natural populations (Fig. 2a) (**b**). A haplotype network of the locus containing the top-associated SNP reveals that a single haplotype is associated with the dark/gold polymorphism (**c**; phenotype colour-coded) and is shared across all lakes with golden morphs (**d**; lake colour-coded with golden individuals in brighter colours). **e–p**, Genome-wide association mapping in the four lakes that harbour thick-lipped fish identifies two regions of high association, one on chr. 8 and another slightly weaker one on chr. 24. The locus on chr. 8 is also the most highly associated one in each of the four lakes separately, whereas association on chr. 24 is strong in GL Nicaragua (**e**), much weaker in GL Managua (**f**) and CL Masaya (**g**), and essentially absent in CL Apoyeque (**h**). Cross-population extended haplotype homozygosity (XP-EHH) analyses show that haplotypes within the most highly associated region on chr. 8 (see **e–h**) are on average much shorter in thin- than thick-lipped fish for GLs Nicaragua (**i**), Managua (**j**) and CLs Masaya (**k**) and Apoyeque (**l**), providing evidence for a strong selective sweep in thick-lipped fish. Independent of their lake of origin, most haplotypes cluster by lip phenotype, suggesting that a shared genetic basis underlies lip size in all populations (**n** and **o**). Estimation of the age of the lip haplotype suggests that it is much older (30–260,000 years) than the divergence time of thin- and thick-lipped species in the great lakes (-16,730 years) (**m**). Lip size is bimodally distributed in all four lakes that harbour thick-lipped fish (**p**).

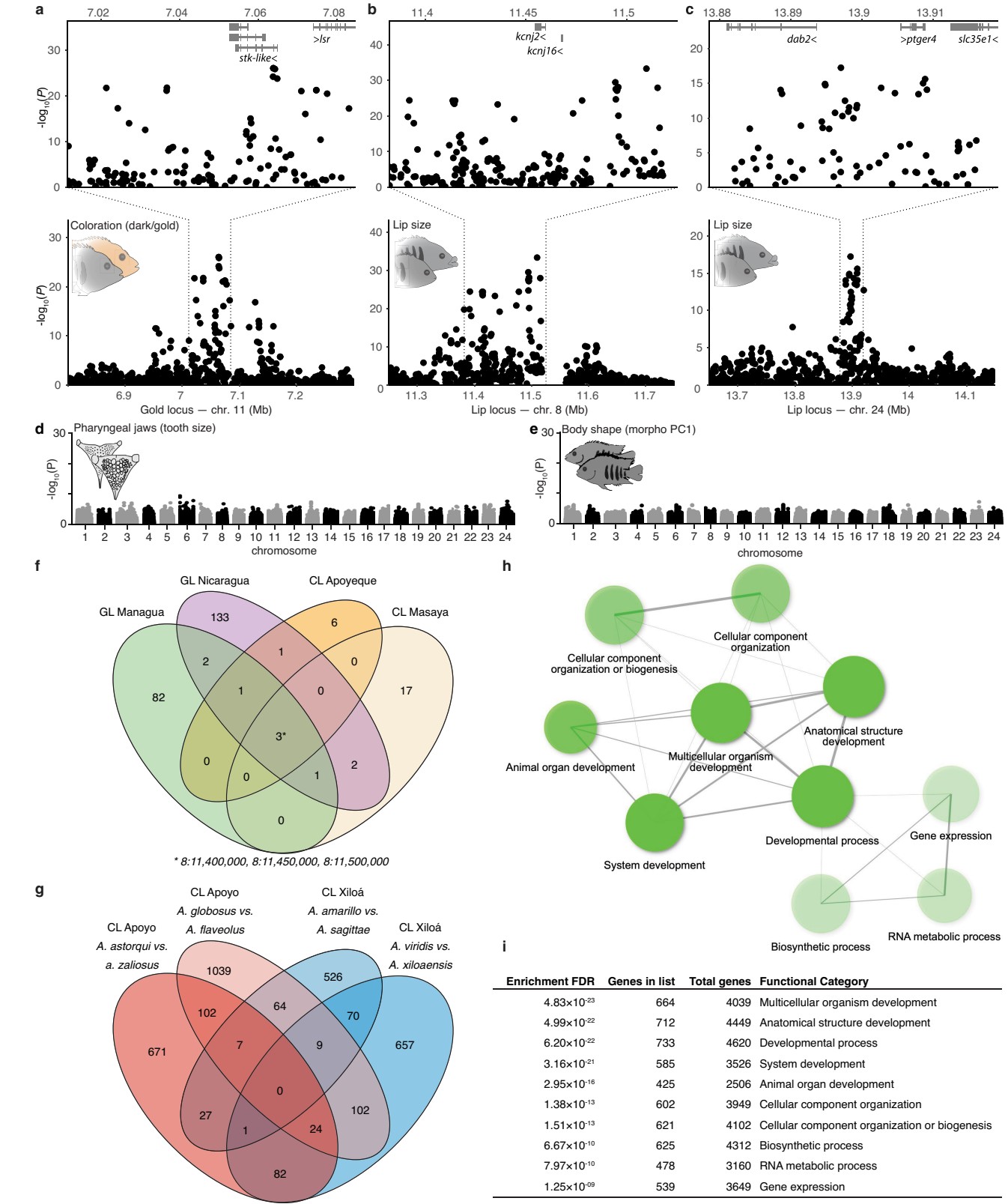

**Extended Data Fig. 6** | See next page for caption.

**Extended Data Fig. 6 | The genomic bases of adaptive divergence in Midas cichlids. a**–**c**, Gene annotation of the gold locus on chr. 11, with a serine-threonine kinase (*stk*) as a candidate gene (**a**). Serine/threonine-protein kinases regulate cell division and apoptosis and could therefore explain the progressive pigment cell loss[137]. The lip loci located on chr. 8 (**b**) and 24 (**c**) include two inward rectifier potassium channels (*kcnj2* and *kcnj16*) and a g-protein coupled receptor (*ptger4*) as top candidate genes, respectively. Kcnj2 has been associated with Andersen–Tawil Syndrome, which involves craniofacial dysmorphogenesis[138]. Kcnj16 has been linked to fluid balance[139] and could therefore trigger tissue swelling in thick-lipped individuals. Ptger4 influences tissue swelling as part of immune responses in mice[140]. **d**, **e**, Genome-wide association (GWA) mapping of pharyngeal jaw (maximum tooth size) (**d**) and body morphology (geometric morphometrics PC1 scores) (**e**). The lack of high association signals ($-\log_{10}(P) > 10$) is consistent with a polygenic bases for these traits. **f**, **g**, Sharing of genomic windows under divergent selection among comparisons of thin- and thick-lipped ecotypes and species (**f**) and among sympatric species of CLs Apoyo and Xiloá. The three windows shared in **f** are three consecutive windows centred around the lip locus (see **b**). No windows are shared among all CL Apoyo and Xiloá species, suggesting that different loci are associated with divergence of the species. **h**, **i**, Gene ontology (GO) term enrichment analysis of genomic windows classified to be under divergent selection between CL Apoyo and CL Xiloá species (see **g** and Fig. 3v–y) reveals an overrepresentation of several biological processes linked to morphological variation (for example, animal organ development, anatomical structure development, cellular component organization).

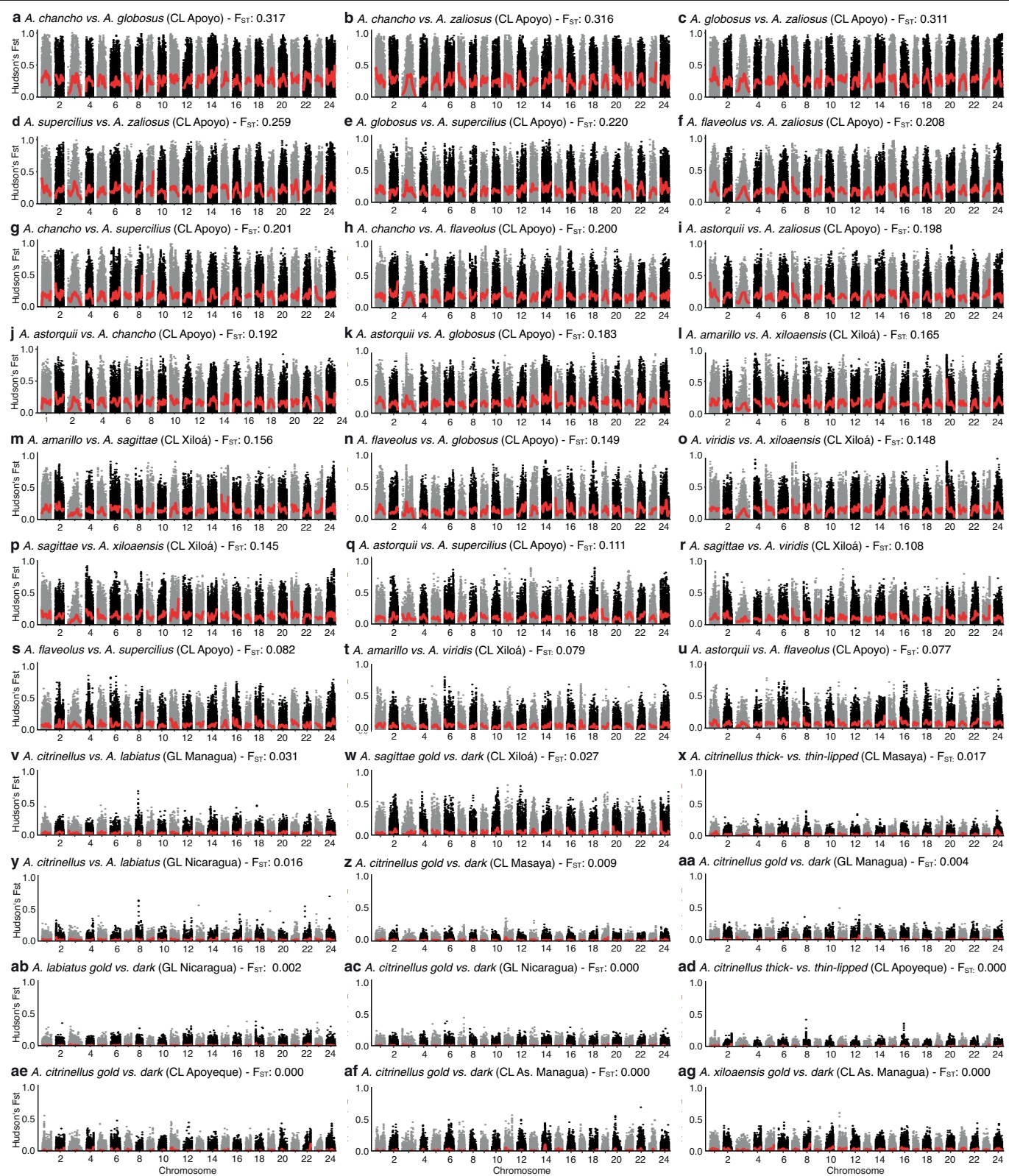

**Extended Data Fig. 7 | Pairwise $F_{ST}$ comparisons.** Genomic landscapes of differentiation among sympatric species in CLs Apoyo and Xiloá (**a**–**u**), between thin- versus thick-lipped fish (**v**, **x**, **y** and **ad**), and between dark versus golden-coloured fish within populations/species (**w**, **z**–**ac** and **ae**–**ag**). Panels are sorted by decreasing levels of overall genetic differentiation (Hudson's $F_{ST}$). Manhattan plots show $F_{ST}$ values in 10-kb non-overlapping windows and red lines indicate loess-smoothed values.

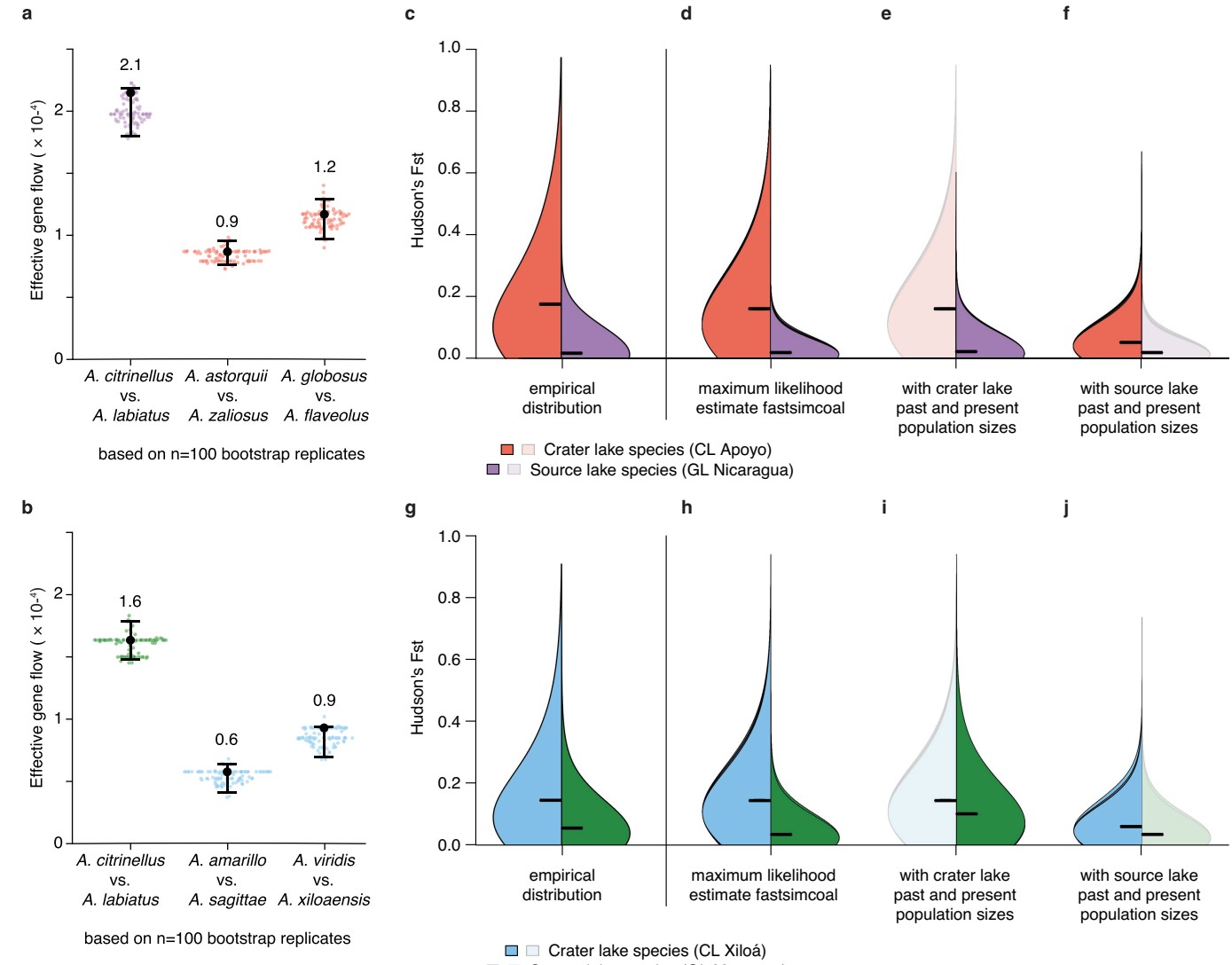

**Extended Data Fig. 8 | Effective gene flow and the role of genetic drift in genomic differentiation among Crater Lake Apoyo and Xiloá species.**
**a**, **b**, Estimates of effective gene flow (migration rate per generation) between CL Apoyo (**a**) and CL Xiloá (**b**) species pairs as well as between the species *A. citrinellus* and *A. labiatus* in GL Nicaragua (**a**) and GL Managua (**b**). Shown are maximum likelihood point estimates together with error bars denoting 95% confidence intervals obtained from $n = 100$ parametric bootstrap replicates. **c**–**j**, Distributions of $F_{ST}$ values in 10-kb windows of the empirical data (**c**, **g**) and genome-scale simulations (**d**, **h**–based on the inferred maximum-likelihood parameter estimates of our most supported demographic models). Genome-scale simulations with matched past and present population sizes

from crater lake (**e**, **i**) and great lake species (**f**, **j**) demonstrate that crater lake species pair comparisons retain a consistently higher mean $F_{ST}$ compared to the great lake comparisons, even though divergence times are more recent in the crater lakes (Extended Data Table 1). Violin plots in **d**–**f** and **h**–**j** show 100 replicates plotted on top of each other, horizontal bars are box plots of the mean values of the 100 replicates demonstrating little variance across simulation runs. Transparent parts of violin plots in **e**, **f**, **i**, **j** correspond to the respective estimates from **d**, **h** for ease of comparisons. Box plots are shown as median (solid line), interquartile (that is, 25th–75th percentiles, box), and ± 1.5 × IQR (whiskers) of the mean values (not visible due to low variance).

## Extended Data Table 1 | Parameter estimates of demographic models

| Model | Nic_cit_species_lake | Nic_Apo_4pops_admix | Man_Xil_4pops_admix | Man_Nic_Mas_3pops_admix |
|---|---|---|---|---|
| Population | $N_{anc}$: 72,402 (66,635–71,323) | $N_{anc}$: 72,026 (66,137–70,718) | $N_{anc}$: 64,609 (59,737–64,609) | $N_{anc}$: 67,016 (63,738–67,194) |
| | $N_{bot}$: 13,423 (11,833–20,353) | $N_{bot}$: 13,724 (13,469–24,839) | $N_{bot}$: 2104 (2,311–4,195) | $N_{bot}$: 4,032 (3,628–6,551) |
| | $N_{nic\_cit}$: 198,186 (145,976–198,058) | $N_{cit}$: 216,416 (195,897–283,063) | $N_{cit}$: 241,629 (136,819–367,323) | $N_{man}$: 415,077 (246,185–811,940) |
| | $N_{div}$: 1,042 (1,386–2,152) | $N_{div1}$: 753 (814–1264) | $N_{div1}$: 873 (654–1,052) | $N_{div}$: 2,324 (2,248–3,360) |
| | $N_{nic\_lab}$: 63,890 (51,164–62,870) | $N_{lab}$: 84,797 (82,112–141,762) | $N_{lab}$: 167,837 (124,268–418,791) | $N_{nic}$: 275,567 (203,784–527,098) |
| | $N_{col1}$ (*A. labiatus*): 617 (577–1,067) | $N_{col}$: 372 (380–586) | $N_{col}$: 283 (228–361) | $N_{col}$: 849 (695–1,227) |
| | $N_{man\_lab}$: 201,019 (101,925–513,912) | $N_{ast}$: 57,507 (29,315–61,528) | $N_{ama}$: 35,765 (29,913–52,742) | $N_{mas}$: 28,172 (24,644–52,025) |
| | $N_{col2}$ (*A. citrinellus*): 1,094 (1,054–1,661) | $N_{div2}$: 259 (261–367) | $N_{div2}$: 395 (380–535) | |
| | $N_{man\_cit}$: 497,656 (221,123–841,643) | $N_{zal}$: 50,911 (30,093–58,222) | $N_{sag}$: 32,411 (23,791–46,571) | |
| Admixture | | $A_{cit1}$: 0.602 (0.883–0.625) | $A_{cit1}$: 0.857 (0.943–0.771) | $A_{man1}$: 0.723 (0.617–0.82) |
| | | $A_{lab1}$: 0.398 (0.117–0.375) | $A_{lab1}$: 0.143 (0.057–0.229) | $A_{nic1}$: 0.217 (0.383–0.18) |
| | | $A_{cit2}$: 0.097 (0.04–0.182) | $A_{cit2}$: 0.321 (0.283–0.368) | $A_{man2}$: 0.311 (0.219–0.36) |
| | | $A_{lab2}$: 0.06 (0.021–0.214) | $A_{lab2}$: 0.13 (0.097–0.171) | $A_{nic2}$: 0.248 (0.207–0.295) |
| Migration | $M_{nic}$: $2.1\times10^{-4}$ ($1.8\times10^{-4}$–$2.2\times10^{-4}$) | $M_{nic}$: $2.1\times10^{-4}$ ($1.6\times10^{-4}$–$1.9\times10^{-4}$) | $M_{man}$: $1.3\times10^{-4}$ ($1.2\times10^{-4}$–$1.4\times10^{-4}$) | $M_{gls}$: $5.8\times10^{-5}$ ($4.7\times10^{-5}$–$6.4\times10^{-5}$) |
| | $M_{lab}$: $2.5\times10^{-5}$ ($1.3\times10^{-5}$–$3.1\times10^{-5}$) | $M_{apo}$: $8.7\times10^{-5}$ ($7.6\times10^{-5}$–$9.5\times10^{-5}$) | $M_{xil}$: $5.8\times10^{-5}$ ($4.1\times10^{-5}$–$6.4\times10^{-5}$) | |
| | $M_{man}$: $1.6\times10^{-4}$ ($1.5\times10^{-4}$–$1.8\times10^{-4}$) | | | |
| | $M_{cit}$: $8.3\times10^{-5}$ ($6.9\times10^{-5}$–$9.5\times10^{-5}$) | | | |
| Time | $T_{bot}$: 11,256 (9,313–12,375) | $T_{bot}$: 9,583 (7,823–10,493) | $T_{bot}$: 5,296 (4,250–5,486) | $T_{bot}$: 7,056 (5,794–7,088) |
| | $T_{div}$: 11,153 (8,978–11,927) | $T_{div1}$: 9,185 (7,470–9,666) | $T_{div1}$: 4,723 (3,878–4,934) | $T_{div}$: 5,319 (4,887–5,843) |
| | $T_{col1}$ (*A. citrinellus*): 3,827 (3,501–4,547) | $T_{col}$: 3,151 (2,996–3,785) | $T_{col}$: 2,863 (2,631–3,071) | $T_{adm}$: 904 (864–1,194) |
| | $T_{col2}$ (*A. labiatus*): 3,330 (2,885–3,764) | $T_{adm}$: 2,503 (2,397–2,977) | $T_{adm}$: 1,831 (1,712–1,955) | $T_{col}$: 1,859 (1,657–2,312) |
| | | $T_{div2}$: 2,488 (2,302–2,879) | $T_{div2}$: 1,798 (1,702–1,914) | |
| Growth | $R_{nic\_lab}$: $-3.7\times10^{-4}$ ($-3.9\times10^{-4}$– $-2.8\times10^{-4}$) | $R_{cit}$: $-2.9\times10^{-4}$ ($-3.3\times10^{-4}$– $-2.5\times10^{-4}$) | $R_{cit}$: $-9.0\times10^{-4}$ ($-1.1\times10^{-3}$– $-6.5\times10^{-4}$) | $R_{man}$: $-9.0\times10^{-4}$ ($-1.1\times10^{-3}$– $-7.1\times10^{-4}$) |
| | $R_{man\_lab}$: $-1.5\times10^{-3}$ ($-1.8\times10^{-4}$– $-1.0\times10^{-3}$) | $R_{lab}$: $-5.1\times10^{-4}$ ($-6.3\times10^{-4}$– $-4.5\times10^{-4}$) | $R_{lab}$: $-1.1\times10^{-3}$ ($-1.7\times10^{-3}$– $-1.0\times10^{-3}$) | $R_{nic}$: $-6.6\times10^{-4}$ ($-8.9\times10^{-4}$– $-5.3\times10^{-4}$) |
| | $R_{man\_cit}$: $-1.8\times10^{-3}$ ($-2.2\times10^{-4}$– $-1.4\times10^{-3}$) | $R_{ast}$: $-1.6\times10^{-3}$ ($-1.6\times10^{-3}$– $-1.1\times10^{-3}$) | $R_{ama}$: $-1.7\times10^{-3}$ ($-2.1\times10^{-3}$– $-1.5\times10^{-3}$) | $R_{mas}$: $-1.9\times10^{-3}$ ($-2.3\times10^{-4}$– $-1.5\times10^{-3}$) |
| | $R_{nic\_cit}$: $-2.4\times10^{-4}$ ($-2.6´10^{-4}$– $-2.0\times10^{-4}$) | $R_{zal}$: $-2.1\times10^{-3}$ ($-2.3\times10^{-3}$– $-1.6\times10^{-3}$) | $R_{sag}$: $-2.4\times10^{-3}$ ($-2.7\times10^{-3}$– $-2.0\times10^{-3}$) | |

| Model | Man_Aye_3pops_admix | Man_AsM_3pops_admix | Man_AsL_3pops_admix | Man_Tsc_3pops_admix |
|---|---|---|---|---|
| Population | $N_{anc}$: 113,188 (112,300–132,169) | $N_{anc}$: 102,050 (96,742–116,584) | $N_{anc}$: 94,729 (91,342–104,450) | $N_{anc}$: 118,248 (112,953–139,295) |
| | $N_{bot}$: 3037 (3,484–4,446) | $N_{bot}$: 3071 (3,637–4,596) | $N_{bot}$: 3,716 (4,402–5,711) | $N_{bot}$: 3,046 (3,354–4,283) |
| | $N_{cit}$: 79,043 (76,014–79,454) | $N_{cit}$: 78,142 (75,150–78,743) | $N_{cit}$: 81,366 (78,732–82,156) | $N_{cit}$: 75,220 (72,602–75,554) |
| | $N_{div}$: 736 (740–929) | $N_{div}$: 665 (686–882) | $N_{div}$: 646 (688–861) | $N_{div}$: 852 (841–1,033) |
| | $N_{lab}$: 58,281 (51,502–63,363) | $N_{lab}$: 64,651 (55,484–68,109) | $N_{lab}$: 51,932 (43,915–51,456) | $N_{lab}$: 91,541 (80,203–105,674) |
| | $N_{col}$: 91 (89–117) | $N_{col}$: 64 (63–85) | $N_{col}$: 148 (149–197) | $N_{col}$: 31 (30–39) |
| | $N_{aye}$: 22,378 (18,537–36,369) | $N_{asm}$: 23,806 (18,068–36,264) | $N_{asl}$: 14,044 (11,289–17,887) | $N_{tsc}$: 23,673 (14,934–32,009) |
| Admixture | $A_{cit1}$: 0.864 (0.907–0.766) | $A_{cit1}$: 0.763 (0.804–0.685) | $A_{cit1}$: 0.781 (0.811–0.671) | $A_{cit1}$: 0.803 (0.881–0.721) |
| | $A_{lab1}$: 0.136 (0.093–0.234) | $A_{lab1}$: 0.237 (0.196–0.315) | $A_{lab1}$: 0.219 (0.189–0.329) | $A_{lab1}$: 0.197 (0.119–0.279) |
| | $A_{cit2}$: 0.148 (0.102–0.179) | $A_{cit2}$: 0.206 (0.185–0.227) | $A_{cit2}$: 0.174 (0.155–0.211) | $A_{cit2}$: 0.239 (0.186–0.281) |
| | $A_{lab2}$: 0.056 (0.018–0.08) | $A_{lab2}$: 0.012 (0.002–0.028) | $A_{lab2}$: 0.023 (0.004–0.036) | $A_{lab2}$: 0.012 (0.002–0.054) |
| Migration | $M_{man}$: $1.3\times10^{-4}$ ($1.19\times10^{-4}$–$1.3\times10^{-4}$) | $M_{man}$: $1.3\times10^{-4}$ ($1.3\times10^{-4}$–$1.3\times10^{-4}$) | $M_{man}$: $1.3\times10^{-4}$ ($1.2\times10^{-4}$–$1.3\times10^{-4}$) | $M_{man}$: $1.2\times10^{-4}$ ($1.2\times10^{-4}$–$1.2\times10^{-4}$) |
| Time | $T_{bot}$: 10,394 (9,991–11,004) | $T_{bot}$: 9,457 (8,970–10,004) | $T_{bot}$: 11,573 (11,273–12,534) | $T_{bot}$: 9,264 (8,721–9,632) |
| | $T_{div}$: 9,281 (8,876–9,975) | $T_{div}$: 8,636 (8,061–9,198) | $T_{div}$: 10,825 (10,568–11,971) | $T_{div}$: 7,557 (7,050–8,063) |
| | $T_{adm}$: 430 (411–526) | $T_{adm}$: 512 (506–622) | $T_{adm}$: 1,125 (1,100–1,283) | $T_{adm}$: 422 (390–470) |
| | $T_{col}$: 598 (566–709) | $T_{col}$: 832 (825–1,025) | $T_{col}$: 1,947 (1,922–2,282) | $T_{col}$: 535 (488–584) |
| Growth | $R_{cit}$: $-3.5\times10^{-4}$ ($-3.5\times10^{-4}$– $-3.1\times10^{-4}$) | $R_{cit}$: $-3.7\times10^{-4}$ ($-3.8\times10^{-4}$– $-3.1\times10^{-4}$) | $R_{cit}$: $-2.8\times10^{-4}$ ($-2.7\times10^{-4}$– $-2.3\times10^{-4}$) | $R_{cit}$: $-3.9\times10^{-4}$ ($-4.2\times10^{-4}$– $-3.5\times10^{-4}$) |
| | $R_{lab}$: $-4.7\times10^{-4}$ ($-4.9\times10^{-4}$– $-4.1\times10^{-4}$) | $R_{lab}$: $-5.3\times10^{-4}$ ($-5.5\times10^{-4}$– $-4.6\times10^{-4}$) | $R_{lab}$: $-4.0\times10^{-4}$ ($-4.0\times10^{-4}$– $-3.4\times10^{-4}$) | $R_{lab}$: $-6.2\times10^{-4}$ ($-6.6\times10^{-4}$– $-5.6\times10^{-4}$) |
| | $R_{aye}$: $-9.2\times10^{-3}$ ($-1.0\times10^{-2}$– $-7.7\times10^{-3}$) | $R_{asm}$: $-7.1\times10^{-3}$ ($-7.4\times10^{-4}$– $-5.5\times10^{-3}$) | $R_{asl}$: $-2.3\times10^{-3}$ ($-2.4\times10^{-4}$– $-1.8\times10^{-3}$) | $R_{tsc}$: $-1.2\times10^{-2}$ ($-1.3\times10^{-2}$– $-1.1\times10^{-2}$) |

Maximum-likelihood parameter point estimates with 95% confidence intervals in parentheses. Abbreviations: A, admixture (proportion of gene pool that was replaced); adm, admixture event ; ama, *A. amarillo*; anc, ancestral population; asl, CL As León; ast, *A. astorquii*; aye, CL Apoyeque; bot, bottleneck; cit, *A. citrinellus*; cl, crater lake; col/col1: (first) colonization event; col2, second colonization event; div/div1, (first) divergence event (speciation in GLs); div2, second divergence event; lab, *A. labiatus*; M, symmetric migration rate (probability of lineages to move between populations per generation); man, GL Managua; mas, CL Masaya; N, population size (in individuals); nic, GL Nicaragua; R, population growth; sag, *A. sagittae*; T, time (in generations); tsc, CL Tiscapa; zal, *A. zaliosus*.

**Extended Data Table 2 | Mate choice experiments**

| Lake | Species | Setting | n | Finding | Assortative | P-value | Reference |
|---|---|---|---|---|---|---|---|
| **dark *versus* gold** | | | | | | | |
| CL Xiloá | *A. sagittae* | Field study: observational | 223 | assortative | 95% | G-test=37.2, df=3, *P*<0.001 | 26 |
| CL Xiloá | *A. xiloaensis* | Field study: observational | 136 | assortative | 77% | G-test=9.6, df=3, P=0.022 | 26 |
| Unknown | Unknown | Mesocosms: observational | 38 | assortative | 71% | $\chi^2$=10.99, *P*=0.0009 | 25 |
| CL Xiloá | Unknown | Field study: observational | 171 | assortative | 95% | – | 136 |
| CL Xiloá | Unknown | Field study: observational | 330 | assortative | >99% | – | 136 |
| **thin-lipped *versus* thick-lipped** | | | | | | | |
| CL Apoyeque | *A. citrinellus* | Field study: observational | 68 | assortative | 96% | $\chi^2$=25.77, *P*<0.0001 | 23 |
| GL Nicaragua | *A. citrinellus* – *A. labiatus* | Lab: group mate choice | 25 | assortative | 100% | $\chi^2$=22.18, *P*<0.0001 | 23 |
| Inter-lacustrine | *A. labiatus* (GL Nicaragua) – *A. citrinellus* (CL Masaya) | Lab: group mate choice | 7 | assortative | 86% | – | 141 |
| **species with distinct body shape types (e.g. limnetic versus benthic)** | | | | | | | |
| CL Apoyo | all species | Field study: observational | NA | assortative | NA | – | 142 |
| CL Apoyo | *A. zaliosus* – "*A. citrinellus*" | Lab: group mate choice | 30 | assortative | 100% | $\chi^2$=30.28, *P*<0.001 | 141 |
| CL Apoyo | *A. zaliosus* – *A. astorquii* | Lab: group mate choice | 19 | assortative | 89% | $\chi^2$=11.842, *P*=0.00058 | this study |
| CL Xiloá | *A. sagittae* – *A. xiloaensis* | Field study: observational | 359 | assortative | 100% | | 26 |
| CL Xiloá | *A. sagittae* – *A. amarillo* | Lab: group mate choice | 12 | assortative | 100% | $\chi^2$=12.00, *P*=0.00053 | this study |
| Inter-lacustrine | *A. zaliosus* (CL Apoyo) – *A. citrinellus* (CL Masaya) | Lab: group mate choice | 4 | assortative | 100% | | 141 |
| Inter-lacustrine | ♀ *A. sagittae*, *A. amarillo* (CL Xiloá) ♂ *A. zaliosus*, *A. astorquii* (CL Apoyo) | Lab: group mate choice | 12 | random | 58% | $\chi^2$=0.33, *P*=0.5637 | this study |
| Inter-lacustrine | ♀ *A. zaliosus*, *A. astorquii* (CL Apoyo) ♂ *A. sagittae*, *A. amarillo* (CL Xiloá) | Lab: group mate choice | 28 | disassortative | 7% | $\chi^2$=18.75, *P*=0.000015 | this study |
| Inter-lacustrine | ♀ *A. astorquii* (CL Apoyo) ♂ *A. sagittae*, *A. amarillo* (CL Xiloá) | Lab: group mate choice | 12 | random | 58% | $\chi^2$=0.33, *P*=0.5637 | this study |
| Inter-lacustrine | ♀ *A. sagittae*, *A. amarillo* (CL Xiloá) ♂ *A. astorquii* (CL Apoyo) | Lab: group mate choice | 5 | random | 60% | $\chi^2$=0.20, *P*=0.65472 | this study |

Summary of previous and new mate choice experiments and observational studies[23,25,26,136,141,142]. Two-sided Chi-Square tests were used to determine statistical significance.

# nature research

| | |
|---|---|

# Reporting Summary

Nature Research wishes to improve the reproducibility of the work that we publish. This form provides structure for consistency and transparency in reporting. For further information on Nature Research policies, see Authors & Referees and the Editorial Policy Checklist .

## Statistics

For all statistical analyses, confirm that the following items are present in the figure legend, table legend, main text, or Methods section.

| n/a | Confirmed | |
|---|---|---|
| ☐ | ☒ | The exact sample size (*n*) for each experimental group/condition, given as a discrete number and unit of measurement |
| ☐ | ☒ | A statement on whether measurements were taken from distinct samples or whether the same sample was measured repeatedly |
| ☐ | ☒ | The statistical test(s) used AND whether they are one- or two-sided<br>*Only common tests should be described solely by name; describe more complex techniques in the Methods section.* |
| ☒ | ☐ | A description of all covariates tested |
| ☒ | ☐ | A description of any assumptions or corrections, such as tests of normality and adjustment for multiple comparisons |
| ☐ | ☒ | A full description of the statistical parameters including central tendency (e.g. means) or other basic estimates (e.g. regression coefficient) AND variation (e.g. standard deviation) or associated estimates of uncertainty (e.g. confidence intervals) |
| ☐ | ☒ | For null hypothesis testing, the test statistic (e.g. *F*, *t*, *r*) with confidence intervals, effect sizes, degrees of freedom and *P* value noted<br>*Give P values as exact values whenever suitable.* |
| ☒ | ☐ | For Bayesian analysis, information on the choice of priors and Markov chain Monte Carlo settings |
| ☒ | ☐ | For hierarchical and complex designs, identification of the appropriate level for tests and full reporting of outcomes |
| ☒ | ☐ | Estimates of effect sizes (e.g. Cohen's *d*, Pearson's *r*), indicating how they were calculated |

*Our web collection on statistics for biologists contains articles on many of the points above.*

## Software and code

Policy information about availability of computer code

| | |
|---|---|
| Data collection | Geometric morphometric landmark data acquisition was performed with tpsDig v.2.32. Lip size and maximum pharyngeal jaw tooth size were measured from photographs with Fiji (ImageJ) v.2.0.0. |
| Data analysis | Genome assembly and annotation:<br>MARVEL and custom code for genome assembly (https://github.com/MartinPippel/DAmar), DBdust (https://github.com/thegenemyers/DAZZ_DB; commit: 0bd5e07), datander and TANmask (MARVEL developmental branch), daligner (https://github.com/thegenemyers/DALIGNER), Bionano Solve v3.1, pbalign (https://github.com/PacificBiosciences/pbalign; commit: 0669a4e;), blasr (5.3.2-a579bd5), 3d-dna (https://github.com/theaidenlab/3d-dna; commit 5baf854), Juicer v.1.7.6, Arrow (https://github.com/PacificBiosciences/GenomicConsensus; commit c92ef5d), freebayes v.1.1.0, samtools v.1.8, bcftools v.1.7 consensus, LASTZ v.1.02.00, EvidenceModeler v.1.1.1, Braker v.2.0.4, HISAT v.2.1.0, Stringtie v.1.3.3b, exonerate v.2.4.0, PASA v.2.0.2, Trinity v.2.6.0, Cufflinks v.2.2.1, BLASTp v.2.2.31, gVolante v.1.2.1, BUSCO v.2/v3, gffreads v.0.11.4<br><br>Population genomics, phylogenomics, GWAS:<br>Picard tools v.2.9.4, BWA mem v.0.7.15, freebayes v.1.1.0, vcftools v.0.1.1, plink v.1.90/v. 2.00, gem-mappability v.1.315 (GEM library), SHAPEIT2 v.2.r900, FastEPRR v.2.0, RAxML v.8, ASTRAL III v5.6.1, BEAST v.2.4.7, PhyParts v.0.0.1, ChromoPainter v.2, GLOBETROTTER v.1, MSMC v.2.1.2, Fastsimcoal v.2.6, ANGSD v.0.929, EIGENSOFT v.7.2.1, Admixture v.1.3.0, EMMAX beta-07Mar2010, Saguaro r44, pegas v.0.11 (R package), Relate v.1.0.16, MSMS v.1.3, libsequence v.1.9.8, BEDTools v2.29.2, ShinyGO v.0.61, REHH v.2.0.2 (R package), dadi v.1.7.0 (python package), custom code (https://github.com/alexnater/midas-genomics)<br><br>QTL mapping:<br>Trimmomatic v.0.36, PicardTools v.1.141, bwa-mem v.0.7.15, freebayes v.1.3.0, JoinMap v.4.0, R/qtl (R package)<br><br>Geometric morphometrics and partial least squares regressions:<br>geomorph v3.0.6 (R package), MASS v.7.3 (R package), plsdepot v.0.1.17 (R package), custom code (https://github.com/alexnater/midas- |

October 2018

genomics)

For manuscripts utilizing custom algorithms or software that are central to the research but not yet described in published literature, software must be made available to editors/reviewers. We strongly encourage code deposition in a community repository (e.g. GitHub). See the Nature Research guidelines for submitting code & software for further information.

## Data

Policy information about availability of data

All manuscripts must include a data availability statement. This statement should provide the following information, where applicable:

- Accession codes, unique identifiers, or web links for publicly available datasets
- A list of figures that have associated raw data
- A description of any restrictions on data availability

The genome assembly has been deposited at DDBJ/ENA/GenBank under accession JACBYM000000000. The version described in this paper is version JACBYM010000000. Whole-genome resequencing data of all 453 samples in form of unmapped BAM files (PRJEB38173) and previously unpublished transcriptomic data (PRJNA635556) have been deposited to ENA and NCBI/SRA, respectively. Geometric morphometric data, information on samples, and downstream data to reproduce our results can be downloaded from Dryad (10.5061/dryad.bcc2fqz91).

# Field-specific reporting

Please select the one below that is the best fit for your research. If you are not sure, read the appropriate sections before making your selection.

☐ Life sciences ☐ Behavioural & social sciences ☒ Ecological, evolutionary & environmental sciences

For a reference copy of the document with all sections, see nature.com/documents/nr-reporting-summary-flat.pdf

# Ecological, evolutionary & environmental sciences study design

All studies must disclose on these points even when the disclosure is negative.

| Study description | This study comprises a de novo genome assembly (single sample), whole-genome re-sequencing of 453 samples (sample sizes of 10-24 per species/lake), analysis of a mapping panel for determining the genetic basis of the dark/gold polymorphism (one cross), a quantitative trait locus mapping panel for pharyngeal jaws and body morphology (one cross), and mate choice experiments (two experiments with two trials each. For the mate choice experiments, the first experiment tested mate choice between sympatric species from two different lakes (with trials corresponding to the two lakes). The second experiment tested mate choice between allopatric species (trials one and two corresponded to fish from one sex of lake A vs. fish of the other sex from lake B and vice versa). Each trial was analyzed independently and the response variable was categorical with the two levels assortative or disassortative. The analysis tested the probability of assortative over disassortative mating. |
|---|---|
| Research sample | This work is based on a new, high-quality reference genome of Amphilophus citrinellus and 453 re-sequenced genomes that were sampled between 2003 and 2015. We aimed to sample at least 20 individuals per species, lake and/or ecomorph whenever possible. Sample sizes for the resequenced genomes (in brackets) are: GL Nicaragua A. citrinellus [24], GL Nicaragua A. labiatus [24], GL Managua A. citrinellus [25], GL Managua A. labiatus [25], CL Apoyeque A. cf. citrinellus (thin-lipped and thick-lipped) [20+20], CL Apoyo A. astorquii [23], CL Apoyo A. chancho [16], CL Apoyo A. flaveolus [16], CL Apoyo A. globosus [25], CL Apoyo A. supercilius [10], CL Apoyo A. zaliosus [21], CL Apoyo admixed individuals [9], CL As. León A. cf. citrinellus [20], CL As. Managua A. tolteca [20], CL Xiloá A. amarillo [21], CL Xiloá A. sagittae [27], CL Xiloá A. viridis [24], CL Xiloá A. xiloaensis [16], CL Xiloá admixed individuals [14], CL Masaya A. cf. citrinellus (thin-lipped and thick-lipped) [20+20], CL Tiscapa A. cf. citrinellus [20]. Age could not be determined reliably in wild-caught fish, but all sampled individuals were, based on their size, adult fish (standard length =14.54 ± 3.43 cm (mean ± SD)). Whenever possible, we determined the sex of individuals. Of those that could be sexed, 58% were males and 42% females. We used existing RNA-seq data sets from lip tissue (Manousaki et al. 2013; doi: 10.1111/mec.12034), and whole-body extractions at 1-day (Franchini et al. 2019; doi:10.1093/molbev/msz168) and 1-month post-hatch (Franchini et al. 2016; doi:10.1093/gbe/evw097) for the generation of the genome annotation. For mate choice experiments, we used a single brood of wild-caught fish for each of the four studies species. All fish of these broods that survived to adulthood were considered in this study. Individuals in each trial were selected randomly. |
| Sampling strategy | Fish were caught with gill nets or by harpooning in crater lakes Asososca León, Asososca Managua, Apoyeque, Apoyo, Masaya Tiscapa, and Xiloá. Fish or tissue samples from the great lakes Nicaragua and Managua were obtained from local fishermen, mostly the big fish market in Granada (Lake Nicaragua fish) and Mateares (Lake Managua fish). To obtain sufficient power (estimated based on previous studies in non-model organisms; the necessary sample size is impossible to assess in advance as it depends, for example, on the genetic architecture of the respective trait or on demographic parameters that were unknown prior to this study) for the conducted analyses we aimed to sample around 20 individuals per species, lake and/or ecomorph. For dark/gold and thin-/thick-lipped morphs we also aimed for balanced sample numbers whenever possible. However, several species and morphs are very rare in certain lakes, so sample size was dictated by the number of available samples. In these cases we aimed to include the maximum number of available samples. |
| Data collection | Adult fish were collected in field expeditions of the Meyer lab to Nicaragua between 2003 and 2015. Photographs were taken in the field using a digital camera. Tissues (fin and muscle tissue) were dissected with scissors and scalpels and stored in pure Ethanol before DNA extraction. The following authors participated in field trips: Axel Meyer, Andreas F. Kautt, Andreas Härer, Gonzalo Machado-Schiaffino, and Julian Torres-Dowdall. Genomic libraries preparation and data generation was performed at the University of Konstanz, the Max Planck Institute of Molecular Cell Biology and Genetics in Dresden, BGI Hong Kong, Rockefeller University, and PhaseGenomics. |

For mate choice experiment, data was collected by trapping the couple defending a nest (most commonly with eggs) and determining the identity of the fish by reading their individual transponders (all adult fish at the University of Konstanz Animal Facility are individually tagged with a transponder). After pair formation, fish were removed from the experiment and relocated into stock tanks. These experiments were performed by Andreas F. Kautt, Gonzalo Machado-Schiaffino, and Julian Torres-Dowdall.

**Timing and spatial scale**

Adult fish were collected in field expeditions of the Meyer lab to Nicaragua between 2003 and 2015. To maximally reduce allometric effects as well as to avoid sampling untransformed golden Midas cichlid fish (that may transform only after >1 year), only adult fish were sampled. We focused on Nicaraguan lakes that were known to harbor natural populations of Midas cichlids based on previous studies. Excursions with sampling have been conducted in several years between 2003 and 2015, as sampling included many lakes and certain species and morphs were difficult to obtain in sufficient numbers (see sampling strategy).

**Data exclusions**

Whenever data were excluded (e.g. lakes, traits, SNPs, genomic regions) this is indicated and described in the text. Specifically, lakes were excluded when there was no variation in focal traits (e.g. golden-colored fish). For the partial least squares regression, dark/gold coloration was also excluded for lakes where golden fish are very rare (<1%; CL Apoyo and CL As. Leon). Regarding phenotypic traits — as discussed in the main text — we focused on four major axes of divergence (body shape, pharyngeal jaws, lip size, and dark/gold coloration). The criteria used for variant calling and filtering were based on established bioinformatic pipelines, that is, we applied standard quality filters (mapping quality >=30, base quality >=20). Hard variant site filters were applied using the vcffilter script from the vcflib package (https://github.com/vcflib/vcflib) (-s -f "QUAL > 1 & QUAL / AO > 10 & SAF > 0 & SAR > 0 & RPR > 1 & RPL > 1") to remove low-quality variant sites. Unplaced scaffolds showed signs of low quality including aberrantly high SNP density and heterozygosity and were therefore excluded from analyses. We hard-masked the following sites in the assembly: i) sites with a sequencing coverage across all Midas cichlid samples more than four standard deviation above the mean; ii) sites with a mappability score of less than 0.5. Mappability was calculated with the gem-mappability program v.1.315 of the GEM library74, using a k-mer size of 150 bp and allowing for up to two mismatches; iii) sites within 5 bp of an InDel variant; iv) sites within annotations of repetitive regions (repCov2), gaps, low complexity regions, or tandem repeats produced by MARVEL; v) sites in non-overlapping 10-kb windows with an average root mean square of mapping quality less than 30. In total, we masked 37.99% of all sites in the reference genome. For the machine learning analyses to detect divergent selection, we excluded regions that were not present (due to filtering) in all comparisons to allow direct comparison among species and lake populations.

**Reproducibility**

To test and demonstrate reproducibility and robustness we used bootstrap replication for:

a) species tree inference (gene trees were obtained with RaxML v.8 using the rapid bootstrap analysis and search of best-scoring maximum likelihood tree (option a) under a GTR+G substitution model and including 100 bootstrap replicates. Subsequent species tree estimation was inferred using ASTRAL III v5.6.1, from all individual unrooted gene trees under the multi-species coalescent model. A total of 200 bootstrap trees were obtained and used to plot the density tree.)

b) demographic inference (to estimate confidence intervals around the maximum likelihood parameter point estimates, we applied a parametric bootstrapping approach.)

c) partial least squares regressions (1,000 non-parametric bootstrap replicates (i.e. resampling with replacement))

For the genome-scale coalescent simulations we conducted 100 simulations of entire genomes (both with estimated and modified parameters), which were very consistent and showed a close fit to those in our empirical data.

We performed two mate choice (sub)experiments (intra-lacustrine and inter-lacustrine mate choice) that consisted of two trials each. In the first trial of the first experiment (intra-lacustrine mate choice), 114 adult fish (>2 years old) were available. A total of 48 of them were tested, comprising the limnetic species A. zaliosus (nfemales = 25 available (13 tested), nmales = 13 (12)) and the benthic species A. astorquii (nfemales = 30 (10), nmales = 46 (13)) from CL Apoyo. In the second trial, 115 (33 tested) adult fish were available, comprising the limnetic species A. sagittae (nfemales = 26 (6), nmales = 17 (6)) and the benthic species A. amarillo (nfemales = 12 (11), nmales = 60 (10)) from CL Xiloá. For these experiment we were restricted to using a single brood from wild-caught parents per species. In a second experiment, we aimed to test whether fish would mate assortatively by ecomorph even when exposed to fish from a different lake (inter-lacustrine mate choice). To test this hypothesis, we exposed five females of each of the limnetic and benthic species from one of the two crater lakes CL. Apoyo and Xiloá (the only two lakes harboring small adaptive radiations) to five males of each of the limnetic and benthic species from the other lake. Again, this was conducted in two different trials, one with females from CL Xiloá (nA. sagittae = 26 available (12 finally tested), nA. amarillo = 12 (10)) and males from CL Apoyo (nA. zaliosus = 13 (11), nA. astorquii = 30 (9)), and a second one with females from CL Apoyo (nA. zaliosus = 25 (21), nA. astorquii = 36 (12)) and males from CL Xiloá (nA. sagittae = 22 (13), nA. amarillo = 36 (18)). The lack of replication for both experiments and the use of a limited number of families is a limitation of this experiment, which was discussed during the review process and is acknowledged in the publication.

Raw and intermediate data as well as code to reproduce results and figures are publicly available:
- Reference assembly can be accessed at NCBI: PRJNA643830
- Whole-genome resequencing data of all 453 samples in form of unmapped BAM files at ENA: PRJEB38173
- Transcriptomic data at NCBI SRA: PRJNA635556
- Geometric morphometric data, information on samples, and downstream data to reproduce our results can be downloaded from Dryad (10.5061/dryad.bcc2fqz91).
- Custom code used for the genome assembly (https://github.com/MartinPippel/DAmar) as well as custom code for genomic and morphometric analyses (https://github.com/alexnater/midas-genomics) can be accessed on GitHub.

**Randomization**

Within groups (i.e. lakes/species/morphs) individuals included in this study were sampled randomly. Groups were defined based on sampling location (lake), species characteristics, and morph (i.e. gold or dark; thin- or thick-lipped). As all subsequent analyses were performed on all individuals (for exceptions see Data exclusions above), no randomization was necessary. For mate choice experiments, the order of trials was randomized, but only one arena was available for the experiment.

**Blinding**

All measurements and morphometric data were collected blind to the identity of species and lake of origin of fish. Mate choice data was collected without knowledge of fish identities, as they were individually tagged with transponders, and only identified after they were taken out of any experiment (i.e. after pair formation and breeding).

Did the study involve field work?    ☒ Yes    ☐ No

# Field work, collection and transport

| Field conditions | Fish for whole genome resequencing were collected over a twelve year time period. Environmental conditions were not relevant (i.e. for sampling) and are therefore not reported here. |
| --- | --- |
| Location | Adult fish (standard length =14.54 ± 3.43 cm (mean ± SD)) were collected in field expeditions of the Meyer lab to Nicaragua between 2003 and 2015. Fish were caught with gill nets or by harpooning in crater lakes Asososca León (12°26'08"N 86°39'50"W), Asososca Managua (12°08'15"N 86°18'55"W), Apoyeque (12°14'38"N 86°20'31"W), Apoyo (11°55'23"N 86°01'57"W), Masaya (11°58'17"N 86°06'53"W), Tiscapa (12°08'22"N 86°16'15"W), and Xiloá (12°13'16"N 86°19'16"W) at water depths between 0 and 5m. Fish or tissue samples from the great lakes Nicaragua and Managua were obtained from local fishermen, mostly the big fish market in Granada (Lake Nicaragua fish; 11°56'00"N 85°57'20"W) and Mateares (Lake Managua fish; 12°14'10"N 86°25'48"W). |
| Access and import/export | All field work and export of samples was approved by the local authorities, the Ministerio del Ambiente y los Recursos Naturales (MARENA), Nicaragua (permit numbers DGRNB-ACHL-0078, DGRNB-IC-006-2007, No. 026-11007/DGAP, DGPN/DB-27-2010, DGPN/DB/DAP-IC-0003-2012, DGPN/DB-02-2012, DGPN/DB-IC-004-2013, DGPN/DB-011-2014, DGPN/DB-IC-015-2015). |
| Disturbance | Midas cichlids are very abundant and a common food source in Nicaragua. The extractions of a limited number of animals for this study is therefore very unlikely to have caused any kind of disturbance of this natural system. |

# Reporting for specific materials, systems and methods

We require information from authors about some types of materials, experimental systems and methods used in many studies. Here, indicate whether each material, system or method listed is relevant to your study. If you are not sure if a list item applies to your research, read the appropriate section before selecting a response.

## Materials & experimental systems

| n/a | Involved in the study |
| --- | --- |
| ☒ | Antibodies |
| ☒ | Eukaryotic cell lines |
| ☒ | Palaeontology |
| ☐ | ☒ Animals and other organisms |
| ☒ | Human research participants |
| ☒ | Clinical data |

## Methods

| n/a | Involved in the study |
| --- | --- |
| ☒ | ChIP-seq |
| ☒ | Flow cytometry |
| ☒ | MRI-based neuroimaging |

# Animals and other organisms

Policy information about studies involving animals; ARRIVE guidelines recommended for reporting animal research

| Laboratory animals | Laboratory animals were bred and raised in the animal research facility of the University of Konstanz. We performed two different (sub)experiments. The first one with fish from the same lake and the second with fish from different lakes. Both of the experiments consisted of two trials (experimental groups) each. In the first trial, 114 adult fish (>2 years old) were available. A total of 48 of them were tested, comprising the limnetic species A. zaliosus (nfemales = 25 available (13 tested), nmales = 13 (12)) and the benthic species A. astorquii (nfemales = 30 (10), nmales = 46 (13)) from CL Apoyo. In the second trial, 115 (33 tested) adult fish were available, comprising the limnetic species A. sagittae (nfemales = 26 (6), nmales = 17 (6)) and the benthic species A. amarillo (nfemales = 12 (11), nmales = 60 (10)) from CL Xiloá. The trials were terminated when no more naïve fish of a certain sex and species were available for replacement (indicated as the final number of tested fish above). In the second (sub) experiment the first trial was conducted with females from CL Xiloá (nA. sagittae = 26 available (12 finally tested), nA. amarillo = 12 (10)) and males from CL Apoyo (nA. zaliosus = 13 (11), nA. astorquii = 30 (9)), and the second trial with females from CL Apoyo (nA. zaliosus = 25 (21), nA. astorquii = 36 (12)) and males from CL Xiloá (nA. sagittae = 22 (13), nA. amarillo = 36 (18)). All animals were kept at a constant 12:12 light-dark cycle and at a constant temperature of 28 degrees celsius. At no time were fish held on any kind of caloric restrictions. |
| --- | --- |
| Wild animals | This study did not involve experimentation on wild animals. |
| Field-collected samples | Laboratory strains of Midas cichlids were established prior to this study. Wild-caught animals were (as reported above under laboratory animals) also kept at a constant 12:12 light-dark cycle and at a constant temperature of 28 degrees celsius. At no time were fish held on any kind of caloric restrictions. |
| Ethics oversight | Euthanization of animals, animal husbandry, and mate choice experiments were approved by the German authorities (permit numbers T-16/13 and G-15/89, Regierungspräsidium Freiburg, Abteilung 3, Referat 35, Veterinärwesen & Lebensmittelüberwachung, Germany). |

Note that full information on the approval of the study protocol must also be provided in the manuscript.

