## [Peer Review File · Nature]

Manuscript Title:**Editorial Notes:****Reviewer Comments & Author Rebuttals****Reviewer Reports on the Initial Version:**Referee #1 (Remarks to the Author):

The authors have gathered a massive data set, including a reference genome sequence for the Midas cichlid, and resequencing data for 453 fish from a number of Nicaraguan lakes. The thesis of this manuscript is that sympatric speciation in two of these crater lakes (Apoyo and Xilola) has been driven by selection on traits that have a polygenic genetic architecture. This proposal contrasts with most theory, that suggests that selection dispersed over many loci is likely too weak to prevent gene flow.

A clean test of this hypothesis would be to perform QTL analyses for various traits in laboratory crosses in order to characterize the 'genetic architecture' of the traits. This was not done. While it is clear that the gold coloration and fat lip phenotypes are controlled by 1-2 genes, the authors did not independently ascertain the genetic architecture of body or pharyngeal shape. Instead they infer a polygenic architecture because they were not able to identify GWAS signals for these traits in fish collected from the wild. This is a weak argument.

The main objective finding of the paper seems to be a high level of F_{st} differentiation among species within lakes Apoyo and Xilola. This differentiation is maintained despite the identification of a significant number of hybrids. The authors infer that selection against the hybrids must be strong in order to maintain this differentiation. I would offer an alternative hypothesis that the high F_{st} values represent differentiation by drift in these relatively small populations. In general, there is a failure to evaluate alternative hypotheses throughout the manuscript.

In sum, there is a lot of interesting data described in the paper. But I do not agree that they have "identified the most important traits related to sympatric divergence in each lake", or that they have demonstrated that "multifarious polygenic selection can drive the build-up of persistent combinations of alleles until genomic 'tipping points' are reached and speciation unfolds".

Referee #2 (Remarks to the Author):

The authors of the manuscript "Contrasting signatures of genomic divergence in rapidly speciating crater lake cichlid fishes" describe detailed insight in these fascinating fish populations including a first chromosome level assembly. They leverage this assembly to remap 453 samples for four different subpopulations of Midas cichlid species and investigate the differences in the genome to the phenotypic differences. Overall, I found the paper very well written and interesting. As I cannot judge the biology side too much, I will give comments on the computational and sequencing side of the paper. The approach and methods that they choose are state of the art. In the following, I list my comments/suggestions without special order:

1. As a minor point: I am missing the method to realign the PacBio reads to the assembled scaffold was it blasR / minimap2 ??
2. Please also report the version of the programs used. You did that for some but not for other essential programs (e.g. MARVEL)
3. Given the "shorter" read length (N50 ~14kbp) for the PacBio libraries, I would assume there are multiple gaps in the scaffold. Can you report the number of gaps you have in your scaffolds? Have you tried a gap filling method?
4. Did you run a Busco analysis to assess/report on the overall quality of your assembly?
5. It would be further interesting to the reader to report the contig N50 values.

6. Data availability section is missing links/details (e.g. Github could have been at least put there!).
7. How many SNV / SV were identified in the 453 individuals? I am missing standard descriptions e.g. about MAF spectrum, types and frequency of mutations etc. What is the rate of heterozygous vs homozygous variations etc.
8. Minor: The genome assembler is cited by two seemingly biological papers. I was wondering if that is a mistake.
9. I am maybe missing it, but I didn't see a clear description/ assessment on the quantity and quality of the RNA-Seq for annotation. From the text, I assume you had 12 different samples. What read length /coverage where they sequenced with, did they include replicates? What were the intermediate results on the RAN-Seq assemblies? How many RNA-Seq data sets were prepared? Trinity typically report transcripts, how where these processed?

Referee #3 (Remarks to the Author):

In this fascinating and important work the authors report the results of a de novo genome assembly and extensive resequencing, combined with large morphological and behavioral data sets, for the Nicaraguan lakes comprising the Midas Cichlid radiation. They find wide variation in the degree of genomic divergence within and between lakes. Within water bodies, there is more genomic divergence in two lakes where two traits with a polygenic architecture are the most conspicuously divergent and possibly the foci of reproductive isolation, in contrast to two (mainly, others were also examined) other lakes where reproductive isolation centers on traits mediated by allelic differences at one or two genes. From this result they infer that oligogenic traits under divergent selection and for which mating is assortative do not necessarily enable speciation to completion or even initiation, whereas multifarious ecological selection on phenotypic traits with polygenic architectures leads to more extensive genomic divergence and more complete reproductive isolation.

The scale and thoroughness of the work is impressive, but even so it is difficult to come to firm general conclusions with the number of lakes and lineages involved. Thus the case for publication in Nature rests more on the novelty and unexpectedness of the findings, for a difficult problem, than on this being a definitive result (at least with regard to the general point about trait architecture and speciation).

Inevitably in a nonexperimental study there is the problem of potentially confounding variables. In particular, the relative importance of polygenic trait architectures vs. multifarious selection in facilitating genomic divergence in the multi-species crater lakes is unclear. The authors rightly emphasize what they do know with confidence, which is that oligogenic traits under divergent selection and for which mating is assortative do not necessarily enable speciation, but rapid apparently sympatric speciation can occur in this system. I think it would be helpful to be clearer about the limitations of our understanding of the importance of trait architecture vs. more multifarious selection (or unknown factors) in the broader divergence seen in the multi-species crater lakes. Or if they think they can distinguish these effects, that would be interesting to see clearly presented.

And I wonder why thick-lipped fish are absent from the multi-species crater lakes? That is surprising.

Detailed comments:

85: is gold color really a key ecological trait?

101-104: This sentence could be dropped.

114: The term loci here does not seem quite what is needed as that is usually associated with a gene, more or less. Maybe genome regions or sequences? But if this is a common usage that I have overlooked, OK.

132: varied strongly – awkward wording—maybe emphasize varying 20 fold

291: should probably cite Schluter and colleagues for this design, as in the SI

332: “usually” might be stronger than appropriate; could say “often” or “widely”

342: new environments may also lead to breakdown of assortative mating.

351: at least the ones we know about

352: and/or

976-979: Any citation for this approach?

1062: how many different families, and sires/dams were used, assuming these were laboratory crosses? The lack of replication is unfortunate and it might be appropriate to report these as preliminary (or somehow indicate the limitation of the work) in the main text. Additional details on the tank, lighting and backgrounds would be helpful.

1068: how many families in this experiment?

Fig.1: Really useful. And if the colors could match even more exactly across the different parts of the figure, that would be helpful.

Fig. 2: Nice if the titles within each figure could indicate how many species are in each lake, e.g. GL Nicaragua (2 sp), at least for the multi-species lakes

Extended Data Fig. 7p: are all the labels correct?

Author Rebuttals to Initial Comments:

Reviewer 1

R1.1: The authors have gathered a massive data set, including a reference genome sequence for the Midas cichlid, and resequencing data for 453 fish from a number of Nicaraguan lakes. The thesis of this manuscript is that sympatric speciation in two of these crater lakes (Apoyo and Xilola) has been driven by selection on traits that have a polygenic genetic architecture. This proposal contrasts with most theory, that suggests that selection dispersed over many loci is likely too weak to prevent gene flow.

Our response: Thank you for your appreciation of the large amount of data that was collected for our study and the assessment of its importance.

R1.2: A clean test of this hypothesis would be to perform QTL analyses for various traits in laboratory crosses in order to characterize the 'genetic architecture' of the traits. This was not done. While it is clear that the gold coloration and fat lip phenotypes are controlled by 1-2 genes, the authors did not independently ascertain the genetic architecture of body or pharyngeal shape. Instead they infer a polygenic architecture because they were not able to identify GWAS signals for these traits in fish collected from the wild. This is a weak argument.

Our response: We agree that the lack of significant loci in our GWAS analysis is not a sufficiently strong argument for the presumed polygenic basis of body shape and pharyngeal jaw morphology. To provide independent evidence that both body shape and pharyngeal jaw morphology are polygenic, we added new QTL mapping analyses to this revised manuscript. We used an F₂ mapping panel of n=305 (one of the largest to date for cichlids) from an *Amphilophus astorquii* × *A. zalius* intercross from CL Apoyo – fish that differ in both focal phenotypic traits (Extended Data Fig. 4b, d). These new results strongly support the polygenic nature of both traits. Our final QTL model contains seven and five loci of small effect (2.0-6.7%) that jointly account for 29.8% and 22.7% of the phenotypic variance in body shape and maximum pharyngeal jaw tooth size, respectively. Owing to the inherent limitations (false-negative rate, Beavis effect, subsampling of alleles segregating in population) of QTL mapping experiments, it is likely that many additional small effect loci remain undetected with this approach. In fact, we did detect additional suggestive peaks for both traits. These new results are therefore a good fit to the patterns expected for polygenic traits with effect sizes that follow a gamma distribution. Both traits also carry many of the classic hallmarks of Fisherian polygenic traits, including being normally distributed in the cross and displaying both intermediate values, as well as increased variation in F₂ hybrids. We therefore think these substantial and complementary analyses provide compelling support for our statement that body and pharyngeal jaw shape are polygenic and contrast strongly with the simple genetic architectures of gold/dark coloration and lip size.

R1.3: The main objective finding of the paper seems to be a high level of F_{st} differentiation among species within lakes Apoyo and Xiloa. This differentiation is maintained despite the identification of a significant number of hybrids. The authors infer that selection against the hybrids must be strong in order to maintain this differentiation. I would offer an alternative hypothesis that the high F_{st} values represent differentiation by drift in these relatively small populations. In general, there is a failure to evaluate alternative hypotheses throughout the manuscript.

Our response: We agree with the reviewer that the relatively high F_{st} values among CL Apoyo and Xiloá species compared to the great lake species could in theory solely be explained by differences in genetic drift, as effective population sizes and particularly founder sizes differ strongly between great lakes and crater lakes. This was one of the main reasons why we additionally performed a genome-wide analysis for windows under divergent selection (Fig. 4j-q), which showed that numerous regions in the genome are under divergent selection among the sympatric species in CLs Apoyo and Xiloá. But we agree that this was not a direct test of the alternative hypothesis that drift alone can explain the observed F_{st} patterns. To address this issue and make better use of our extensive demographic modeling results, we have now conducted genome-scale coalescent simulations based on the inferred maximum-likelihood parameter estimates of our most supported demographic models (Extended Data Fig. 3, Extended Data Table 1). We did this for both great lakes species pairs and one species pair each in CLs Apoyo and Xiloá. Demonstrating the robustness of this approach, the distributions of F_{st} values obtained from these simulated data provide a very close fit to our empirical data (Extended Data Fig. 8). To test for the effect of genetic drift, we repeated the simulations, but replaced both the founder and current effective population sizes of both species in GL Nicaragua with those of the CL Apoyo species pair and vice versa. The same was done for GL Managua and the CL Xiloá species pair and we performed 100 independent simulations of entire genomes (matching our empirical sample sizes) followed by inferring the distribution of F_{st} values. These results show that the differences in effective population size alone cannot explain the observed differences in the empirical levels of genomic differentiation between great lake and crater lake species pairs (Extended Data Fig. 8). Moreover, we inferred that effective migration rates within lakes – a measure of overall reproductive isolation with lower effective migration implying stronger reproductive isolation – are consistently and significantly lower among each two species pairs in CLs Apoyo and Xiloá than the two species in both great lakes.

Based on these analyses, we can therefore reject the hypothesis that differences in genome-wide differentiation between great lakes and crater lakes species can be explained by differences in effective population sizes / drift alone. Consistent with our main conclusion, only differences in the strength of reproductive isolating mechanisms, such as mate choice and extrinsic postzygotic isolation, can explain our empirical findings.

R1.4: In sum, there is a lot of interesting data described in the paper. But I do not agree that they have “identified the most important traits related to sympatric divergence in each lake”, or that they have demonstrated that “multifarious polygenic selection can drive the build-up of persistent combinations of alleles until genomic ‘tipping points’ are reached and speciation unfolds”.

Our response: We are thankful for the comments that helped us identify points of contention in our main theses. We believe that in our revised version we present new data and analyses that provide additional support to our main conclusions. Briefly, the new QTL mapping data clearly demonstrate that a polygenic architecture underlies both body shape and pharyngeal jaw morphology. Moreover, newly conducted simulations confirm our previous conclusion that genome-wide reduction of effective gene flow was a crucial factor in causing the substantial differentiation among species in CL Apoyo and CL Xiloá as compared to great lake species. Although these new data and analyses strengthen our conclusions, we qualified some statements that might have sounded too bold in appreciation of the reviewer’s comments above.

The original statement “Having identified the most important traits related to sympatric divergence...” might have sounded too categorical and we therefore added a qualifier to this sentence “...(among the focal traits)...”. Surely, the four traits we focus on in this study (coloration, lip size, pharyngeal jaw type, and body shape) are not necessarily the only traits important for sympatric divergence in Midas cichlids. But, we focus on these traits, 1) because they are considered to be ecologically-relevant in other organisms in general – and thus important to consider for ecological speciation (coloration in general: e.g. Orteu & Jiggins 2020; hypertrophied lips in cichlids: e.g. Colombo et al. 2013, Baumgarten et al. 2015; pharyngeal jaws in cichlids: e.g. Liem 1973, Hulsey 2005; body shape in fish: e.g. Webb 1984,1988) – and 2) because these four traits had previously been hypothesized to be important for speciation in our system in particular (gold/dark: e.g. Elmer et al 2009, Kusche et al. 2015; hypertrophied lips: e.g. Machado-Schiaffino et al. 2017; pharyngeal jaws: e.g. Meyer 1990; body shape: e.g. Elmer et al. 2014, Raffini et al. 2019). Thus, there is strong evidence from this and other systems that the four focal traits in our study are of particular ecological relevance. Additionally, it is important to note that our findings of numerous, genome-wide signatures of divergent selection in CLs Apoyo and Xiloá did not depend on our choice of these focal traits, as these analyses are completely phenotype-independent.

Regarding our statement that “multifarious polygenic selection can drive the build-up of persistent combinations of alleles until genomic ‘tipping points’ are reached and speciation unfolds”, we acknowledge that it is difficult to come to irrefutable conclusions — despite the amount and diversity of data that we collected and although we are probably analyzing one of the best-suited systems to address this question. We acknowledge these limitations of our study in more detail in this revised version. Nonetheless, our main conclusion remains that polygenic selection is the most parsimonious explanation for stable sympatric speciation in CLs Apoyo and

Xiloá. This conclusion is now strengthened by the additional data and analyses we provide that 1) confirm a polygenic basis for pharyngeal jaw and body morphology and 2) rule out neutral processes (genetic drift) as an alternative hypothesis.

References:

- Orteu, A., Jiggins, C.D. The genomics of coloration provides insights into adaptive evolution. *Nat Rev Genet* (2020). <https://doi.org/10.1038/s41576-020-0234-z>
- Colombo M, Diepeveen ET, Muschick M, Santos ME, Indermaur A, Boileau N, Barluenga M, Salzburger W. 2013. The ecological and genetic basis of convergent thick-lipped phenotypes in cichlid fishes. *Molecular Ecology*, 22: 670–684.
- Baumgarten, L., Machado-Schiaffino, G., Henning, F. and Meyer, A. (2015), Adaptive function of hypertrophied lips. *Biol J Linn Soc Lond*, 115: 448–455
- Karel F. Liem (1973). Evolutionary Strategies and Morphological Innovations: Cichlid Pharyngeal Jaws, *Systematic Biology*. 22:425–441
- Hulsey, C.D. (2005). Function of a key morphological innovation: fusion of the cichlid pharyngeal jaw. *Proc. R. Soc. B*. 273:669–675
- Webb, P. W. 1984. Body form, locomotion and foraging in aquatic vertebrates. *Am. Zool.* 24:107–120.
- Webb, P. W. 1988. Simple physical principles and vertebrate aquatic locomotion. *Am. Zool.* 28:709–725.
- Elmer, K.R., Lehtonen, T.K. and Meyer, A. (2009), COLOR ASSORTATIVE MATING CONTRIBUTES TO SYMPATRIC DIVERGENCE OF NEOTROPICAL CICHLID FISH. *Evolution*, 63: 2750–2757.
- Kusche H., Elmer K., & Meyer A. (2015). Sympatric ecological divergence associated with a color polymorphism. *BMC Biology*, 13,82
- Machado-Schiaffino, G., Kautt, A.F., Torres-Dowdall, J., Baumgarten, L., Henning, F. and Meyer, A. (2017), Incipient speciation driven by hypertrophied lips in Midas cichlid fishes? *Mol Ecol*, 26: 2348–2362.
- Meyer A (1990). Ecological and evolutionary consequences of the trophic polymorphism in *Cichlasoma citrinellum* (Pisces: Cichlidae), *Biological Journal of the Linnean Society*. 39:279–299
- Elmer, K., Fan, S., Kusche, H. et al. (2014). Parallel evolution of Nicaraguan crater lake cichlid fishes via non-parallel routes. *Nat Commun* 5, 5168.
- Raffini, Francesca, et al. "Diving into divergence: Differentiation in swimming performances, physiology and gene expression between locally-adapted sympatric cichlid fishes." *Molecular ecology* (2019).

Reviewer 2

R2.1: The authors of the manuscript “Contrasting signatures of genomic divergence in rapidly speciating crater lake cichlid fishes” describe detailed insight in these fascinating fish populations including a first chromosome level assembly. They leverage this assembly to remap 453 samples for four different subpopulations of Midas cichlid species and investigate the differences in the genome to the phenotypic differences. Overall, I found the paper very well written and interesting. As I cannot judge the biology side too much, I will give comments on the computational and sequencing side of the paper. The approach and methods that they choose are state of the art. In the following, I list my comments/suggestions without special order:

Our response: We thank the reviewer for this very positive comment on our approaches and methods, and for highlighting that our paper is interesting and well written.

R2.2: As a minor point: I am missing the method to realign the PacBio reads to the assembled scaffold was it blasR / minimap2 ??

Our response: We used PacBio’s palign tool (github commit: 0669a4e), which internally uses blasr to map PacBio raw reads back to the scaffolds. We added this information to the Material & Methods section.

R2.3: Please also report the version of the programs used. You did that for some but not for other essential programs (e.g. MARVEL)

Our response: Thank you for pointing this out. It seems we had inadvertently missed to report this in the previous version. Below is a summary of the programs and versions that we used for the genome assembly. This information has also been added to the relevant sections of the manuscript.

Program	Version
Marvel (inhouse version)	https://github.com/MartinPippel/DAMAR
samtools	1.8
blasr	5.3.2-a579bd5
palign	github commit: 0669a4e (https://github.com/PacificBiosciences/palign)
bamtools	2.5.1
pbindex	0.18.0
DAZZ_DB	github commit: 0bd5e07 (https://github.com/thegenemyers/DAZZ_DB)
DAMASKER	github commit: bc7e49c (https://github.com/thegenemyers/DAMASKER)
arrow	github commit: c92ef5d (https://github.com/PacificBiosciences/GenomicConsensus)
bwa	0.7.17-r1188
Picard	2.10.9-SNAPSHOT

Freebayes	1.1.0
Bcftools	1.7
3d-dna	github commit: 5baf854 (https://github.com/theaidenlab/3d-dna)
Juicer_tools	1.7.6
Bionano Solve	3.1

R2.4: Given the “shorter” read length (N50 ~14kbp) for the PacBio libraries, I would assume there are multiple gaps in the scaffold. Can you report the number of gaps you have in your scaffolds? Have you tried a gap filling method?

Our response: Yes, it is correct that there are multiple gaps in the scaffolds. The final scaffolds consisted of 8,683 contigs and the cumulative number of gaps is: 30,847,507 bp. A gap filling step was indirectly performed in the final arrow scaffold polishing round: the PacBio polishing tool Arrow, which is part of PacBio’s GenomicConsensus github project (<https://github.com/PacificBiosciences/GenomicConsensus>), is creating a consensus sequence based on the alignment piles of the scaffolds and all PacBio raw reads. In case alignment piles span gap regions, Arrow will close them. The following table shows the contig Nx values, before and after applying Arrow.

	#contigs	N10	N20	N30	N40	N50	N60	N70	N80	N90
Before Arrow	9521	9.95	8.05	5.53	4.38	3.38	2.41	1.62	0.90	0.18
After Arrow	8632	11.46	8.26	6.30	4.98	3.84	2.60	1.80	1.02	0.22

R2.5: Did you run a Busco analysis to assess/report on the overall quality of your assembly?

Our response: Yes, we did perform such an analysis and we believe it speaks favorably to the very high quality of our assembly with 99% of core genes being present in our annotation. The details can be found in the methods section: *“Completeness of the two gene sets was assessed with gVolante v.1.2.159, using the ortholog search pipeline BUSCO v2/v360 and 233 Core Vertebrate Genes (CVG) as the reference gene set. The two sets of annotations captured 230 (98.71%) and 231 (99.14%) complete core genes, respectively.”*

R2.6: It would be further interesting to the reader to report the contig N50 values.

Our response: The contig N50 value was 3.84 Mb. Please see also our more detailed response above (#2.4).

R2.7: Data availability section is missing links/details (e.g. Github could have been at least put there!).

Our response: We have now prepared a comprehensive Github repository containing all of our scripts (<https://github.com/alexnater/midas-genomics>) and <https://github.com/MartinPippel/DAMar>). Raw and intermediate data to reproduce our results have been uploaded to a Dryad repository (<https://datadryad.org/stash/share/pET5RRe8nNtwD30pSFRyuYigoZPTjMon4yXUwraJ2wY>) please note that we did not include the ~50GB VCF file yet, as the data can only be provided as one single 'reviewer file').

R2.8: How many SNV / SV were identified in the 453 individuals? I am missing standard descriptions e.g. about MAF spectrum, types and frequency of mutations etc. What is the rate of heterozygous vs homozygous variations etc.

Our response: In total, we called 7,560,356 SNPs and 597,215 indels. This information has been added to the Methods section. We have also uploaded a summary (vt_peek suffix) that shows the breakdown of variant calls in detail (e.g. bi-allelic, tri-allelic, multi-allelic, clumped variants, complex substitutions) to the accompanying Dryad repository. Moreover, we have estimated the levels of nucleotide diversity (θ pi) for all species and lake populations and added these estimates to Extended Data Fig. 1a. We thank the reviewer for the suggestion, as we agree that adding this information will hopefully be useful and easy to use for the community, for example for comparative studies.

R2.9: Minor: The genome assembler is cited by two seemingly biological papers. I was wondering if that is a mistake.

Our response: The Marvel genome assembler was never published in an algorithmic-focused journal. It was mainly developed for large and repeat-rich genomes and was tested and validated on *Ambystoma mexicanum* (Nowoshilow et al. 2018) and *Schmidtea mediterranea* (Grohme et al. 2018). Both publications were published side-by-side in Nature and contain the complete assembly pipeline as well as the general algorithmic structure of the Marvel assembler in the supplementary information, which is why we decided to cite them.

References:

Nowoshilow, S. *et al.* The axolotl genome and the evolution of key tissue formation regulators. Nature 554, 50 (2018).

Grohme, M. A. *et al.* The genome of *Schmidtea mediterranea* and the evolution of core cellular mechanisms. Nature 554, 56 (2018).

R2.10: I am maybe missing it, but I didn't see a clear description/ assessment on the quantity and quality of the RNA-Seq for annotation. From the text, I assume you had 12 different samples. What read length /coverage where they sequenced with, did they include replicates?

What were the intermediate results on the RNA-Seq assemblies? How many RNA-Seq data sets were prepared? Trinity typically report transcripts, how were these processed?

Our response: We apologize for the limited information we provided here before. For the annotation, we used a total of 194 samples including seven types of tissues. The data is composed of new (that will be uploaded to the European Nucleotide Archive; PRJEB38173) and published (Franchini et al., 2016 and 2019; Manousaki et al. 2013) transcriptomes. We had between 15 and 42 replicates per tissue. In the revision, we added more detailed information (see below) that we also added to our dryad repository (10.5061/dryad.bcc2fqz91). We obtained a total of 521,031 transcripts using the *de novo* assembly algorithm implemented in Trinity and 637,341 transcripts using the genome-guided approach, again using Trinity. The two Trinity assemblies (minimum transcript length 200 bp) were combined and processed in order to remove poly-A tails and other contaminant sequences. Additionally, we reconstructed 77,685 transcripts using the program Cufflinks by parsing the alignments obtained by mapping the whole RNA-Seq dataset against the Midas genome. The three different transcript sets were used as input files in the “annotation comparison and update” step of the PASA pipeline, that allowed us to refine intron/exon boundaries and add UTRs to the EvidenceModeler gene prediction (the detailed workflow is described in the Extended Methods). We also added the FASTA files of the three transcriptome assemblies to the dryad repository (10.5061/dryad.bcc2fqz91).

Tissue	Stage	Type	Total Reads	Mean read length	n	Reference
Whole body	1 day	PE	198,655,568	142	15	Franchini et al. 2019
Whole body	1 month	PE	217,708,953	144	30	Franchini et al. 2016
Whole body	3 months	PE	1,982,034,127	144	27	this study
Eye	Adult	PE	80,909,046	146	38	this study
Lips	Adult	SE	204,959,450	69	42	Manousaki et al. 2013
Pharyngeal jaw	Adult	PE	857,131,978	135	24	this study
Scales	Adult	PE	1,108,817,455	141	18	this study
Total			4,650,216,577			

References:

- Franchini P *et al.* (2019) MicroRNA Gene Regulation in Extremely Young and Parallel Adaptive Radiations of Crater Lake Cichlid Fish. *Molecular Biology and Evolution* 36, 2498-2511.
- Franchini P, Xiong P, Fruciano C, Meyer A (2016) The role of microRNAs in the repeated parallel diversification of lineages of Midas cichlid fish from Nicaragua. *Genome biology and evolution* 8, 1543-1555.
- Manousaki T *et al.* (2013) Parsing parallel evolution: ecological divergence and differential gene expression in the adaptive radiations of thick-lipped Midas cichlid fishes from Nicaragua. *Molecular Ecology* 22, 650-669.

Reviewer 3

R3.1: In this fascinating and important work the authors report the results of a de novo genome assembly and extensive resequencing, combined with large morphological and behavioral data sets, for the Nicaraguan lakes comprising the Midas Cichlid radiation. They find wide variation in the degree of genomic divergence within and between lakes. Within water bodies, there is more genomic divergence in two lakes where two traits with a polygenic architecture are the most conspicuously divergent and possibly the foci of reproductive isolation, in contrast to two (mainly, others were also examined) other lakes where reproductive isolation centers on traits mediated by allelic differences at one or two genes. From this result they infer that oligogenic traits under divergent selection and for which mating is assortative do not necessarily enable speciation to completion or even initiation, whereas multifarious ecological selection on phenotypic traits with polygenic architectures leads to more extensive genomic divergence and more complete reproductive isolation.

Our response: We appreciate the excellent summary and are glad that the reviewer deems our work important and fascinating.

R3.2: The scale and thoroughness of the work is impressive, but even so it is difficult to come to firm general conclusions with the number of lakes and lineages involved. Thus the case for publication in Nature rests more on the novelty and unexpectedness of the findings, for a difficult problem, than on this being a definitive result (at least with regard to the general point about trait architecture and speciation).

Inevitably in a nonexperimental study there is the problem of potentially confounding variables. In particular, the relative importance of polygenic trait architectures vs. multifarious selection in facilitating genomic divergence in the multi-species crater lakes is unclear. The authors rightly emphasize what they do know with confidence, which is that oligogenic traits under divergent selection and for which mating is assortative do not necessarily enable speciation, but rapid apparently sympatric speciation can occur in this system. I think it would be helpful to be clearer about the limitations of our understanding of the importance of trait architecture vs. more multifarious selection (or unknown factors) in the broader divergence seen in the multi-species crater lakes. Or if they think they can distinguish these effects, that would be interesting to see clearly presented.

Our response: It is a difficult task to determine whether the wide-spread signatures of divergent selection that we observe are solely due to selection acting only on one or a few traits with polygenic bases or potentially multifarious selection on (so far) additional undetected traits. It is, in fact, likely that selection is acting on traits that remain so far unknown (also including physiology and behavior). We want to note that we believe that, conceptually, it does not make a difference whether selection is acting on one trait that is influenced by many loci or a

combination of traits that are each influenced by fewer loci (Feder & Nosil 2010). Our empirical data show that divergent selection is more prevalent among species in CLs Apoyo and Xiloá compared to the great lakes and lip ecotypes. The fact that the focal traits, which are of particular ecological relevance, differ in their genetic architecture is one factor that explains these contrasting signatures, but it is certainly possible (and likely) that some of the widespread signatures of divergent selection are related to traits other than body shape and pharyngeal jaw type.

To better acknowledge this fact, we have changed our previous statement “Multifarious selection acting on a suite of polygenic traits seems therefore most effective at building up and maintaining genomic differentiation of cichlids in sympatry.” to: “Divergent selection affecting a large number of loci across the genome (Fig. 4r–y) — by acting, for example, on one or several traits with a polygenic basis or a combination of multiple traits each with a simpler genetic basis (i.e. multifarious selection) — seems therefore most effective at building up and maintaining genomic differentiation of Midas cichlids in sympatry.” and added the following sentence to place our results in a broader context: “This is in line with the notion that highly polygenic barriers likely underlie the maintenance of *Heliconius* butterfly species and with simulations suggesting that speciation-with-gene-flow may often require selection acting on many unlinked genes.”. The second instance of the use of “multifarious” in the final concluding paragraph of our manuscript “...multifarious, polygenic selection can drive the build-up of persistent combinations of alleles likely until genomic “tipping points” are reached and speciation unfolds.” has now been deleted so that it now reads “...polygenic selection can drive...”

References:

Feder JL, Nosil P. (2010). The efficacy of divergence hitchhiking in generating genomic islands during ecological speciation. *Evolution*.4(6):1729-1747

R3.3: And I wonder why thick-lipped fish are absent from the multi-species crater lakes? That is surprising.

Our response: Thick-lipped fish are actually not only absent from the two multi-species crater lakes but also several other single-species crater lakes (As. León, As. Managua, Tiscapa). Founder effects are a possible explanation as thick-lipped fish are much more rare in the source lakes than thin-lipped fish, but any explanation would be purely speculative at this point, which is why we think it is better to refrain from providing any. But this is certainly a very interesting question that we plan to address in the future.

R3.4: 85: is gold color really a key ecological trait?

Our response: Gold/dark coloration has previously been implicated to play a role in social behavior (dominance and aggression; Barlow 1973 and 1983) and risk of predation (Kusche et

al. 2014, Torres-Dowdall et al. 2014, Torres-Dowdall et al. 2017) and can thus be considered ecologically relevant. We have added references suggesting that gold/dark coloration is ecologically relevant to the manuscript. Nonetheless, to acknowledge that our current understanding of the ecological relevance of gold/dark coloration is incomplete, we have also decided to tone down our statement to “putatively ecologically relevant traits”.

References:

- Barlow, George W. "Competition between color morphs of the polychromatic Midas cichlid *Cichlasoma citrinellum*." *Science* 179.4075 (1973): 806-807.
- Barlow, George W. "The benefits of being gold: behavioral consequences of polychromatism in the midas cichlid, *Cichlasoma citrinellum*." *Predators and prey in fishes*. Springer, Dordrecht, 1983. 73-85.
- Kusche, H, Meyer, A. 2014. One cost of being gold: selective predation and implications for the maintenance of the Midas cichlid colour polymorphism (Perciformes: Cichlidae). *Biological Journal of the Linnean Society* 111: 350–358.
- Torres-Dowdall J et al. (2014). Differential predation on the two colour morphs of Nicaraguan Crater lake Midas cichlid fish: implications for the maintenance of its gold-dark polymorphism. *Biological Journal of the Linnean Society* 112:123–131.
- Torres-Dowdall J et al. (2017). The role of rare morph advantage and conspicuousness in the stable gold-dark colour polymorphism of a crater lake Midas cichlid fish. *J Anim Ecol.* 86: 1044–1053

R3.5: 101-104: This sentence could be dropped.

Our response: Thanks for the suggestion. Sentence has been dropped.

R3.6: 114: The term loci here does not seem quite what is needed as that is usually associated with a gene, more or less. Maybe genome regions or sequences? But if this is a common usage that I have overlooked, OK.

Our response: We respectfully disagree with the reviewer here and decided to keep the term ‘loci’, because we believe that locus can refer to any kind of genomic position/region and it is frequently used for non-coding parts of the genome (e.g. microsatellite or RFLP loci). Sequences is not a good alternative in our opinion, as it might be confused with particular haplotypes at a given locus.

R3.7: 132: varied strongly – awkward wording—maybe emphasize varying 20 fold

Our response: We agree that the wording was awkward and have removed ‘strongly’.

R3.8: 291: should probably cite Schluter and colleagues for this design, as in the SI

Our response: Done. Thank you for the suggestion.

R3.9: 332: “usually” might be stronger than appropriate; could say “often” or “widely”

Our response: We agree and have changed this to “often”.

R3.10: 342: new environments may also lead to breakdown of assortative mating.

Our response: While we agree that new environments may lead to a breakdown of assortative mating, there is evidence for strong positive assortative mating in CL Apoyeque in the wild (Machado-Schiaffino et al. 2017). We emphasize this more clearly and write now “but this low genomic divergence also seems to break down easily and repeatedly, as evidenced by the populations of CLs Masaya and Apoyeque — despite evidence for strong assortative mating in their current environment (Extended Data Table 2).”

References:

Machado-Schiaffino, G., Kautt, A.F., Torres-Dowdall, J., Baumgarten, L., Henning, F. and Meyer, A. (2017), Incipient speciation driven by hypertrophied lips in Midas cichlid fishes?. *Mol Ecol*, 26: 2348-2362.

R3.11: 351: at least the ones we know about

Our response: Agreed. In our revised version we are more specific and write “Moreover, our laboratory mate choice experiments suggest that divergence in their ecologically-relevant traits (at least the ones that have been previously linked to speciation in Midas cichlid fishes) is not coupled with assortative mating (Extended Data Table 2).”

R3.12: 352: and/or

Our response: Done. Changed to “demographic and/or environmental fluctuations”.

R3.13: 976-979: Any citation for this approach?

Our response: We are not aware of any other study that describes similar analyses, but in our opinion this was the best approach to identify the primary axes of genomic differentiation – the variables we are interested in – in the multi-species crater lakes in which standard PC axes do not necessarily fall along the axis of divergence of individual species. We mention this now more clearly in the Methods section. Below we show a plot of the first two axes of between-group PCAs for each species versus all other sympatric species to illustrate this approach. Note that PC1 in each comparison clearly separates an individual species from all other sympatric species.

R3.14: 1062: how many different families, and sires/dams were used, assuming these were laboratory crosses? The lack of replication is unfortunate and it might be appropriate to report these as preliminary (or somehow indicate the limitation of the work) in the main text. Additional details on the tank, lighting and backgrounds would be helpful.

Our response: We can see now that this information was not clearly presented in the manuscript. In the revised version, we provide detailed information in the Methods section and

provide further information in the Supplementary Materials. Briefly, we were restricted to using a single brood from wild-caught parents per species. The experimental tank was 340 x 160 x 80 cm in dimensions and fish were kept at a constant temperature of $28 \pm 1^\circ\text{C}$ and a 12:12 light/dark cycle.

We want to point out that these fish are very large (females were on average 11 cm and males were 20 cm long), and often aggressive and territorial, which makes rearing and conducting mate choice experiments on a large number of them logistically very challenging. To be able to carry out this experiment we had to rear more than 200 fish from four different species over a period of two years, so that they would all reach sexual maturity and were of comparable size/age at the time of the experiment. We also want to note that the results we obtained are replicated for a pair of benthic-limnetic species from two different crater lakes (Apoyo and Xiloá), and that they are in line with the other studies previously conducted in this system (Extended Data Table S2, Supplementary Materials). Nonetheless, we agree with the reviewer that the lack of replication and the use of a limited number of families is a limitation of this experiment, and we now acknowledge this more clearly in the manuscript.

R3.15: 1068: how many families in this experiment?

Our response: As described above, these fish are from a single brood per species from a couple of wild-caught fish. As also mentioned above, because of the large size, aggression, and territoriality of our fish, we were unfortunately limited by space constraints and the requirement to simultaneously rear a large number of fish of four different species to maturity, which took more than two years. We now acknowledge this limitation in the text.

R3.16: Fig. 1: Really useful. And if the colors could match even more exactly across the different parts of the figure, that would be helpful.

Our response: Thank you. We modified the colors to match now.

R3.17: Fig. 2: Nice if the titles within each figure could indicate how many species are in each lake, e.g. GL Nicaragua (2 sp), at least for the multi-species lakes

Our response: This will clearly help the reader. We added the number of species per lake.

R3.18: Extended Data Fig. 7p: are all the labels correct?

Our response: Thank you, some of the labels were indeed swapped. This has been corrected (now Extended Data Fig. 5).

Reviewer Reports on the First Revision:

Referee #2 (Remarks to the Author):

I thank the authors for addressing all my questions and concerns.

Referee #3 (Remarks to the Author):

The authors have responded effectively to my comments and I am now satisfied with the MS. I appreciate the clarifications of what their data can and cannot tell us with regard to multifarious selection and polygenic vs. oligogenic trait architectures and speciation. I think the value of the work has not been lessened as a result, and remains high. Similarly, the limited number of crosses in the mate choice work is unfortunate, but far from fatal for the MS, especially in light of the cumulative weight of the various related studies. In sum, key parts of the MS have been improved substantially, and the work was already very strong.

I would also note that the QTL analyses are a valuable addition.